# Lipidomic signatures align with inflammatory patterns and outcomes in critical illness

Junru Wu [1,2,3,4], Anthony Cyr [1,2], Danielle S. Gruen [1,2], Tyler C. Lovelace[5,6], Panayiotis V. Benos [5], Jishnu Das [7], Upendra K. Kar[1,2], Tianmeng Chen[1,8], Francis X. Guyette[9], Mark H. Yazer[10], Brian J. Daley [11], Richard S. Miller[12], Brian G. Harbrecht[13], Jeffrey A. Claridge[14], Herb A. Phelan[15], Brian S. Zuckerbraun[1,2], Matthew D. Neal [1,2], Pär I. Johansson[16], Jakob Stensballe[16,17,18], Rami A. Namas[1,2], Yoram Vodovotz [1,2], Jason L. Sperry[1,2] ✉, Timothy R. Billiar [1,2] ✉ & PAMPer study group*

Alterations in lipid metabolism have the potential to be markers as well as drivers of pathobiology of acute critical illness. Here, we took advantage of the temporal precision offered by trauma as a common cause of critical illness to identify the dynamic patterns in the circulating lipidome in critically ill humans. The major findings include an early loss of all classes of circulating lipids followed by a delayed and selective lipogenesis in patients destined to remain critically ill. The previously reported survival benefit of early thawed plasma administration was associated with preserved lipid levels that related to favorable changes in coagulation and inflammation biomarkers in causal modelling. Phosphatidylethanolamines (PE) were elevated in patients with persistent critical illness and PE levels were prognostic for worse outcomes not only in trauma but also severe COVID-19 patients. Here we show selective rise in systemic PE as a common prognostic feature of critical illness.

Acute critical illness is a major healthcare burden and commonly leads to short and long-term morbidity and mortality[1,2]. Common causes of acute critical illness, including severe injury and infections, are among the leading causes of death worldwide[3]. Most recently, the COVID-19 pandemic has emerged as a major etiology for acute critical illness and death. Patients hospitalized for SARS CoV-2 infection that develop critical illness have mortality rates up to 39%[4]. For those that develop organ dysfunction, treatment options are limited and those targeting the host response are often nonspecific. Common features across these different etiologies of critical illness include dysregulated

[1]Department of Surgery, University of Pittsburgh, Pittsburgh, PA, USA. [2]Pittsburgh Trauma Research Center, Division of Trauma and Acute Care Surgery, Pittsburgh, PA, USA. [3]Department of Cardiology, The 3rd Xiangya Hospital, Central South University, Changsha, China. [4]Eight-year program of medicine, Xiangya School of Medicine, Central South University, Changsha, China. [5]Department of Computational and Systems Biology, University of Pittsburgh, Pittsburgh, PA, USA. [6]Joint CMU-Pitt PhD Program in Computational Biology, Pittsburgh, PA, USA. [7]Center for Systems Immunology, Departments of Immunology and Computational & Systems Biology, University of Pittsburgh School of Medicine, Pittsburgh, PA, USA. [8]Cellular and Molecular Pathology Program, University of Pittsburgh School of Medicine, Pittsburgh, PA, USA. [9]Department of Emergency Medicine, Medicine, University of Pittsburgh, Pittsburgh, PA, USA. [10]The Institute for Transfusion Medicine, Pittsburgh, PA, USA. [11]Department of Surgery, University of Tennessee Health Science Center, Knoxville, TN, USA. [12]Department of Surgery, Vanderbilt University Medical Center, Nashville, TN, USA. [13]Department of Surgery, University of Louisville, Louisville, KY, USA. [14]Metro Health Medical Center, Case Western Reserve University, Cleveland, OH, USA. [15]Department of Surgery, University of Texas Southwestern, Dallas, TX, USA. [16]Section for Transfusion Medicine, Capital Region Blood Bank, Rigshospitalet, Copenhagen University Hospital, Copenhagen, Denmark. [17]Department of Anesthesia and Trauma Center, Centre of Head and Orthopaedics, Rigshospitalet, Copenhagen University Hospital, Copenhagen, Denmark. [18]Emergency Medical Services, The Capital Region of Denmark, Hillerød, Denmark. *A list of authors and their affiliations appears at the end of the paper. ✉e-mail: sperryjl@upmc.edu; billiartr@upmc.edu

metabolism, an inflammatory "genomic storm", immune suppression, and endothelial/ coagulation dysfunction[4–10]. The validation of accurate prognostic biomarkers and a better understanding of the pathobiology of acute critical illness would facilitate the identification of effective targeted therapies.

A limitation in the study of human critical illness is knowing the time of onset of the patient's disease process[9]. This is especially true for infections for which time of onset is often unclear. In addition, serious infections are commonly seen on the background of other chronic diseases that can confound interpretation of results. Traumatic injury is one of the most common causes of acute critical illness and often occurs in otherwise healthy individuals. This, coupled to the fact that the time of onset of the acute disease process can be known with precision, makes trauma an attractive model for the study of the dynamic events leading up to acute critical illness.

Lipids comprise 30% of the body's non-water mass and are not only a main component of cell membranes but also important energy substrates and signaling molecules[11]. Previous studies in critically ill humans provide evidence that lipolysis and lipogenesis are altered dramatically in acute critical illness. For example, circulating levels of glycerolipids, sphingolipids, phospholipids, and lyso-phospholipids vary from baseline in patients with acute critical illness[12–18]. Many lipids can serve as regulators of inflammatory and immune responses[19]. In addition, certain lipids (e.g. triacylglycerides, fatty acids) can serve as essential energy substrates for immune cell subsets (e.g. Memory T cells\Tregs\M2 Macrophages)[20,21]. However, a comprehensive assessment of the changes in common circulating lipids that correlate with outcomes in acute critical illness is lacking.

To define the changes in the circulating lipidome associated with acute critical illness, we utilized a database and biobank established during the Prehospital Air Medical Plasma (PAMPer) Trial[22]. This prospective, multi-institutional randomized trial enrolled severely injured patients transported to level I Trauma Centers by helicopter. The trial demonstrated that administration of thawed allogeneic plasma (TP) during transport improved 30-day survival when compared to standard-of-care, which does not include TP in the pre-hospital setting. Because of this striking treatment effect, we hypothesized that early TP administration would favorably impact circulating lipidomic patterns. Causal modeling was used to integrate the major changes in lipidomic profiles with immune mediator profiles and tissue injury/ coagulation markers observed after trauma and during critical illness. The lipidomic findings were further translated into a Lipid Reprogramming Score that was found to correlate highly with later patient outcomes. These findings were confirmed in a second trauma database and two publicly available databases comprised of critically ill COVID-19 patients, suggesting that some of the unique lipidomic patterns identified in this study may be generalizable to critical illness resulting from diverse etiologies.

## Results
### Lipid profiling of plasma from patients with severe trauma
To determine the dynamics changes in circulating lipids after severe injury in humans, we carried out a quantitative analysis of plasma lipid levels in samples obtained during the PAMPer trial[22]. This prospective, multi-institutional, pragmatic trial enrolled seriously injured humans suffering polytrauma at risk for hemorrhagic shock. Only patients that were transported by helicopter to a Level 1 trauma center were included and randomization took place in the pre-hospital setting. Patients in the treatment arm received two units of TP initiated during helicopter transport, while the control group was assigned randomly to standard-of-care, which did not include TP in the pre-hospital setting. The use of pre-hospital TP was associated with a 9.8% reduction in 30-day mortality ($p = 0.03$)[22]. A total of 193 of the original 523 patients were selected for lipidome analysis (Supplementary Fig. 1). This cohort included both non-survivors ($n = 83$) and survivors ($n = 110$) selected to

represent the overall cohort. Samples were obtained at admission to the trauma center (0 h) and at 24 and 72 h after admission. Only the time 0 h sample was obtained in the early (died within the 72 h) non-survivors ($n = 51$). A group of 17 non-fasting healthy subjects was used as controls for baseline values. The detailed demographic information of healthy subjects and patients is shown in Table 1. Since underlying medical conditions and medication history can influence circulating lipid profiles, we also provide this information (Supplementary Data 4). Chronic health conditions and medications were rare in the trauma patient population and evenly distributed across the outcome groups (Supplementary Data 1).

The overall data analysis workflow is shown in Fig. 1a. Liquid chromatography mass spectrometry (LC-MS) was used to carry out targeted lipidomic analysis on the plasma samples. In total, 996 lipids were quantified using internal standards. In the quality control analysis, the median relative standard deviation (RSD) for the lipid panel was 4%. Lipids are named according to sub-class and acyl chains detected. For example, PE (16:0_18:2) has a phosphatidylethanolamine (PE) backbone and two acyl chains comprised of palmitic acid (C16:0) and linoleic acid (C18:2). The representation of lipids from 14 sub-classes is shown in Fig. 1b. Triglyceride (TAG) (glycerol backbone + three acyl chains) was the most abundant lipid class identified in the plasma ($n = 518$). Phosphatidylethanolamine (PE), phosphatidylcholine (PC), and diacylglycerols (DAG) all containing 2 acyl chains were the next most abundant classes ($n = 128, 121, 58$ respectively).

We first explored the dynamic changes in the global pattern of the circulating lipidome in trauma patients. Uniform Manifold Approximation and Projection (UMAP) is a non-linear method for dimension reduction that can identify the global structure of multi-dimensional data. In Fig. 1c, each dot represents a single subject and the distance between dots in the UMAP plot reflects the global similarity/ differences in overall lipid profiles between samples[23]. We observed that trauma patients at 0 h were quite dispersed and partially overlapping with healthy subjects, suggesting an early and rapidly evolving response pattern immediately post-injury. There was excellent separation across the three time points on UMAP, underscoring the role of time in the major changes in lipid patterns after trauma.

To depict the differences between the healthy controls and patients across time, we projected relative levels of all lipids assayed on a heatmap (Fig. 1d). Compared to healthy controls, most lipid species were persistently lower after trauma. This dramatic shift between healthy controls and injured humans was also observed when total lipid concentrations were compared (Fig. 1e).

### Association between lipidome pattern and outcome of trauma patients
We next investigated the association between the circulating lipidome and patient outcomes. The three outcomes used for this analysis included (1) early non-survivors (death within 3 days of admission), (2) non-resolving patients (survivors with duration of intensive care unit [ICU] stay ≥7 days or patients that died after day 3 following admission), and (3) resolving patients (survivors with duration of ICU stay <7 days). UMAP plots of the global lipidomic patterns indicated enrichment of early non-survivors in the region encircled in red at 0 h and an enrichment of the non-resolving patients in the region encircled by the blue line at 72 h (Fig. 2a, b). Furthermore, we observed a dramatic drop in the levels of nearly all major lipid species at 0 h for early non-survivors compared to the other patient groups or healthy controls (Fig. 2c). Patients in both the resolving and non-resolving groups at 0 h also exhibited a drop in most lipid species compared to healthy controls, but not to the degree seen in the non-survivors. Patients in the resolving group exhibited a persistent suppression in most lipids at 24 and 72 h (Fig. 2d, e). Remarkably, patients in the non-resolving group at 72 h demonstrated an increase in a subset of lipids. Further characterization of lipid class and fatty acid types indicated

**Table 1 | Demographic characteristics of the patients by outcome**

| Variables | Healthy subjects (N = 17) | Resolving (N = 41) | Non-resolving (N = 101) | Early-nonsurvivors (N = 51) | p-value |
|---|---|---|---|---|---|
| **Demographics** | | | | | |
| Age (Median [IQR]) | 38 (±31) | 48 (±34) | 46 (±37) | 46 (±42) | 0.836[a] |
| Sex (% Male) | 12 (70.6%) | 31 (75.6%) | 78 (77.2%) | 36 (70.6%) | 0.809[a] |
| Race (% White) | | 35 (85.4%) | 89 (88.1%) | 48 (94.1%) | 0.365 |
| **Injury characteristics** | | | | | |
| ISS (Median [IQR]) | | 21 (±10) | 30 (±16) | 24 (±23) | <0.001 |
| Head AIS (Median [IQR]) | | 0 (±3.0) | 3.0 (±2.0) | 3.0 (±4.0) | <0.001 |
| TBI (%) | | 14 (34.1%) | 66 (65.3%) | 29 (56.9%) | 0.003 |
| GCS (Median [IQR]) | | 14 (±7.0) | 3.0 (±9.0) | 3.0 (±8.0) | <0.001 |
| SBP < 70 mmHg (%) | | 19 (46.3%) | 41 (40.6%) | 25 (49.0%) | 0.580 |
| HR (Median [IQR]) | | 120 (±16) | 120 (±21) | 120 (±39) | 0.218 |
| Injury type (% Blunt) | | 30 (73.2%) | 93 (92.1%) | 47 (92.2%) | 0.017 |
| **Prehospital** | | | | | |
| Treatment arm | | | | | |
| Standard care (%) | | 25 (61.0%) | 48 (47.5%) | 36 (70.6%) | 0.021 |
| TP (%) | | 16 (39.0%) | 53 (52.5%) | 15 (29.4%) | |
| Transport time (Median [IQR]) | | 39 (±18) | 44 (±17) | 42 (±18) | 0.771 |
| CPR (%) | | 0 (0%) | 3 (2.97%) | 5 (9.80%) | 0.044 |
| Intubation (%) | | 13 (31.7%) | 65 (64.4%) | 40 (78.4%) | <0.001 |
| Blood (%) | | 11 (26.8%) | 32 (31.7%) | 22 (43.1%) | 0.214 |
| Crystalloid (Median [IQR]) | | 800 (±1400) | 830 (±1300) | 1000 (±1600) | 0.891 |
| PRBC (Median [IQR]) | | 0 (±1.0) | 0 (±1.0) | 0 (±2.0) | 0.233 |
| **Hospital** | | | | | |
| Transfusion 24 h (Median [IQR]) | | 2.0 (±8.0) | 7.0 (±14) | 12 (±20) | <0.001 |
| PRBC 24 h (Median [IQR]) | | 2.0 (±5.0) | 5.0 (±7.0) | 8.0 (±10) | <0.001 |
| Plasma 24 h (Median [IQR]) | | 0 (±0) | 2.0 (±4.0) | 4.0 (±8.0) | <0.001 |
| Platelets 24 h (Median [IQR]) | | 0 (±0) | 0 (±1.0) | 1.0 (±2.0) | 0.002 |
| Crystalloid 24 h (Median [IQR]) | | 4800 (±3800) | 5300 (±4000) | 4600 (±3000) | 0.095 |
| Vasopressors 24 h (%) | | 19 (46.3%) | 68 (67.3%) | 44 (86.3%) | <0.001 |
| INR (Median [IQR]) | | 1.2 (±0.20) | 1.3 (±0.36) | 1.6 (±0.72) | <0.001 |
| **Other outcomes** | | | | | |
| Coagulopathy (%) | | 16 (39.0%) | 54 (53.5%) | 44 (86.3%) | <0.001 |
| ALI (%) | | 2 (4.88%) | 47 (46.5%) | 3 (5.88%) | <0.001 |
| NI (%) | | 3 (7.32%) | 43 (42.6%) | \ | <0.001 |
| MOF (%) | | 31 (75.6%) | 98 (97.0%) | \ | <0.001 |
| Vent days (Median [IQR]) | | 2.0 (±3.0) | 10 (±8.0) | 1.0 (±0) | <0.001 |
| ICU LOS (Median [IQR]) | | 4.0 (±3.0) | 13 (±9.0) | 1.0 (±1.5) | <0.001 |
| Hospital LOS (Median [IQR]) | | 9.0 (±10) | 19 (±19) | 1.0 (±1.0) | <0.001 |

Pearson's χ2 test was used for calculating p value of categorical variables. Kruskal-Wallis test was used for calculating p value of continuous variables.

*ISS* injury severity score, *AIS* abbreviated injury score, *TBI* traumatic brain injury, *GCS* Glasgow coma score, *SBP* systolic blood pressure, *HR* heart rate, *TP* thawed allogeneic plasma, *CPR* cardiopulmonary resuscitation, *PRBC* packed red blood cells, *INR* international normalized ratio, *ALI* acute lung injury, *NI* nosocomial infection, *MOF* multiple organ failure, *ICU* intensive care unit, *LOS* length of stay.

[a]Test was conducted across both healthy subjects and the three outcome groups of trauma patients.

that all 14 classes, including both saturated and unsaturated fatty acids, were suppressed in injured patients at 0 h. However, there was selective elevation of TAG, DAG, PE, and ceramides (CER) at 72 h in the non-resolving cohort. A quantitative time-series analysis showed that total lipid levels were higher at 72 h in the non-resolving patients and that unsaturated fatty acids predominated in TAG and DAG, while PE and CER contained a mixture of saturated and unsaturated fatty acids (Fig. 2f). Interestingly, there was excellent correlation across the elevated lipids from the TAG, DAG, and PE classes (Supplementary Fig. 3a, b). The interconnections between biochemical pathways involved in the synthesis of these lipid classes are shown in Supplementary Fig. 3c. Our findings point to a rapidly evolving

pattern in the circulating lipidome after severe injury that includes a loss of all classes of lipids in the circulation that is evident early after injury. This process is exaggerated in patients that die early, suggesting that an abrupt loss of circulating lipids contributes to adverse outcomes. Furthermore, there is a selective increase in four lipid classes by 72 h in patients that remain critically ill or die later in their clinical course.

We next examined the impact of injury severity reflected by injury severity scores (ISS) on lipid levels and profiles. Patients were separated into minimal (ISS < 10), moderate (ISS 10–25), or severe (ISS ≥ 25) injury (Supplementary Fig. 2a). Exploration of the lipid profiles by either UMAP or heatmap demonstrated no major impact of ISS on the

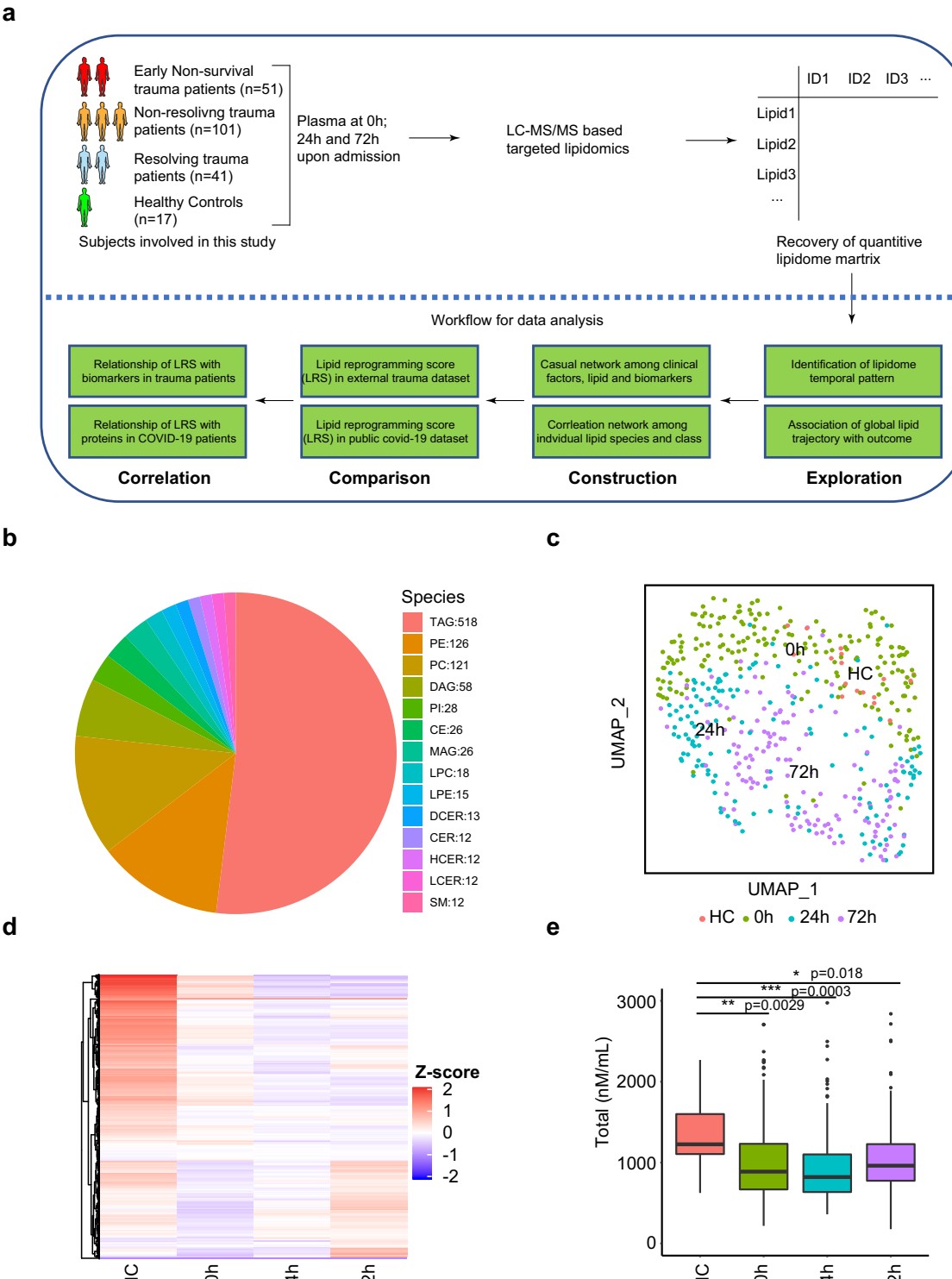

**Fig. 1 | Temporal patterns in the circulating lipidome after severe trauma.**
**a** Scheme of overall analysis strategy. **b** Representation of 996 lipid species detected in the lipidomic platform grouped by lipid classes. **c** Uniform Manifold Approximation and Projection (UMAP) plot shows the distribution of healthy subjects (*n* = 17) and patients with trauma (*n* = 193), grouped by sampling timepoints (0 h, 24 h, 72 h after admission). **d** Heatmap shows relative levels of 996 lipid species for healthy subjects and trauma patients, grouped by sampling timepoints using z-score normalized concentrations. Lipid species are clustered by Hierarchical clustering. **e** Quantitative comparison of circulating total lipid concentration among healthy controls (HC, *n* = 17) and trauma patients (*n* = 193), grouped by

sampling timepoints. Asterisks indicate statistical significance based on the Kruskal–wallis test with post-hoc analysis using the Dunn test. The *p* value was adjusted by the Benjamini–Hochberg method: *<0.05; **<0.01; ***<0.001. Box and whisker plots represent mean value, standard deviation, maximum and minimum values, and outliers. TAG triacylglycerol, DAG diacylglycerols, MAG mono-acylglycerols, PE phosphatidylethanolamine, PC phosphatidylcholine, PI phos-phatidylinositol, LPE Lysophosphatidylethanolamine, LPC Lysophosphatidylcholine, CER Ceramides, HCER hexosylceramides, LCER lacto-sylceramide, DCER dihydroceramides, CE cholesterol ester. Source data are provided as a Source Data file.

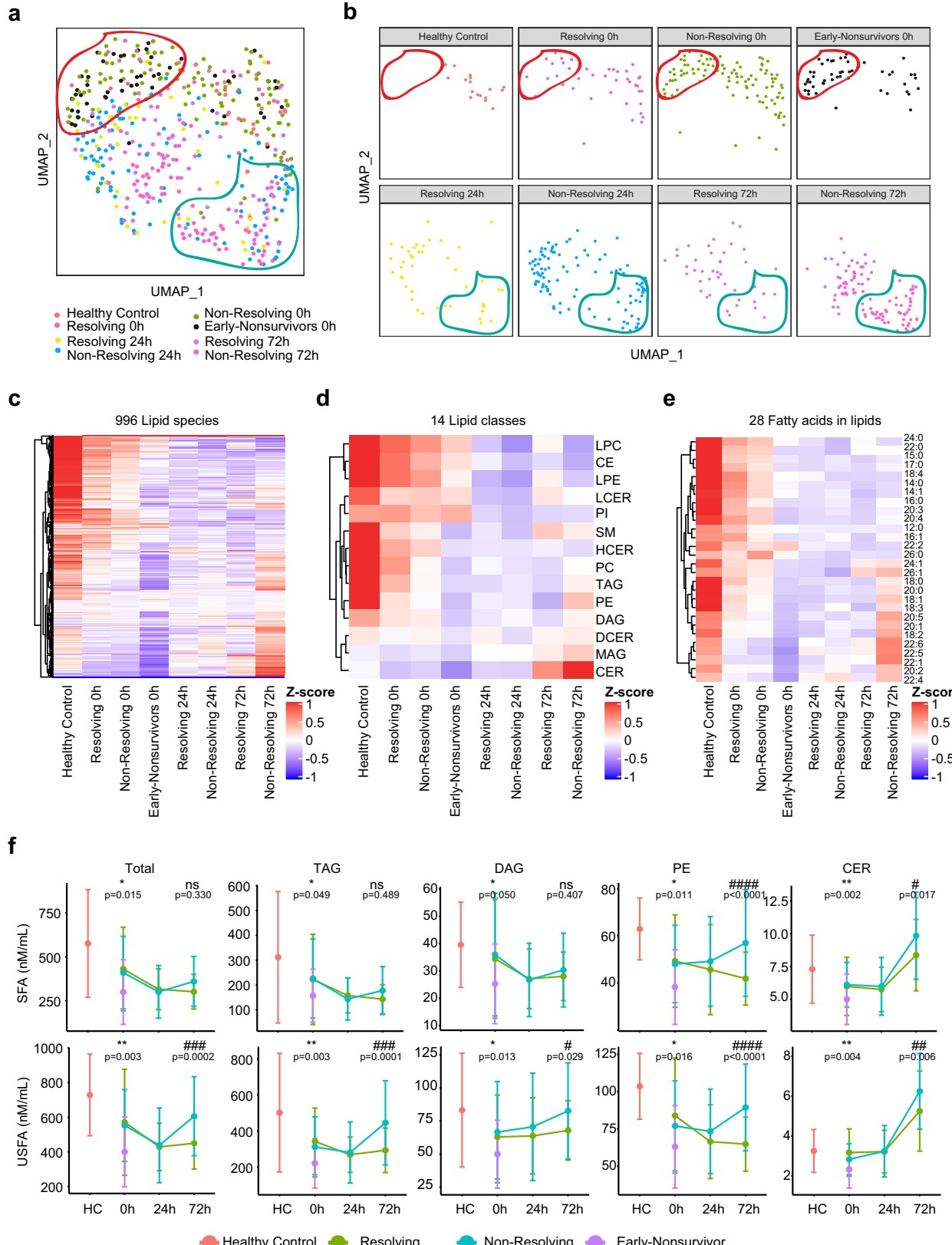

post-injury lipid patterns (Supplementary Fig. 2b). We also observed poor correlation between ISS and total lipids concentrations of either saturated or unsaturated fatty acids (Supplementary Fig. 2c, d, 0 h timepoint shown). Thus, while injury induces major changes in the circulating lipidome, in this cohort of patients with shock on presentation, ISS alone does not associate with lipid patterns.

**Pre-hospital TP enhances lipid levels early after severe injury**

The key observation of the PAMPer trial was the demonstration that initiating TP administration in the pre-hospital setting reduced early mortality when compared to standard care[22]. To assess for an impact of TP, we compared lipid profiles in patients in the treatment arm to those in the standard-of-care arm. UMAP plots demonstrated a

**Fig. 2 | Association between temporal patterns of the circulating lipidome and outcome.** Uniform Manifold Approximation and Projection (UMAP) plot shows the distribution of healthy control subjects ($n = 17$) and trauma patients ($n = 193$), grouped together (**a**) and separated (**b**) by outcome and sampling timepoints. Heatmaps show relative levels of 996 lipid species (**c**); 14 lipid classes (**d**) and 28 fatty acids labeled by carbon number: double bonds (**e**) for healthy subjects and trauma patients, grouped by outcome and sampling timepoints. z-score represents normalized concentrations. Rows are clustered by method of hierarchical clustering. **f** Quantitative comparison of circulating total lipid concentrations among healthy controls (HC) and trauma patients. Lipids are grouped by classes and fatty acids (saturated or unsaturated) identified as the acyl chains in the lipid classes.

Patients are grouped by outcome and sampling timepoints. Center dots and error bars represent median value and median absolute deviation, respectively. SFA saturated fatty acid, USFA unsaturated fatty acid. Asterisks indicate statistical significance based on Kruskal–wallis test among 3 groups at 0 h with post-hoc analysis of Dunn test. The $P$ value was adjusted by Benjamini–Hochberg method: *<0.05; **<0.01. Number sign indicates statistical significance based on 2-way AVOVA test of time-series analysis of resolving and non-resolving groups. Pairwise Comparisons were conducted by Estimated Marginal Means test. The $P$ value was adjusted by Benjamini–Hochberg method: #<0.05; ##<0.01; ###<0.001, ####<0.0001. Source data are provided as a Source Data file.

skewing in the lipid profiles towards the healthy controls in the TP treatment group at 0 h (Fig. 3a, b). However, this preservation of lipid levels associated with pre-hospital TP was seen to dissipate at 24 and 72 h, with no difference in lipid levels or patterns between the TP and standard-of-care groups at these time points. Both the qualitative and quantitative analysis revealed that patients receiving TP had less of a drop in the levels of most classes of circulating lipids at time 0 h, with higher levels of TAG, DAG, and MAG compared to standard-of-care patients (Fig. 3c, Supplementary Fig. 4A). We then assessed the relationship between the predicted mortality, calculated from the Trauma and Injury Severity Score (TRISS), and lipid levels in the two cohorts (Fig. 3d). Average lipid levels were higher in the TP group across all TRISS values. All unexpected deaths (low TRISS Score: predicted mortality rate less than 50%) were in the standard-of-care patients and 11/14 had lipid levels below the mean for the overall cohort. Deaths seen in the TP group were limited to those with a high expectation for death for all except one patient (high TRISS Score: predicted mortality rate of greater than 75%). A Forest plot of log-odds ratios from a multi-variable logistic regression (generalized estimating equation) is shown in Fig. 3e. This analysis revealed that lower lipid levels at 0 h significantly favored mortality within the first 72 h while TP administration favored survival (OR:2.50, CI: 1.24–5.01). Only TRISS had a higher association with early mortality than TP or lipid levels even when traumatic brain injury (TBI) and sex were added to the model.

We next carried out correlation analysis to identify the factors that associate with circulating lipid levels in the early response to severe injury. Included in the analysis were 21 inflammatory and immune mediators, 6 markers of endotheliopathy/ tissue injury, and 2 measures of coagulation abnormalities, all measured at time 0 h. Also included in the analysis were typical measures of injury severity and interventions associated with adverse outcomes. Interestingly, the mediators segregated into three subsets, each with strong internal correlation (Fig. 3f). These included a subset represented by pro-inflammatory cytokines and chemokines that mostly positively correlated with early death, injury severity, endotheliopathy, and abnormal coagulation (Subset 1: IL-6, IL-8, IL-10, MCP-1/CCL2, IP-10/CXCL10, and MIG/CXCL9) and two subsets that correlated inversely with the pro-inflammatory mediators and adverse outcomes including, mediators associated with type 2 and 3 immune responses (Subset 2: IL-2, IL-4, IL-5, IL-7, IL-17A, and GM-CSF) and mediators associated with either tissue protection/ repair or lymphocyte regulation (Subset 3: IL-9, IL-22, IL-25, IL-27, IL-33 and IL-21, IL-23). The relationships between these three mediator subsets remained mostly consistent at 24 and 72 h (Supplementary Fig. 7a, b). However, low lipid levels at time 0 h positively correlated only with standard-of-care, early death, coagulation abnormalities and the endotheliopathy marker, sVEGFR, and not with any of the mediator subsets (Fig. 3f).

We next used probabilistic graphical models for mixed data types[24,25] to infer potential direct (cause-effect) relationships within the multi-modal observational data included in Fig. 3f. These features were loaded into the algorithm and nodes and edges projected onto a graph with early mortality as the endpoint of interest (Fig. 3g). The α-value of 0.2 for the conditional independence tests of the algorithm was selected using nested leave-one-out cross-validation to select the

model with the best predictive performance of patient outcome (see Methods). Circulating lipid concentrations, coagulopathy (including INR), volume of crystalloid used in first 24 h and the pro-inflammatory mediators (via MIG) were identified as direct causal factors contributing to early death (demonstrated by red arrows). The sequential edges connected TP administration to circulating lipid concentrations, coagulopathy, INR, and volume of crystalloid used in first 24 h. These connections indicated a potential mixed causal relationship linking TP with all these factors and fewer early deaths. Other features known to be important to early mortality, including patient and injury characteristics, endothelial and tissue injury, and subset 2 and 3 mediators were indirectly linked to outcomes. Thus, correlation analysis and causal modeling related an interaction between INR and lipid concentration to early death and identified a direct impact of TP on both of these causative factors. These findings further support the notion that a rapid loss of circulating lipids contributes to the early pathogenic state cause by severe injury with shock.

## Confirmation of outcome-based changes in the plasma lipidome in trauma and patients with critical illness due to COVID-19

To determine if our findings could be recapitulated in an independent trauma patient cohort, we conducted an in-depth comparison between the PAMPer dataset and a separate trauma dataset[26] (Trauma dataset-2:TD-2, $n = 86$). Because there were differences in the methods used to quantify lipid species across the datasets, we only carried out indirect comparisons of the relative changes (Z-scores) of lipids species within each dataset across the datasets. A total 75 lipids from 9 sub-classes were found to be in common between the PAMPer and TD-2 datasets (Supplementary Fig. 5a, b). There was remarkable consistency in the relative changes of the early drop and late increase in most lipids over time and based on outcome group. The elevated lipids in the non-resolving patients at 72 h were almost entirely in the PE, MAG and DAG classes in both the PAMPer (23/26) and TD-2 (18/19) datasets. TAG, LPE, LPC, and DCER were not measured in TD-2 and therefore, are not included in this comparison.

To further generalize our findings of outcome-associated changes in circulating lipids to another cause of acute critical illness, we analyzed two public datasets derived from COVID-19 patients[16,17]. Unlike trauma, the onset of critical illness in Covid-19 patients can be highly variable relative to the onset of infection and the time the infection started is often unclear. To assist with the comparison between the trauma and COVID-19 datasets, we set the 0 timepoint in the COVID-19 datasets as the day of symptom onset for non-severe patients and day of progression for severe patients. A total of 29 lipids were identified in common among the 4 datasets (Fig. 4a–d, Supplementary Data 2). Of these, only a subset of PE species were found to be significantly elevated from baseline during critical illness (Supplementary Data 5). We identified eight PE species and one PC specie significantly higher in the non-resolving group (72 h) in PAMPer dataset, while three, six, and five of these PE were elevated in the TD-2, Covid-19 (Guo et al.)[16], and Covid-19 (Shui et al.)[17] datasets, respectively. Eight of these PE species could be identified when combining PAMPer with any single other database, five PE species were in common when combining PAMPer

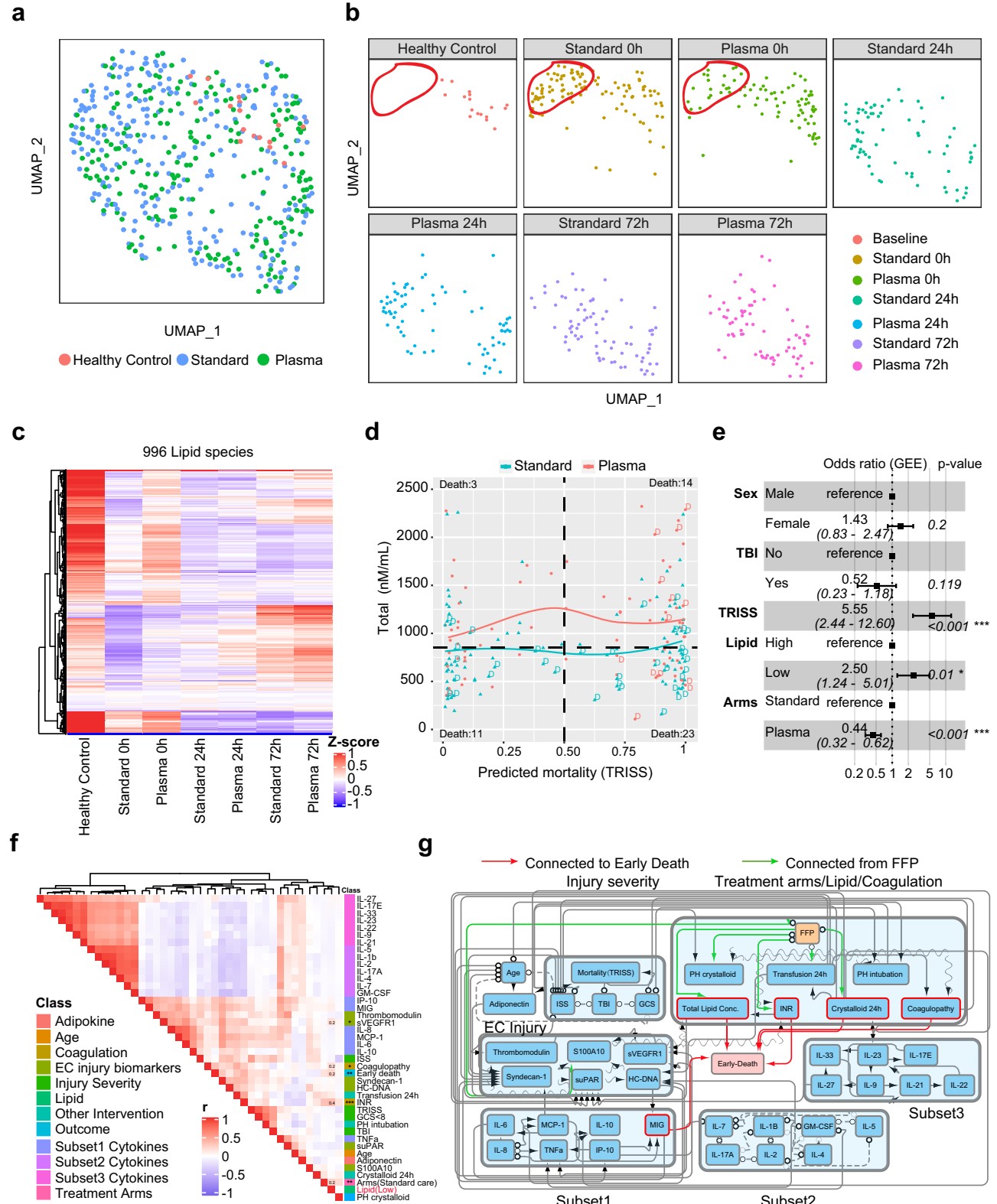

with any two of the other databases, and a single PE was found to be significantly elevated during critical illness in common across all four databases (Supplementary Data 5). Thus, increases in PE consistently associate with critical illness in trauma and COVID-19.

## Generation and evaluation of a lipid reprogramming score

We next sought to determine if a combination of PE species common to the four trauma and COVID-19 patient datasets could be optimized

to generate a Lipid Reprogramming Score (LRS) (Fig. 5a, see also methods for a detailed description). Briefly, the eight PE species detected across four datasets were selected as the starting pool. All the eight PE species were highly correlated with the 37 other lipids (mostly TAG species, Supplementary Data 3) identified as significantly higher in non-resolving PAMPer patients (72 h) by logistic regression taking into account cofounders, including ISS, age, and treatment (Supplementary Fig. 6a, b). A sensitivity analysis identified a model comprised

**Fig. 3 | Potential causal effect for thawed plasma (TP), Lipid concentration and early mortality.** Uniform Manifold Approximation and Projection (UMAP) plot shows the distribution of healthy subjects (HC, $n = 17$) and trauma patients ($n = 193$) (**a**), separated by treatment arms with sampling timepoints (**b**). **c** Heatmap shows relative levels of 996 lipid species for healthy subjects and trauma patients, grouped by treatment arms and sampling timepoints. Exp, z-score normalized concentration. Rows are clustered by hierarchical clustering. **d** Relationship of predicted mortality and total lipid concentration at 0 h upon admission. Trauma patients are grouped by treatment arms; tendency lines are modeled by loess methods for 2 groups separately, dash line in the x-axis means 0.5 and y-axis means the median concentration. **d** indicates patients who died less than 72 h after admission. **e** Forest plot showing odds ratios from logistic regression (generalized estimating equatio) of clinical factors; Lipid concentration; TP effect for early-

nonsurvivors ($n = 51$) versus others ($n = 142$). Error bars: 95% confidence interval. **f** Correlation heatmap showing correlation among cytokines, biomarkers, clinical variables, total lipid concentration and outcome. r: Spearman correlation coefficient. **g** Causal network among factors in **e** constructed by FCI (see also methods) in patients with complete lipid and biomarker data ($n = 170$). The presence of "edges" or connections between nodes in the graph correspond to conditional dependencies relationships. Detailed interpretation of the edges can be found in Methods. Abbreviations: TRISS Trauma and injury severity score, TP thawed plasma, TBI traumatic brain injury, ISS injury severity score, GCS Glasgow coma score, PH Prehospital, INR international normalized ratio. Asterisks in **e** indicate statistical significance in multi-variable logistic regression model: *<0.05; **<0.01. Asterisks in **f** indicate statistical significance for correlation coefficient. *P*-values are approximated by using the t distributions: *<0.05; **<0.01; ***<0.001.

of five PE showing the best performance (Supplementary Data 6). Thus, we defined the LRS as the mean z-score of five PE species (PE (16:0_18:2), PE (16:0_20:4), PE (16:0_22:6), PE (18:0_18:1), PE (18:0_22:6)) representative of PE from all four trauma and COVID-19 datasets. To further technically validate the results, we utilized another platform (LC-HRMS, Platform2) from matched trauma patients ($n = 29$) and healthy controls ($n = 8$) to quantify the concentrations of 5 PE species (Supplementary Data 8). All of them were highly correlated between the two platforms (LC-MS/MS, Platform 1, PF1; LC-HRMS, Platform 2, PF2) (Supplementary Fig. 9a) and selectively up-regulated in non-resolving trauma patients at 72 h (Supplementary Fig. 9b).

We next calculated the LRS for each patient across the three timepoints and plotted these in a UMAP plot (Supplementary Fig. 6c) in order to further reveal their relationships with global lipidome patterns. We found that the gradient in the LRS increased from left-to-right along the x-axis in the UMAP plot, which was consistent with the outcome-based pattern at 72 h. We then transformed the score into a categorical variable with three thresholds based on tertiles (Low, Medium, High) for all PAMPer patients surviving at 72 h (Supplementary Fig. 6c). When displayed on a UMAP plot, the separation of patients into low, medium, and high LRS tertiles distributed the patients similarly to that seen using the continuous LRS. Thus, both the continuous and categorical LRS values represent the magnitude of global changes in the circulating lipidome and may be useful for correlating the lipidomic changes with other patient features. We also explored the relationship between the LRS and either the BMI or early lipid levels in 89 PAMPer patients (Supplementary Fig. 6d, e). There was weak relationship between BMI and total lipid levels ($r = 0.18$, $p = 0.094$). The LRS was independent of BMI ($r = -0.03$, $p = 0.73$). Thus, the LRS could be a representative marker of the changes in circulating lipids in critically trauma patients.

**Risk assessment using LRS for patients with trauma or COVID-19**

We next investigated whether the LRS was associated with outcomes in trauma or COVID-19 patients. A time-series analysis demonstrated that non-resolving trauma patients exhibited dramatic increases in the LRS at 24 to 72 h post-trauma compared to resolving patients (Fig. 5b). Recovery analysis revealed that LRS-high and LRS-medium groups experienced a longer period to recovery than patients in the LRS-low group (Supplementary Fig. 6f). In addition, trauma patients with medium or high LRS were associated with higher injury severity, lower admission blood pressure, mass transfusion, higher INR, and higher incidence of NI and MOF (Supplementary Data 4). High LRS was also associated with lower probability of recovery (HR:0.73, Cl:0.56−0.95) even when adjusted for age, ISS, TBI, and treatment effect in a Cox regression mixed effect model (Fig. 5c). To confirm our finding using a second trauma population, we adopted the same strategy to construct the LRS using the TD-2 dataset, which was dominated by resolving trauma patients. The recovery curve, and Cox regression model all showed similar correlations of LRS with outcomes in TD-2 as seen in PAMPer trial patients (Supplementary Fig. 6h, i). Therefore, the LRS

showed an independent relationship with persistent critical illness after trauma.

We next explored the prognostic value of the LRS and the five individual PE species that comprise the LRS for predicting whether trauma patients would progress to a non-resolving pattern (Supplementary Data 7). Here, we set the standard-of-care arm in PAMPer dataset as the training set ($n = 73$). The TP arm from PAMPer dataset was set as an internal test set ($n = 69$) and the TD-2 dataset was set as an external test set ($n = 86$). Compared to the reference model[27] (ISS + IL6, AUC = 0.798), adding the LRS moderately improved the performance of discrimination (AUC = 0.816, added AUC = 0.018) in the training set (Fig. 5d, Supplementary Fig. 7a). Interestingly, of the five PE that comprise the LRS, PE (18:0_18:1) (RSD:9.82%) also greatly improved the performance of discrimination (AUC = 0.873, added AUC = 0.075) in the training set (Fig. 5d, Supplementary Fig. 7a). We further utilized an established two-step machine learning approach[28–30] to identify a minimal set of predictive lipid biomarkers and clinical features for predicting the outcome in the PAMPer dataset (standard-of-care arm). This approach was based on feature selection (L1 Regularization − LASSO to avoid overfitting) followed by classification (Support Vector Machine) using the down-selected features. The results suggested that PE (18:0_18:1) was the top selected feature among lipids and only IL6 and ISS ranked higher overall (Supplementary Fig. 7b). The performance of calibration (Supplementary Fig. 7c) was also improved by adding either the LRS or the single PE (18:0_18:1) to ISS + IL6 (Brier Score: ISS + IL6, 0.177; ISS + IL6 + LRS, 0.166; ISS + IL6 + PE (18:0_18:1), 0.139). The results were consistent in the internal (Supplementary Fig. 7D, AUC: ISS + IL6, 0.876; ISS + IL6 + LRS, 0.916; ISS + IL6 + PE (18:0_18:1), 0.900) and external test sets (Supplementary Fig. 7E, AUC: ISS + IL6, 0.797; ISS + IL6 + LRS, 0.814; ISS + IL6 + PE (18:0_18:1), 0.841).

We then tested whether we could generalize the LRS for the two COVID-19 patient datasets using a similar approach. The Shui, et al.[17] COVID-19 dataset lacked detailed clinical data, therefore, we only compared differences in LRS among the four outcome groups defined by the authors of the study. We found that mild, moderate and severe COVID-19 patients had a higher LRS compared to healthy subjects (Supplementary Fig. 6g). Consistent with these findings, the LRS was also significantly higher in the severe group when compared to the non-severe COVID-19 patients in the dataset of Guo, et al.[16] (Fig. 5e). We also observed an upward trend in the LRS during the time window preceding progression (<48 h and D6-D14 after progression, Fig. 5e). Finally, multi-variable logistic regression suggested that LRS is an independent risk factor for COVID-19 patients (OR: 9.88, Cl: 2.09−78.5, Fig. 5f). C-reactive protein (CRP) and lymphocyte count (Lym) are known to correlate with worse outcomes in COVID-19 patients[31]. We compared the LRS and its five individual PE species with these two variables to classify severe versus non-severe patients in both a training set (C1, $n = 45$) and test set (C2, $n = 10$) (Supplementary Data 7). We found that the LRS alone moderately improved the performance of discrimination (AUC = 0.814, added AUC = 0.028), however a single PE specie from the LRS (PE (16:0_22:6), RSD:5.51%) alone greatly improved

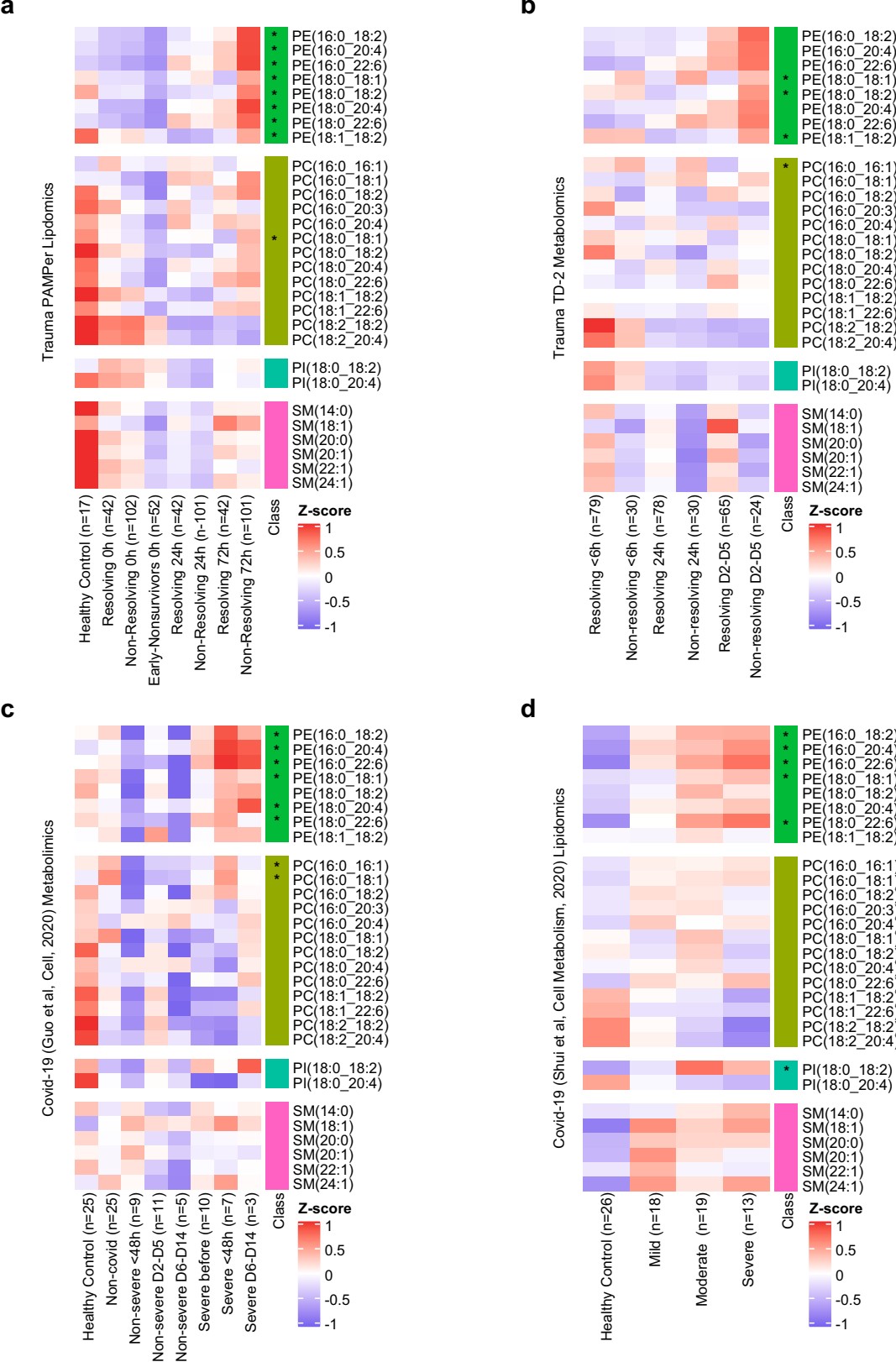

**Fig. 4 | Comparison of temporal patterns of common lipids for patients with trauma or COVID-19. a**, **d** Heatmaps show the relative levels of 29 common lipid species from four major classes across patients. Data comes from trauma patients from the PAMPer lipidomics dataset (**a**) and TD-2 untargeted metabolomics dataset (**b**); COVID-19 patients from untargeted metabolomics dataset (Guo et al. Cell, 2020) (**c**) and lipidomics dataset (Shui et al., Cell metabolism, 2020) (**d**). Patients are grouped by outcome and sampling timepoint (except for **d**). Asterisks indicate lipids with statistical significance (*p* value <0.05) and log2 fold change >0.4 by two-sided Wilcoxon Rank Sum test between non-resolving and resolving trauma patients at 72 h (**a**); non-resolving and resolving trauma patients at D2-D5 (**b**); severe and non-severe Covid-19 patients (**c**); severe and mild Covid-19 patients (**d**). Abbreviations: PE phosphatidylethanolamines, PC phosphatidylcholines, PI phosphatidylinositols, SM sphingomyelins.

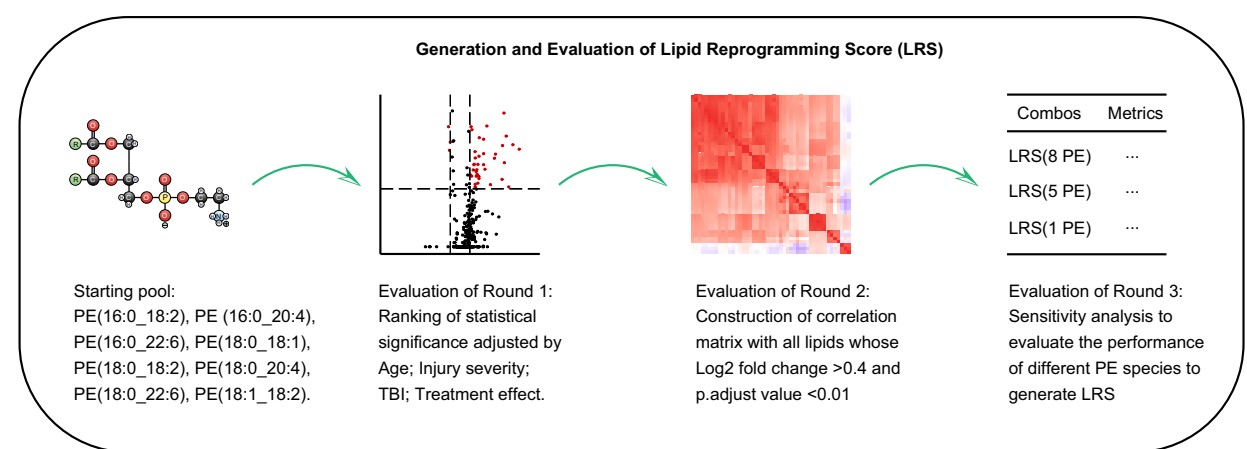

the performance (AUC = 0.862, added AUC = 0.076) (Fig. 5g, Supplementary Fig. 7a). The two-step machine learning approach also revealed that the PE (16:0_22:6) were the top selected features (Supplementary Fig. 7C). The performance of calibration was also improved by using PE (16_22:6) (Brier Score: Lym+CRP:0.177; LRS: 0.166; PE (16:0_22:6): 0.139; Supplementary Fig. 7G). The performance of PE

(16:0_22:6) also had the highest AUC compared to other two models in the test set (Supplementary Fig. 7f, AUC: CRP + Lym, 0.917; LRS, 0.833, PE (16:0_22:6), 0.958). Interestingly, we also noticed that PE (16:0_22:6) showed similar performance for prognostication as the random forest model based on 17 proteins and 9 metabolites reported in the original manuscript. It is notable that only one patient (XG43) was mislabeled in

**Fig. 5 | Lipid Reprogramming Score (LRS) is an independent risk factor for outcome after trauma or COVID-19. a** Graphical scheme of generation and evaluation of LRS. **b** Comparison of LRS from patients with trauma ($n = 142$). Patients are grouped by outcome and sampling timepoint. Center dots and error bars represent median value and median absolute deviation, respectively. **c** Forest plot showing hazard ratio of clinical factors and LRS score for recovery using a Cox regression mixed effect model in patients surviving at 72 h ($n = 142$). Error bars: 95% confidence interval. **d** ROC curve for three prognostic models in training cohort from Standard-of-care arm in the PAMPer dataset (trauma patients, $n = 73$). **e** Comparison of LRS for patients with COVID-19. Healthy Subjects ($n = 25$), Non-COVID ($n = 25$) and COVID-19 patients ($n = 45$) are grouped with diseases outcome and sampling timepoint. Center dots and error bars represent median value and median absolute deviation, respectively. **f** Forest plot showing odds ratio of clinical factors from logistic regression and LRS score for Non-severe ($n = 25$) versus Severe

COVID-19 patients ($n = 20$). Error bars: 95% confidence interval. **g** Comparison of prognostic value of LRS, PE (16:0_22:6), lymphocyte count, and CRP for Non-severe ($n = 25$) versus Severe ($n = 20$) outcome for the COVID-19 cohort (C1) from Guo. et al by ROC curve. ISS injury severity score, Lym lymphocyte count, CRP C-reaction protein. Asterisks in **b** indicate statistical significance in based on 2-way AVOVA test of time-series analysis of resolving and non-resolving groups. Pairwise Comparisons was conducted by Estimated Marginal Means test. The $P$ value was adjusted by Benjamini−Hochberg method: ****<0.0001. Asterisks in **e** indicate statistical significance based on Kruskal−wallis test among 6 groups of COVID-19 patients with post-hoc analysis of Dunn test. The $P$ value was adjusted by Benjamini−Hochberg method: *<0.05. Asterisks in **d** and **g** indicate statistical significance in multivariable regression model: *<0.05; **<0.01. Source data are provided as a Source Data file.

the test cohort (C1) using PE (16:0_22:6) for prognostication (Supplementary Fig. 7g). Thus, similar to our observations in trauma, a single PE specie derived from the LRS performed well of prognostication for severe disease in COVID-19.

## Association between LRS and systemic markers of inflammation and endothelial dysfunction in trauma patients

We next sought to determine if the LRS correlated with circulating markers of inflammation or endothelial and tissue damage. A correlation matrix was constructed using data from the 121 PAMPer patients alive at 72 h that had complete data for lipids, 21 cytokines and chemokines, endotheliopathy markers, and tissue injury markers across time after injury (Time 0 h: Fig. 6a, Times 24 and 72 h: Supplementary Fig. 8a, b). Across the three time points, LRS correlated positively with pro-inflammatory cytokines/chemokines (defined above as subset 1), as well as endotheliopathy and tissue injury biomarkers. Conversely, LRS correlated negatively with subset 2 (lymphocyte-related) and subset 3 (protective/ reparative) cytokines and an adipokine (Adiponectin). These findings suggest that the changes in the circulating lipidome at 72 h, represented by an elevated LRS, associates with biological process that drive worse outcomes (e.g., inflammation, endotheliopathy, and tissue injury), and therefore, may contribute to or be part of the pro-inflammatory/ tissue injury processes that are known to contribute to adverse outcomes in trauma.

## Discussion

The main goal of this study was to correlate the temporal patterns in the circulating lipidome with outcomes in the early evolution of critical illness in humans. Using trauma as a model, we found that three distinct clinical trajectories each align with comprehensive changes in the patterns of circulating lipids. These relationships are depicted in a summary diagram in Fig. 6b. The findings include: (1) A dramatic drop in all classes of lipids in the hyperacute phase after of severe injury that was most extreme in patients destined to die. Early TP mitigated this rapid drop in lipid levels and was associated with improved outcomes; (2) the drop in circulating lipids persisted through 72 h in patients destined to resolve their critical illness earlier; (3) a delayed rise in circulating in DAG, TAG, and PE species in patients that went on to experience persistent critical illness. Remarkably, the over-representation of PE species in trauma patients with critical illness was easily identified in critically ill patients in a validation trauma dataset and two COVID-19 datasets. A Lipid Reprogramming Score derived from PE was an independent risk factor for worse outcome and correlated with excessive proinflammatory responses. Although there have been multiple metabolomics studies characterizing the circulating metabolome in critical illness[12,16,17,32,33], to date there are no reports focusing on the comprehensive temporal lipidome changes in this disease context. We show that lipids may be sensitive markers of the host response to systemic stress and serve as prognostic biomarkers of critical illness.

Among the most pronounced changes observed in our study was the early loss of all classes of lipids in the circulation after injury. A study of 32 trauma patients showed that blood triglyceride levels were significantly lower in 9 non-survivors within 28 min of injury, suggesting that injury-induced decreases in circulating lipids may begin very early after a severe trauma[34]. Our healthy controls were non-fasting and sampled throughout the day to align with the presentation of the typical trauma patient. Therefore, the differences between controls and injured at time 0 h are unlikely to be due to dietary effects. While the degree of the decline in lipids associated with clinical outcomes, the incidence was not dependent on injury severity. A stress hormone-induced hypermetabolic state with associated increased catabolism is seen after trauma and other causes of critical illness[6,35] and may explain the persistent decline in circulating lipids. The catabolism response generates energy substrates from carbohydrates, fats, and protein in an "all or none" manner that, like our findings, is not influenced by injury severity[36]. It is reasonable to speculate that the abrupt loss of lipids may be due, in part, to the uptake and catabolism of lipids to meet the energy demands. The finding that patients that die within first 72 h experience the greatest magnitude in lipid loss from the circulation raises the interesting possibility that a circulating energy substrate crisis contributes to the early mortality.

Administration of TP in route to the trauma center improves early survival and we show here that this also results in higher levels of circulating lipids. This was especially true for glycerolipids, including TAG, DAG, and MAG, which are rich energy substrates. In addition to providing a source of lipids, TP also contains proteins involved in coagulation, and many other factors likely to contribute to its salutary actions. TP is well known to reduce bleeding complications and we have recently reported an association of TP administration with a prevention of endothelial dysfunction and an excessive inflammatory response[22,37]. The correlative changes in early lipid levels and outcomes in our study point to lipids as another potential beneficial component of TP. In the future, circulating lipid profiles may be useful for prognostication or for guiding early interventions with plasma or other strategies to replace specific lipid deficiencies in trauma or in critical illness from other etiologies.

In stark contrast to the early changes in circulating lipids, a subset of lipids (predominantly TAG, DAG, and PE) began to rise in the circulation between 24 and 72 h in patients that subsequently exhibited a slow recovery or die. In addition to lipolysis and hypermetabolism, patients with critical illness experience pathologic alterations in liver such as hepatic steatosis[38–42]. Studies in severe burn trauma associate the browning of white adipose tissue with enhanced lipogenesis in liver[43,44]. Interestingly, the inter-class correlation network among the lipids we identified at 72 h is similar to the lipogenesis pathway in the liver. This suggests that the liver is one of the sources of the glycerolipids and PE that appear in the circulation and that these reflect ongoing systemic inflammation and metabolic stress. That DAG, TAG, and PE are linked though a common synthesis pathway further

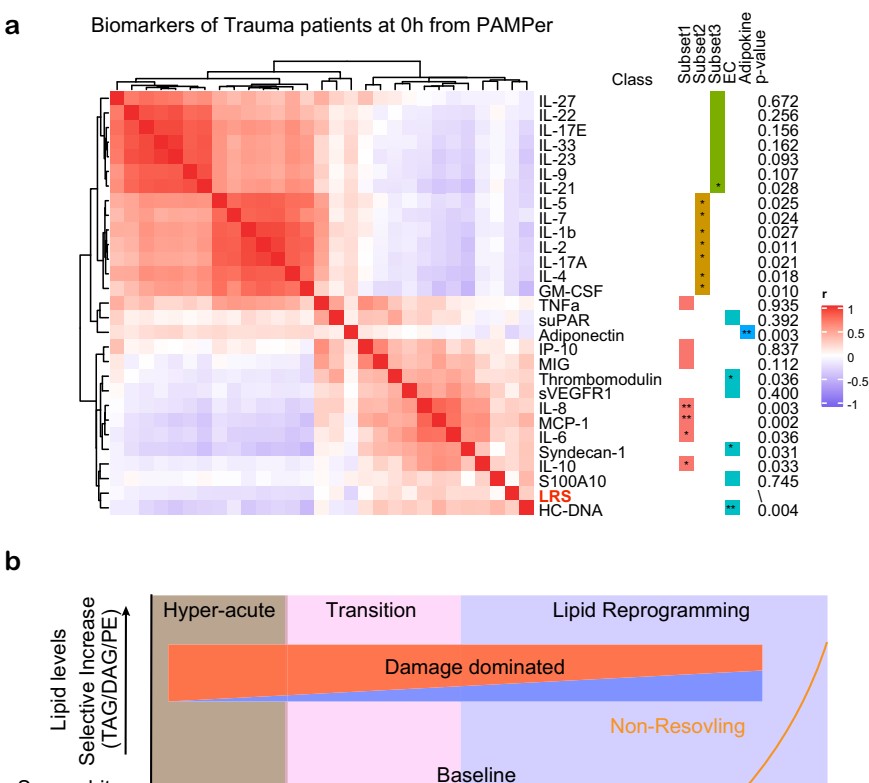

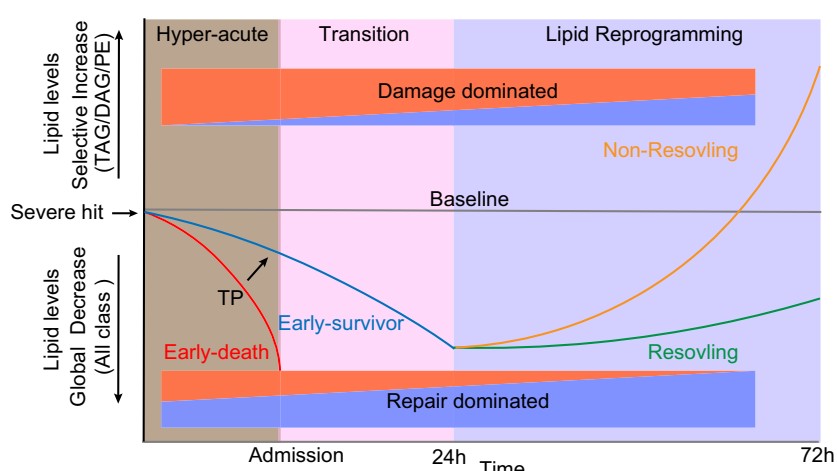

**Fig. 6 | Association between LRS and circulating biomarkers. a** Heatmap showing correlation of LRS and circulating biomarkers at 0 h in trauma patients (*n* = 121), measured by Spearman correlation coefficients. Asterisks in **a** and **b** indicate statistical significance for correlation coefficient. Unadjusted *p*-values are approximated by using the two-sided *t* distributions: *<0.05; **<0.01; ***<0.001.

**b** Schematic of proposed paradigm showing the relationship between circulating lipid levels and outcomes after severe injury. Early loss of circulating lipids correlates with adverse outcomes while failure to resolve critical illness is associated with the selective increase in glycerolipids and PE.

supports this possibility[45]. The majority of circulating lipids are complexed with lipoproteins. The various classes of lipoproteins complexes vary in lipid composition; therefore, it is likely that the classes of lipoprotein complexes also vary over time after severe injury[46]. Several specific lipid species [e.g. PC(16:0_18:1), PC(18:0_18:1)] contribute to inter-organ (liver, muscle and adipose tissue) communication[47]. We observed that PC (16:0_18:1) and PC (18:0_18:1) were higher at 72 h in the non-resolving trauma patients or severe Covid-19 patients, raising the possibility for a lipid reprogramming process across organs during persistent critical illness.

The LRS that we generated from five PE species had prognostic value when measured at 72 h in trauma patients or at the onset of symptoms COVID-19 patients. Higher PE levels as represented by the LRS and even a single PE derived from the LRS could prognosticate for severe critical illness with combined with ISS and IL-6 in trauma or when used alone in COVID-19. The specificity of single PE species for prognostication for trauma and COVID-19 might represent characteristics specific to the patients, the causes of the critical illness, or the differences in the methods used to measure the lipids in the studies. Prospective studies will be needed to confirm the prognostic value of measuring subsets of PE in trauma and other forms of critical illness。

however these findings raise the possibility that increases in PE synthesis and other lipids might be a feature common to persistent critical illness resulting from diverse etiologies. Noticeably, only TAG and DAG comprised of unsaturated fatty acids increased in non-resolving patients. These fatty acids include Eicosapentaenoic Acid (EPA) and Docosahexaenoic Acid (DHA), which are precursors for lipid mediators involved in inflammation resolution and tissue repair[11,48,49]. Thus, in addition to providing a source of lipids for systemic energy needs through the release of acyl glycerides, this response might reflect the host's attempt to resolve the ongoing inflammatory response and tissue injury. Thus, the LRS or induvial PE species (PE(18:0_18:1),PE(16:0_22:6)), like other biomarkers of the host response (e.g., IL6 and CRP), may be useful as early parameters for outcome prediction linked to specific biologic processes. It will also be of interest to determine if the LRS or PE species predicts outcomes in other etiologies of critical illness such as sepsis and burns.

Global lipid metabolism is regulated by many factors such as pro-inflammatory mediators, adrenergic stress, and regulatory hormones[11,39,43,50,51]. Propranolol or IL-6 receptor blockade can attenuate the browning of white adipose tissue and hepatic steatosis in experimental burn trauma[43]. Interestingly, we also found that the LRS

is positively associated with the pro-inflammatory response, the acute phase response, endothelial injury, and coagulation but inversely correlated with mediators shown to contribute to tissue protection and repair. This relationship persisted throughout the 72 h observation period. The adiponectin is produced by liver and adipose tissue, respectively, and are functionally associated[52]. This hormone enhances fatty acid oxidation as an energy source and were negatively correlated with the LRS, consistent with a dysregulated lipid reprogramming in patients with persistent critical illness.

Our study has several limitations. Clinical lipid panels are not routinely measured in severely injured trauma patients. Therefore, we could not correlate changes in these commonly assessed lipids with our lipid panels. Unfortunately, we also could not determine the levels of lipids mediators since these were not measured in our targeted lipid assays. Further studies to validate the prognostic value of the LRS or individual PE are warranted. The mechanistic relationship between the changes in lipids in the circulation do not necessarily reflect lipid metabolism within specific organs or tissues. Finally, the functional contributions of the observed lipid changes to patient outcomes remain to be established in patients.

In conclusion, our findings provide a paradigm for the lipid response to a severe and acute systemic stress leading to critical illness (summarized in Fig. 6c). Our causal modeling and correlation analyses place lipolysis a central regulator of the evolution from acute disease onset to critical illness in humans. The features of lipogenesis we identified appear to be common to critical illness due to multiple etiologies and potentially useful for predictive modeling and target identification. Both the proposed paradigm and our comprehensive datasets will be useful for further study of altered lipid metabolism in acute critical illness.

## Methods
### Study population and samples
The PAMPer trial was approved by the IRB of University of Pittsburgh as previously described[19]. The research reported in this paper is covered by the original IRB approval. Critically injured participants were frequently either in a semi-conscious or unconscious state when they were enrolled at the scene. These patients were too ill to consent to immediate treatment. Thus, for treatments that must be given immediately to be effective, exception from informed consent research is considered appropriate by federal regulations[53] (https://www.fda.gov/regulatory-information/search-fda-guidance-documents/exception-informed-consent-requirements-emergency-research). The Emergency Exception from Informed Consent (EFIC) protocol from the Human Research Protection Office of the US Army Medical Research and Material Command was applied to this study. Further details of emergency exception to informed consent can be found at the official website for the PAMPer trial at https://crisma.upmc.com/apps/PAMPer/home/. Registration information and the detailed study protocol are available on https://clinicaltrials.gov/ct2/show/NCT01818427. All participants or their legally authorized representatives provided written consent to continue participation following admission to the hospital. Participants without this written consent were excluded for analysis. No participant compensation was provided. The primary outcome of PAMPer trial was 30 day survival and this has been previously reported[22]. The administration of thawed plasma resulted in lower 30-day mortality than standard-care resuscitation group (23.2% vs. 33.0%; difference, −9.8 percentage points; 95% confidence interval, −18.6 to −1.0%; $P = 0.03$).

Healthy volunteers were enrolled in an observational study approved by the University of Pittsburgh Institutional Review Board (PRO08010232). The detailed study protocol is available on https://www.clinicaltrials.gov/ct2/show/NCT00250523. Written informed consent was obtained from all the subjects. No participant compensation was involved in the study.

We conducted longitudinal sampling of plasma (0 h; 24 h; 72 h after admission) from 193 patients with trauma prospectively enrolled in the PAMPer trial[22], along with 17 healthy subjects. The detailed workflow is shown in Supplementary Fig. 1. The primary aim of PAMPer trial was to test if administering prehospital thawed allogeneic plasma (TP) during air medical transport can reduce in-hospital mortality for severely injured trauma patients. Values for clinical and physiological variables with biomarkers of injury and inflammation given in the manuscript were reported from previous studies[22,37]. The outcome of trauma patients was defined as: Resolving (Survival with ICU stay <7 days); Non-resolving (Survival with ICU stay > = 7 days or non-survival with death day >3 days) and Early-nonsurvivors (Non-survival with death day < = 3 days). Blood samples were collected using vacuum isolation tubes with anticoagulant of Heparin sodium, which were centrifuged at 4°C and plasma fractions were stored at −80°C for further analysis.

### Plasma lipidomic profiling
Lipidomic profiling was performed through the Complex Lipid Panel™ technique at Metabolon (Metabolon Inc, Morrisville, NC 27560, USA). Briefly; lipids were extracted from the plasma using automated BUME extraction[54]. Samples were analyzed using differential mobility spectrometry (DMS) interface (SCIEX) and a high flow LC-30AD solvent delivery unit (Shimazdu). Each sample was run once on the platform using a method that combines DMS 'on' and 'off' as well as positive and negative ionization modes. The following lipid classes were quantified with i) DMS 'on' and in negative ionization mode: PC, PE, LPC, LPE, ii) DMS 'on' and in positive ionization mode: SM, iii) DMS 'off' and in negative ionization mode: FFA, iv) DMS 'off' and in positive ionization mode: TAG, DAG, CE, CER. The internal standards were selected based on the combination of carbon length and the number of double bonds. Metabolon maintains assay-specific internal standards based on superiority compared to single standards. The panel has an expanded set of internal standards, containing over 50 deuterium-labeled lipid molecular species across 14 lipid classes that mimic the biochemistry found in human plasma. These standards were developed by SCIEX, in collaboration with Avanti Polar Lipids and Metabolon Inc (https://sciex.com/products/consumables/lipidyzer-platform-kits). Full list of internal standards can be found in Supplementary Data 10. Further details can be found in the patent literature (https://patents.google.com/patent/US11181535B2/en,Table 1-8). Lipid species concentrations were background-subtracted using the concentrations detected in process blanks (water extracts) and run day normalized. Background levels were estimated/calculated from the median levels of the three process blanks (water) if there were detectable levels in at least 2 of the 3 blanks in each batch. The background level was subtracted from each sample in the batch prior to any run day normalization. The QC sample was generated by combining a small aliquot from the entire set of samples into a single pooled CMTX (ClientSample matrix). Four aliquots of the CMTX were run on each plate of 36 samples. One each was injected at the beginning and end of the run, with the other two roughly evenly spaced between the remaining samples. The internal standard was run multiple times throughout the experiment. Instrument variability was evaluated by calculating median relative SD (RSD) from the quality control sample matrix. The median RSD values for 14 lipid classes can be found in Supplementary Data 8.

### High-resolution LC-HRMS (Platform 2, PF2) Lipidomic Analysis of 5 PE species
In order to exclude the effect of injury characteristics, we conducted propensity score matching of age, ISS and TBI between resolving (n = 14, one sample was not available) and non-resolving male patients (n = 15) in PAMPer trial (Supplementary Data 9). We also included samples from 8 non-fasting healthy subjects as a control group for comparison. Plasma samples from trauma patients of 72 h and healthy

subjects were analyzed in the metabolic core at university of Pittsburgh. Metabolic quenching, lysis, and lipid extraction was performed via Folch extraction. Briefly, 400 μL of water, 500 μL methanol and 1 mL chloroform was added to 100 μL plasma and spiked with 5 μL PE-UltimateSPLASH deuterated internal standard mix (Avanti Polar Lipids – 330826 Birmingham, AL). Samples were vortexed for 2 min and rested on ice for 10 min before phase separation via centrifugation at 3000 × $g$ for 25 min at 4 °C. 800 μL of organic phase was dried to completed under nitrogen gas and resuspended in 1:1 acetonitrile:isopropanol. 2 μL of sample was subjected to online LC-MS analysis. Calibration curves were prepared using purified PE species with side chain lengths: 16:0–18:2, 16:0–22:6, 16:0–20:4, 18:0–18:1, and 18:0–22:6 by serial dilution from 15 μM down to 0.117 μM for absolute quantification.

Analyses were performed by untargeted LC-HRMS. Briefly, Samples were injected via a Thermo Vanquish UHPLC and separated over a reversed phase Thermo Accucore C-18 column (2.1 × 100 mm, 5 μm particle size) maintained at 55 °C. For the 30 min LC gradient, the mobile phase consisted of the following: solvent A (50:50 H2O:ACN 10 mM ammonium acetate/0.1% acetic acid) and solvent B (90:10 IPA:ACN 10 mM ammonium acetate/0.1% acetic acid). Initial loading condition is 30% B. The gradient was the following: Over 2 min, increase to 43%B, continue increasing to 55% B over 0.1 min, continue increasing to 65%B over 10 min, continue increasing to 85%B over 6 min, and finally increasing to 100% over 2 min. Hold at 100% for 5 min, followed by equilibration at 30%B for 5 min. The Thermo IDX tribrid mass spectrometer was operated in positive ESI mode. A data-dependent MS[2] method scanning in Full MS mode from 200 to 1500 m/z at 120,000 resolution with an AGC target of 5e4 for triggering ms[2] fragmentation using stepped HCD collision energies at 2040 and 60% in the orbitrap at 15,000 resolution. Source ionization settings were set to 3.5 kV for spray voltage in positive mode. Source gas parameters were 35 sheath gas, 5 auxiliary gas at 300 °C, and 1 sweep gas. Calibration was performed prior to analysis using the Pierce™ FlexMix Ion Calibration Solutions (Thermo Fisher Scientific). Standard peak areas were then extracted manually using Quan Browser (Thermo Fisher Xcalibur ver. 2.7), normalized to deuterated internal standard peak area and converted to concentrations using the calibration curves. The calibration curves of 5 PE species are shown in Supplementary Fig. S10.

## Lipidomic data pre-process and dimension reduction
Lipids were named according to its sub-class and fatty acid composition; (e.g., PE (16:0_18:2) means this lipid belongs to phosphatidylethanolamine (PE) class and it was synthesized from palmitic acid (C16:0) and linoleic acid (C18:2)). The nomenclature we used simply lists the 2 sidechains present without attempting to ascribe which resides at which position. Lipids with over 20% missing quantitative values were discarded due to the concern of low quality. Other missing values for each lipid species were imputed with the minimum concentration. Lipid class concentrations were calculated from the sum of all molecular species within a class, and fatty acid compositions were determined by calculating the proportion of each class comprised by individual fatty acids.

Normality of each lipid species distribution was tested by Shapiro–Wilk test and Q-Q plot. No transformation was conducted because most lipid species obey normal distribution or was near normal distribution. A two steps approach of dimension reduction from both linear and non-linear methods were applied. Principle Component Analysis (PCA) was performed on z-score scaled concentration of each lipid species. Then, Uniform Manifold Approximation and Projection (UMAP) was conducted by using the first 20 PCs. All subjects grouped by outcome or timepoint were visualized in UMAP plot. No obvious outliers were identified in the UMAP plot.

## Causal inference analysis
Causal inference was performed by using the on-line CausalMGM[55] and the command-line tool for FCI[56]. Early death (death day ≤ 3 after admission) was set as the outcome and all other variables which may be related to early death were kept as input (Clinical information: Age; Trauma brain injury (TBI), Injury severity (ISS); GCS; TRISS, Hemostasis: INR; Coagulopathy. Intervention: Prehospital thawed allogeneic plasma (TP); Prehospital transfusion volume of crystalloid; Prehospital intubation; Transfusion volume in first 24 h after admission, Biomarkers: 21 cytokines with 7 endothelial injury related markers, total lipid concentration). Continuous variables of biomarkers were log2 transformed and z-score scaled to meet the assumption of normality. Categorical variables were tested to meet the assumption of multi nominal distribution. To select the optimal α-value threshold for the conditional independence tests of the FCI we used a nested leave-one-out cross validation. In each round, directed graphs were learned from all but one samples at different α-values (α = {0.01, 0.05, 0.1, 0.15, 0.2, 0.25}). The variables in the Markov blanket of the "Early death" variable (i.e., parents, children and spouses) in each α-value were used to train a logistic regression model. This model was then used to predict the "Early death" in the left-out sample. The procedure was repeated for all samples and Receiver Operator Characteristic (ROC) curves were constructed for each α-value. The value of α = 0.2 produced models with the best Area Under the ROC Curve (AUC = 0.80). The final causal network presented in Fig. 3g was constructed on the full dataset using the α = 0.2 for the conditional independence tests. A directed edge A ->B indicates that A is a cause of B (i.e., a change in A is expected to affect a change in B). A bidirected edge A <-> B indicates that there is unmeasured confounder affecting both A and B. A partially directed edge A o-> B indicates that B is not a cause of A, but it is unclear whether A is a cause of B or if there is a latent confounder that causes both A and B. An undirected edge A o-o B indicates that we cannot make inferences about the causal orientation of that edge.

## Correlation network and lipid biosynthesis pathway
Correlation networks were constructed using 412 lipids based on a Pearson correlation coefficient matrix from all samples. All lipids in the class of MAG; CE; PI; LPE; LPC; SM; CER; LCER; HCER; DCER were kept. Lipids of TAG; DAG; PE; PC were kept at top 100; 30; 40;40 variable species respectively to reduce the complexity of network. Variance Stabilizing Transformation (vst) method was used for identifying variable lipids and mean-var plot for each class was examined to ensure the stability. The threshold of the correlation coefficient was tuned from 0.5 to 0.8 and then set at 0.7 based on the following considerations: 1. Balance between intra-class correlation and inter-class correlation; 2. Preference for a higher threshold to reduce false positive relationships. Cytoscape (version 3.8.0) was used to construct the inter-class and intra-class network and layout was set as circular[57]. Lipid biosynthesis pathways were summarized from previous published literature.

## Establishment and application of the lipid reprogramming score (LRS)
We generated the Lipid Reprogramming Score (LRS) based on the consideration of following points: 1) The lipids species used to generate the score could be easily detected across the metabolomic\lipidomic platforms; 2) The lipids species reflected the lipidomic patterns associated with outcomes; 3) The LRS associates with outcomes in both trauma and Covid-19 after adjusting for confounding variables. For point 1), we used the 8 PE species as the starting pool since they were detected in both the targeted lipidomic platforms (PAMPer\Covid-19 dataset of Shui et al.) and untargeted metabolomic platforms (TD-2\ Covid-19 dataset of Guo et al.). Other species from PC\PI\SM were not included since on our correlation network analysis (Supplementary Fig. 3a) revealed that only TAG\DAG\PE were highly inter-

correlated. Only z-scores were used since the untargeted platforms did not report the quantitive concentration. For point 2), we explored the relationship of these 8 PE species with the top differentially lipids in PAMPer dataset. Other datasets were excluded since they were not from untargeted assays or without clear timepoints. First, we systematically identified the lipid species that were different between non-resolving and resolving trauma patients using logistic regression with age, ISS, and treatment as co-variables (Supplementary Data 3). This yielded 37 lipids (27 TAG and 9 PE) that were significantly (adjusted $p < 0.01$, log foldchange >0.4) higher at 72 h in the non-resolving PAMPer patients and three LPC that were lower (Volcano plot shown in Supplementary Fig. 6a). We noticed there was one specie PE (18:0_18:1) identified by both approaches. Next, we constructed a matrix that correlated these eight 8 common PE species with the 37 lipids that were differentially expressed in the non-resolving PAMPer patients (Supplementary Fig. 6b). All eight PE were correlated positively with the other PE and 27 TAG, and negatively correlated with the three lower LPC species. For point **3**), only lipid species that preserved the outcome relationships in both trauma and Covid-19 datasets were considered candidates for inclusion to generate the LRS. We then applied different criteria to further reveal the relationship of lipids species between the four datasets. In addition to the eight PE species identified when combining PAMPer with any single other database, five PE species were in common when combining PAMPer with any two of the other databases, and a single PE was found to be in common across all 4 databases (Supplementary Data 5). In order to optimize the combinations of PE species to generate LRS, we conducted a sensitivity analysis by testing the association between outcome and the mean z-score from the eight, five or one PE species (Supplementary Data 5) from the combination of PAMPer with the other trauma and COVID-19 patient datasets. The mean z-scores from eight, five and 1 PE species performed in a comparable manner (Supplementary Data 6) in the trauma dataset (HR in PAMPer: 0.69\0.73\0.64, HR in TD-2: 0.73\0.77\0.67). However, the performance of the five species model was better (Supplementary Data 6) compared to other two models in the COVID-19 patient datasets (OR in dataset of Guo et al.: 4.67\9.98\1.81). Thus, we defined the Lipid Reprogramming Score (LRS) as the mean z-score of five PE species (PE(16:0_18:2),PE(16:0_20:4), PE(16:0_22:6), PE(18:0_18:1), PE(18:0_22:6)). These PE species were representative of all four Trauma/COVID-19 datasets. Trauma patients in the PAMPer trial who survived at 72 h after admission were classified to 3 groups (High, Medium, Low) according to the tertiles of LRS across all patients. LRS was calculated for both trauma and COVID-19 patients as well as healthy subjects when applied in time-series or comparison analysis. LRS was only calculated for trauma or COVID-19 patients when applied in multi-variable model of cox regression or logistic regression.

### Recovery analysis

A Kaplan–Meier Curve was used in the recovery analysis for trauma patients from PAMPer or the TD-2 dataset. ICU length of stay was used to estimate the time to recovery for patients due to lack of detailed variables for dynamically monitoring organ dysfunction since injury. Patients who experience early death were excluded for recovery analysis. The ICU length of stay for patients that died over 3 days after admission was consider as maximum days in this dataset, because they cannot recover from injury. Patients who experience ICU length of stay over 30 days were consider as censored at day 30.

### Multi-variable regression analysis

Multi-variable model of logistic regression and generalized estimating equation (to account for cluster effect) was used for testing the categorical outcomes, such as, survival or severity. For trauma patients, two outcomes Non-resolving (Survival with ICU stay ≥ 7 days or non-survival with death day >3 days) and Early-nonsurvivors (Non-survival with death day ≤ 3 days) were used. For COVID-19 patients, the severity

was defined consistent with the scale of the WHO Clinical Score[58]. Demographic information (e.g. age, sex), TBI, TRISS, treatment arm and total lipid concentration at 0 h upon admission were included in the generalized estimating equation model for early death in PAMPer dataset. Demographic information (e.g. age, sex), Lymphocyte count, CRP, LRS for each patient were included in the logistic regression for modeling severe COVID-19 patients in dataset of Guo et al.[16]. A multi-variable model of Cox regression was used for testing the time to discharged by ICU for trauma patients. Demographic information (e.g. age, sex), TBI, ISS, treatment arm and LRS score among patients at 72 h after admission were included in the Cox regression with mixed effect (to account for cluster effect) for modeling non-resolving patients in the PAMPer dataset. Association test by using same variables except for treatment arm was conducted in TD-2 dataset. For prognostic model generation, we built logistic regression models for predicting patients who developed non-resolving pattern in the trauma cohorts (Training set: standard-of-care arm in PAMPer dataset, Test set: plasma arm in PAMPer dataset or TD-2 dataset) or who progressed into severe COVID-19 in the cohort of Guo et al (Train:C1, Test:C2). All models were internally evaluated by 10-fold cross-validation. The performance of discrimination was assessed by ROC curves, and AUC values. Performance of calibration was assessed by calibration curves. Brier scores were used to evaluate the overall fit of the model.

### A two-step machine learning approach

The importance of clinical features and lipids commonly detected in trauma and Covid-19 datasets were evaluated by two-step feature selection strategy as previously reported[28–30]. Briefly, two models (least absolute shrinkage and selection operator: lasso; support vector machine with the radial basis function: SVM-RBF) were fitted sequentially with 100 times repeated and nested 5-fold cross-validation (Both outer and inner sampling were 5 folds). The lasso model is fitted into inner sampling of each 5-fold and the hyperparameter were tuned by the performance of classification (accuracy). Then a fold-specific classifier was trained by SVM-RBF in the selected features from the lasso model in the same fold. The performance (accuracy) of the fold-specific classifier was internally evaluated in the outer-sampling. The frequency of selected features was summarized in all models with accuracy >0.8. Top10 selected features were kept for visualization.

### Correlation analysis

Two types of correlation analysis either for between two continuous variables or categorial variables and continuous variables were including in this study. Continuous variables like cytokines, biomarkers and total lipid concentration were log2 transformed. Categorial variables like early death, treatment arm, TBI and coagulopathy were transformed into dummy variables. Euclidean distance matrix was calculated for correlation analysis. Spearman correlation coefficient was used for correlation between biomarkers and total lipid concentration or LRS due to consideration of non-linear relationship. Pearson correlation coefficient was used for correlation between lipid species due to the well-identified linear relationship. Statistical analysis for correlation coefficient is conducted by function rcorr() implemented in R package Hmisc(version 4.4.1). P values are approximate by using t distributions.

### Statistical analysis and visualization

Statistical analysis in this study was performed by using R language (version 3.6.0, https://www.R-project.org/)[59]. Pearson's χ2 test and Kruskal–Wallis test were used for categorical variables or continuous variables in the contingency table of clinical data. Kruskal–Wallis test with post-hoc analysis by Dunn test was used for multiple group comparisons. Two-way ANOVA with pair-wise comparisons by Estimated Marginal Means test was applied for time-series analysis. P value was adjusted by Benjamini–Hochberg method with less than

0.05 for establishing significance. Visualization of heatmap was performed by using R package Complexheatmap (version 2.5.2)[60]. Hierarchical clustering based on Euclidean distance was applied in rows or columns for heatmap construction.

## External metabolomics or lipidomics dataset

Three external datasets of untargeted metabolomics or lipidomics were included in this study. The first dataset was from a survival cohort that consisted of trauma patients with untargeted metabolome measurement[26]. The same criterion for outcome classification was applied in this group of patients to that used for the PAMPer dataset (Resolving: ICU Days <7; Non-resolving: ICU Days ≥ 7). The second dataset(Train set:C1, Test set: C2) was from a cohort of COVID-19 patients with both untargeted metabolome and proteome measurements[16]. The patients were grouped by severity defined in the previous study and days to timepoint 0, which was set as day of progression for severe patients and day of symptom onset for non-severe patients. The third dataset was from separate cohort of COVID-19 patients with both targeted and untargeted metabolome measurements[17]. The patients were not grouped by sampling time-point because of limited clinical information. Common lipids were identified by unique molecular formula or HMID from Human Metabolome Database among these 3 datasets and PAMPer lipidomic dataset. Mean z-score scaled value for each group for patients or healthy subjects was used to compare the lipid levels among 4 datasets. The two external datasets of COVID-19 were publicly available and the dataset from survival cohort of trauma patients was available upon request.

## Reporting summary

Further information on research design is available in the Nature Research Reporting Summary linked to this article.

## Data availability

Data underlying Figs. 1d, e, 2f, 5b, e and Supplementary Figs. 4a, 6g and 9b are provided as Source Data files. The lipidomics dataset generated in this study have been deposited in the Mendeley Data under https://doi.org/10.17632/7stf7dtxcz.2 (https://data.mendeley.com/datasets/7stf7dtxcz/draft?a=3e078e7f-5068-4b8e-a5a9-ef414db279bd) and are provided in Supplementary Data 11. The individual internal standard of Plasma lipidomic profiling by Metabolon Inc (Morrisville, NC 27560, USA) is commercially available (https://sciex.com/products/consumables/lipidyzer-platform-kits) and can be found in Supplementary Data 10. The public metabolomic or lipidomic dataset re-used in this study can be found with the original publications[16,17]. The remaining data are available within the article or from the authors upon request. Data from the survival cohort of trauma patients[26] re-used in this study were obtained through request and are not publicly available. Source data are provided with this paper.

## Code availability

Code supporting the current study is deposited at https://github.com/Junru-max/PAMPer-Lipidomic-analysis.

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

## Acknowledgements

This work was supported by US Army Medical Research and Materiel Command (W81XWH-12-2-0023) and National Institutes of Health (R35-GM-127027, U01HL137159, R01LM012087). We acknowledge the contribution of collaborators involved in PAMPer study for the clinical data collection. We acknowledge the contribution of Scott McCullough (Study Director, Academic - Metabolon, Inc) and Brian Ingram (Associate Director, Academic - Metabolon, Inc) for the technical support. JW was supported by Xiangya Medical School, Changsha, China. We thank Tiannan Guo and Guanghou Shui for providing public metabolomics/lipidomics dataset of COVID-19 patients.

## Author contributions

J.W. designed the overall workflow and performed data analysis. A.C., D.S.G., F.X.G., M.H.Y., B.J.D., R.S.M., B.G.H., J.A.C., H.A.P., B.S.Z., M.D.N., J.L.S., and T.R.B. designed the original study and sampling plan. Y.V., P.I.J., J.S., U.K.K., and D.A.B. analyzed samples. T.C.L. and P.V.B. performed analysis of causal modeling and helped interpret the results. J.D. supervised the prognostic analysis and helped interpret the results. T.C. and R.A.N. analyzed the data in TD-2 dataset. J.W. and T.R.B. wrote the manuscript with the feedback of all of the authors who have read and approved the manuscript. PAMPer study authors contributed to patient enrollment and sample procurement.

## Competing interests

The authors declare no competing interests.

## Additional information

## PAMPer study group

**Mazen S. Zenati**[1], **Joshua B. Brown**[1], **Darrell J. Triulzi**[1], **Barbara J. Early Young**[1], **Peter W. Adams**[1], **Louis H. Alarcon**[1], **Clifton W. Callaway**[1], **Raquel M. Forsythe**[1], **Donald M. Yealy**[1], **Andrew B. Peitzman**[1], **Meghan L. Buck**[1], **Ashley M. Ryman**[1], **Elizabeth A. Gimbel**[1], **Erin G. Gilchrist**[1], **Meghan Buhay**[1], **Chung-Chou H. Chang**[1], **Victor B. Talisa**[1], **Tianyuan Xu**[1], **Kyle Kalloway**[11], **Andrew Yates**[11], **Susan Rawn**[11], **Judith M. Jenkins**[12], **Laura S. Trachtenberg**[13], **Randi K. Eden**[13], **Joanne Fraifogl**[14], **Craig Bates**[14], **Christina Howard**[15], **Cari Stebbins**[15], **William R. Witham**[19], **Cathy McNeill**[19], **A. Tyler. Putnam**[20], **Amy Snyder**[20], **Jason Ropp**[20], **Therese M. Duane**[21], **Celeste Caliman**[21] & **Mieshia Beamon**[21]

[19]Department of Surgery, Texas Health Harris Methodist Hospital, Fort Worth, TX, USA. [20]Department of Surgery, Altoona Hospital of University of Pittsburgh Medical Center, Altoona, PA, USA. [21]Department of Surgery, John Peter Smith Health Network, Fort Worth, TX, USA.

