## [Peer Review File · Nature Communications]

Reviewer comments, first round –

Reviewer #1 (Remarks to the Author):

In the manuscript by Wu et al, the authors propose the use of a lipidomic approach to assess the status of acute critical illnesses and the introduction of a Lipid Reprogramming Score (LRS) as a prognostic tool to evaluate progression towards severe conditions. The relevance of this work is high, in principle, as it introduces the use of quantitative molecular readouts to understand the physiological processes underlying insurgence of a critical condition.

The authors made important efforts in terms of sample and data analysis. However, the manuscript as it stands need to be revised in a major way. It is hoped that the following comments will be helpful. Please note that this review is focussed and limited to the lipidomic aspects of this study rather than trauma medicine or general data analysis and statistics. Additional reviewers should be consulted for the latter two areas.

Major areas of concern

1) Lipidomic Methodology and reporting of raw data

The lipidomic methodology is not described in sufficient detail, for example:

a. Line 113: "In the quality control analysis, the median relative standard deviation (RSD) for the lipid panel was 4%". How was the quality control sample generated and how often was it analysed during the experiment? What were the RSD values for the 900 lipids? What was the RSD threshold for lipid selection?

b. Line 464: "Lipid species were background-subtracted using concentrations detected In process blanks..." what was the signal to noise (S/N) threshold for acceptance of results?

c. The nomenclature of all lipids reported is based on sn-1 or -2 location of fatty acids (PE 16:0/18:2); this might not be possible with the used setup unless the authors can explain how the position of the fatty acids was verified. An alternative possible nomenclature for the same species might be according to PE 16:0_18:2. Please see Shorthand notation for lipid structures derived from mass spectrometry. *J. Lipid Res.* 2013. 54: 1523–1530.

d. Line 472: "Lipid with over 80% missing values were discarded..." did you mean over 20%?

e. No raw lipidomic data is reported.

f. While 900 lipids are measured, close to 2/3 of the lipidomic panel covers glycerolipids. No lipid mediators were included which are more closely linked to physiological stress induced by imbalances in redox biology or inflammation. What is the reason for this? In any case, it should be discussed at least.

g. Since there is an analytical bias towards non-polar lipids, have the authors measured these compounds also via more traditional clinical lipid panels? How about cholesterol levels?

h. The healthy subject baseline group is very small (HC, n=17). Furthermore, the authors acknowledge the differences in sampling between the HC and trauma groups. PE is a strong function of daily nutrition and sampling (e.g. Sweeney, G., Nazir, D., Clarke, C. & Goettsche, G. Ethanolamine and choline phospholipids in nascent very-low-density lipoprotein particles. *Clin Invest Med* 19, 243–250 (1996)). Could this be relevant for the observed differences between HC and trauma patients?

2) Meta-analysis of COVID-19 datasets

This aspect of the manuscript is not convincing. It is certainly desirable to extend the primary experimental research findings to more generalisable insight. But the attempt to do this in the context of COVID-19 falls short.

a. First, as mentioned in the introduction, a major limitation in studying critical illness is the lack of knowledge in precise onset of the process. This is in fact the case for COVID-19 but not the trauma cohorts described here.

b. The utilised analytical (metabolomic and lipidomic technologies) differ considerably between the 4 studies compared here (e.g. untargeted vs targeted analysis, without and with chromatography, calibrators for quantification, etc). Therefore, direct comparison of results will require more detailed and systematic cross-comparison which in addition to the biological metabolite variations need to take into account also the associated technical variabilities of the different studies. Accordingly, line 234 reads “Eight lipids from PE class... four PC or PI were higher in the non-resolving trauma patients (72h) or severely ill COVID-19 patients in at least one dataset (out of 4 in total). This seems like a very low bar for inclusion.

c. Validation is not based on independent cohorts interrogated experimentally. One would need prospective trials to support the claims for the prognostic panel.

3) Generalized interpretation

a. The last sentence of the discussion should elaborate more on the possible use of these data and the LRS, especially because a single marker measured in the clinic, CRP, can have a better predictive value. The same is true for clinical lipid measurements which might provide equally informative results (see also 1.g. above).

b. Could other examples of critical illness be mentioned for future validation of this lipid score, beyond COVID-19?

c. The results presented in Figure 4G are interesting. However, it is not clear why lipid metabolism should be at the center and so closely linked to early death.

Other comments

- End of line 378: reference cited should be 38 and not 39

- Line 375: glycolipids? Why?

- Figure 7C is not clear in some parts: lipolysis and lipogenesis should not be on the y axis to avoid

confusion; the lipids increasing in the right part of the figure are apparently coming from few classes only.

- Line 336: a persistent lowering of circulating lipids through 72 hours...I don't think this is what is shown in figure 7C. Could the authors please check?
- How was the fatty acid data displayed in Figure 2 calculated from the lipidomic measurements?
- The main commonality is major reduction of overall lipid concentrations upon trauma.
- Are HC the most relevant reference? A more appropriate reference condition might be patients after they recovered.
- The correlation analysis shown in Fig 3A likely reflects some form of associations between different lipids as part of their lipoprotein status.
- "Causal" vs "Casual" in the context of inference analysis and modeling. Please check spelling.

Reviewer #2 (Remarks to the Author):

The studies provide data extensive data on circulating lipids in the plasma in patients who suffered major trauma, with an effort to identify patterns that would provide prognostic information. Some of the data was also applied to severe COVID-19 patients. While there is some merit in this study, there are several limitations. In the primary cohort, patients were randomized to receive FFP 2 units or standard of care, and as reported before, the FFP-treated patients had increased survival.

1. This is a major descriptive effort to try to identify patterns of changes in lipid levels that would add to our understanding of how trauma causes organ injury and death.
2. There were 996 lipids measured with TG the most abundant followed by PE, PC and DAG. Most lipids declined in following trauma.
3. Figure 2 shows some patterns of lipid changes depending on the patient outcomes with an unexplained increase in four lipid classes at 72 hours in patients that died or remained critically ill. here was no impact of the standard Injury Severity Score on lipid patterns.
4. The FFP-treated patients had a rise in lipid profile toward normal than dissipated at 24 and 72 hours.
5. Lower lipid levels seem to correlate with several know pro-inflammatory cytokines and coagulation factors(Figure 7s).
6. The authors try to make sense of all of this data by then putting it into probabalistic graphical models in an algorithm in Figure 4G. The findings seem to identify potentially causal factors including lipid concentrations, coagulation levels, crystalloid administration plus levels of pro-inflammatory factors.
7. The authors then try to validate these findings in other trauma data sets. The primary pattern of decline in lipid levels after trauma was identified. They then generated a Lipid Reprogramming Score (LRS) using 8 PE species that were common to the each data set. Figure 6 shows how this worked out in analyzing relationships to clinical course and outcomes. The LRS was then correlated with markers

of inflammation and endothelial injury.

8. Then then correlated the LRS with COVID-19 patients. This analysis seems especially weak and under-developed and tacked on - would probably need to be part of a separate report.

9. Having worked through all the results and figures, I do not feel enlightened for pathogenesis of trauma induced injury and the prognostic signals seem so complex to make it difficult to see how they could be used well. The manuscript is long and somewhat tedious and the end result is not illuminating. It reads more as a compendium of findings without clear significance.

10. The authors conclude that they have provided a new paradigm for lipid response to severe and acute stress. But I do not find that I can really understand tangibly this new paradigm except in very broad strokes. Why does it have no relationship to the ISS or clinical outcomes in a more tangible way?

11. The authors have acknowledged the limitations of their study well in the second to last paragraph of the discussion.

Reviewer #3 (Remarks to the Author):

Wu et al provided an impressive study on lipidomic signatures in patients with critical illness provide a new paradigm for the lipid response to a severe and acute 407 systemic stress leading to critical illness.

Despite the complicated methodology, the work looks appropriate, and the authors had the brilliant idea of including a cohort of COVID-19 patients.

I honeslty have some minor comments:

- can the authors provide some more rational on the pathophysiological role of lipids in inflammation, maybe even with a supplementary material included in the introduction? I am aware this is unusual, however this paper can be a practice changing one in the middle term and revolutionize the field of inflammatory biomarkers, which has impact on a wide range of specialists (most of us would not have a basic knowledge on the topic).

-do the authors have basic lipid prophyles of these cohorts (colesterole, LDL, HDL, and so on?) does these values may be influenced by baseline "body composition" or even by drugs which my interefere with lipid methabolisms? If there is a rational, the authors may provide these data (including BMI) or, if not possible , this should be mentioned as a limitation

- the authors may provide in the discussion some perspective for future fields of study. In particular, I am fascinating about the rapid changes in prophiles, which may be particularly useful in a subset of patients that mostly suffer from acute/hyperacute conditions which we often have difficulties for early diagnosis (I mostly think at sepsis in young children, infants and neonates, for which we do not yet have proper biomarkers)

Reviewer #4 (Remarks to the Author):

Overall feedback

This piece of research aims to tackle the problem of accuracy predicting outcomes of patients suffering with trauma and SARS CoV-2 (COVID-19) infection using a derived Lipid Reprogramming Score (LRS). This would prove useful especially in our current environment in which health care facilities across the globe are tacking severe COVID infection. In this regard, this piece of scientific research, from my point of view, aims to add knowledge to existing literature at this relevant time. That said, as I am not an expert on either COVID-19, Critical Care (trauma treatment) or lipid biology, I cannot comment on the clinical relevance of the data used. Therefore, I will only provide feedback related to the statistical methodology used in the paper.

As I already mentioned above, I think this piece of work has potential to contribute in helping tackle an important problem. The strengths of the paper is that the authors come up with the LRS that would be beneficial in making prognostic predictions for trauma and COVID-19 patients. Further to this, the authors have made an attempt to externally validate the use of LDS using real world data (TD-2).

However, I do find that the execution of the work to be limited in the following ways. First, there are too many analyses and figures for the readers to go through and make sense of the results contained in the paper. My impression is that the authors want to include all the results that they deem to be interesting results in the paper. However, by doing this, they (may) inadvertently make it difficult for the reader to follow what is being presented. I think it much better to conduct only the analyses that support the main theme of the paper. For example, it may be that some correlation plots / exploratory graphs may not be needed in passing the intended message. Second, I think somehow related to the first point, since there are many analyses, some were not conducted to the required standard. Having fewer analyses would enable greater depth. For example, not sufficient detail is provided about the performance measures of the internal and external validation as required for prediction models. Third, not sufficient justification was provided for the predictors in the model (specifically, were the predictors selected because of their clinical utility or was a variable model selection procedure utilised), sample sizes (not easy to tell in some instances) and distribution of the outcome and predictors were not provided for the each of the statistical model so I was not able to fully appraise them.

Detailed feedback:

For further information and benefit to the authors, I categorise my feedback into two- (A.) minor, requiring very little effort to address and (B.) major, requiring moderate to substantial effort to deal with. I now list my comments below under each category in random order.

A. Minor :

1. The reference for the PAMPer trial should be numbered (18) and not (19).
2. Line 103 says that there were non-survivors (n=72) and survivors (n=121), a total of 193 patients selected for lipidome analysis. However, Fig S1, contradicts this by showing that the 193 selected consisted of non-survivors (n=83) and survivors (n=101).
3. What was the distribution (in Fig S1) of the non-survivors (plasma / soc)? How were they distributed across the different sites in the original data?

4. Please replace all references of “logistical regression” with “logistic regression”. For example, see lines 192,528,539, 541, 543 etc

5. The statement in line 539 should be re-written to explicitly communicate how survival and severity was categorical (so as not to leave any room for misinterpretation).

6. It is not clear to me what is mean in line 540 by “only main effect of each factor was evaluated”

7. In line 543, I think “across” should be replaced with “for each”.

8. In line 440/441, “trail” should be replaced with “trial”.

9. Line 425, “analyze” should read “analyzed”.

10. In line 170, was there a (clinical) justification for categorising ISS? If this was done arbitrarily or for convenience, I think this should be made clear.

B. Major:

11. For the statistical model fitted in the PAMPer trial, the clustered nature of the data (sites) was accounted for using a hierarchical model using ICC (0.05), I have not seen how the clustered nature of the data used for this study was accounted for in the analysis. This may have type 1 error (false positive) rate implications.

12. How were the 15 control patients (line 106) chosen? Were they matched to ensure valid comparison with the other 193 patients? Table 1 does not provide any information about them. This should be reported, for example, Guo et. al report the controls they used and explain how they were matched.

13. I would recommend that the exploratory data analysis for the predictors and outcome used for each model be conducted. As far as I can tell, Table 1 gives information on all the 193 patients. For reader interested in determining sample size for each analysis, it is not easy to decipher. For example, for the logistic model of early death (early-non survivors), I reckon data for 51 patients were analysed. However, I am having a difficult time telling the sample size for the model analysing COVID-19 data (from Guo et. al – was the entire dataset on which their paper was based used? Or was a subset used? This should be clear in the paper).

14. The authors include references to missing data in line 472, can I confirm that the percentage in Table 1 do not include missing values. In other words, do we have any missing values for the variables? I suspect no missingness based on presentation of the table.

15. Prognostic models, need to be trained and tested using the same variables. Is there a justification for developing the survival model (time to discharge by ICU) using the variables reported in 545/6 but externally validating the model onTD-2 data (line 548) that contains all but one variable used in the internal validation? As I understand this is not typically done.

16. How were the variables included in each model fitted selected? Was this based on clinical utility or a variable selection procedure?

17. How was the prognostic /prediction model developed evaluated? Prognostic models need to be internally validated based on their (i) Clinical utility [1], (ii) Discrimination (iii) and Calibration. The above measures can be corrected for bias and optimism using resampling methods (bootstrap or k-fold cross validation). Without these performance measures, it is difficult to objectively conclude the usefulness of the prognostic model used. Please see this papers as well [2,3].

References:

1. Van Calster B, Wynants L, Verbeek JF, Verbakel JY, Christodoulou E, Vickers AJ, Roobol MJ, Steyerberg EW. Reporting and interpreting decision curve analysis: a guide for investigators. *European urology*. 2018 Dec 1;74(6):796-804.
2. Smith GC, Seaman SR, Wood AM, Royston P, White IR. Correcting for optimistic prediction in small data sets. *American journal of epidemiology*. 2014 Aug 1;180(3):318-24.
3. Steyerberg EW, Bleeker SE, Moll HA, Grobbee DE, Moons KG. Internal and external validation of predictive models: a simulation study of bias and precision in small samples. *Journal of clinical epidemiology*. 2003 May 1;56(5):441-7.

Ms. No.: NCOMMS-20-46047-T

Title: Lipidomic Signatures Align with Inflammatory Patterns and Outcomes in Critical Illness

Summary of the major concerns and general response

We greatly appreciate the reviewers' efforts to review our manuscript and for their insightful comments to improve our paper. By addressing the concerns raised by the reviewers, we believe the paper has been substantially improved. Our revisions include two major re-analysis of the data based on the valuable comments of the reviewers. We also significantly re-wrote sections of the results to improve the readability. These major changes include the following:

1. We conducted the sensitivity analysis for the generation of the Lipid Reprogramming Score (LRS) to identify the phosphatidylethanolamines (PE) most representative of the four trauma and COVID-19 datasets. Briefly, a set of five PE that included PE from all four datasets yielded a LRS that was highly associated with adverse outcomes (critical illness) in both trauma and COVID-19 patients.
2. We adopted new statistical models to account for cluster effect based on standard logistic and cox regression models. This corrected a major shortcoming of the previous analysis without altering the conclusions of study.
3. To increase the readability, we removed one of the main figures (Figure 3) and placed this in the supplemental section and only briefly mention this analysis of the relationships between lipid species in the results. We also significantly simplified the description of the generation of the Lipid Reprogramming Score in the results section. Instead, a more detailed description of the steps used to develop the score has been placed in the Methods section.
4. We have added considerable detail on the technical aspects of the targeted lipidomic analysis platform (Metabolon Inc.).
5. The plasma given to patients en route to the hospital in the treatment arm is now referred to as thawed Plasma (TP) instead of Fresh frozen plasma (FFP). This is more accurate because the plasma had been thawed prior to being placed on the helicopters. Therefore, the reviewers will see that FFP has been changed to TP throughout the manuscript.

Point-by-point Response to the Reviewers' Comments

Reviewer #1:

In the manuscript by Wu et al, the authors propose the use of a lipidomic approach to assess the status of acute critical illnesses and the introduction of a Lipid Reprogramming Score (LRS) as a prognostic tool to evaluate progression towards severe conditions. The relevance of this work is high, in principle, as it introduces the use of quantitative molecular readouts to understand the physiological processes underlying insurgence of a critical condition.

The authors made important efforts in terms of sample and data analysis. However, the manuscript as it stands need to be revised in a major way. It is hoped that the following comments will be helpful. Please note that this review is focussed and limited to the lipidomic aspects of this study rather than trauma medicine or general data analysis and statistics. Additional reviewers should be consulted for the latter two areas.

Response to general comments: We sincerely thank the reviewer for carefully reviewing our manuscript and for the suggestions to improve the paper. The point-by-point responses are listed as follows:

Major areas of concern:

1) Lipidomic Methodology and reporting of raw data

Question 1: Line 113: “In the quality control analysis, the median relative standard deviation (RSD) for the lipid panel was 4%”. How was the quality control sample generated and how often was it analysed during the experiment? What were the RSD values for the 900 lipids? What was the RSD threshold for lipid selection?

Response: We are pleased to provide this important information. The QC sample was generated by combining a small aliquot from entire set of samples into a single pooled CMTX (Sample matrix). Four aliquots of the CMTX were run on each plate of 36 samples. One each was injected at the beginning and end of the run, with the other two roughly evenly spaced in between the remaining samples. A similar process was carried out for three process blanks (water) and MTRX7 (a well characterized plasma sample Metabolon runs on every plate). Unfortunately, Metabolon does not report on the RSD values for each biochemical, only the median value. There is no threshold for lipid selection (or any biochemical selection in the untargeted dataset) based on RSD. Metabolon calculated the median RSD from all species detected in 100% of the CMTX samples. We have now added this information in the methods section.

Question 2: Line 464: "Lipid species were background-subtracted using concentrations detected In process blanks..." what was the signal to noise (S/N) threshold for acceptance of results?

Response: Background levels were estimated/calculated from the median levels of the three process blanks (water) if there were detectable levels in at least 2 of the 3 blanks in each batch. The background level was subtracted from each sample in the batch prior to any run day normalization. If the value was negative, it was set to zero or "not detected". Metabolon explained to us that after extensive analysis, they have found that implementing this step gives much better data quality for some lipid species that can be found on plastic tubes or other artifactual sources in their system. We have now added the approach to subtract background levels to the methods section.

Question 3: The nomenclature of all lipids reported is based on sn-1 or -2 location of fatty acids (PE 16:0/18:2); this might not be possible with the used setup unless the authors can explain how the position of the fatty acids was verified. An alternative possible nomenclature for the same species might be according to PE 16:0_18:2. Please see Shorthand notation for lipid structures derived from mass spectrometry. *J. Lipid Res.* 2013. 54: 1523–1530.

Response: Unfortunately, the complex lipid panel (CLP) platformed by Metabolon does not distinguish between SN1 and SN2 location of the side chains. The nomenclature used simply lists the 2 sidechains present without attempting to ascribe which resides at which position. We have now added this information to the methods section.

Question 4: Line 472: "Lipid with over 80% missing values were discarded..." did you mean over 20%?

Response: Thanks for pointing out the typo. The corrected version states, "Lipids with over 20% missing values were discarded". We attached a histogram plot for the distribution of missing lipids. Based on the exploratory analysis, we kept all lipid species with over 80% non-missing values for the subsequent analysis.

Question 5: No raw lipidomic data is reported.

Response: We also would very much like to have this information, but unfortunately Metabolon does not report this information for their untargeted or CLP platforms. We inquired whether we could purchase this information and this is not feasible.

Question 6: While 900 lipids are measured, close to 2/3 of the lipidomic panel covers glycerolipids. No lipid mediators were included which are more closely linked to physiological stress induced by imbalances in redox biology or inflammation. What is the reason for this? In any case, it should be discussed at least.

Response: We agree that levels of lipid mediators would be of considerable interest. However, largely for technical reasons, Metabolon does not include mediators in their CLP platform. Since the

CLP is technically a targeted assay, they have chosen to detect the lipid classes shown in Fig. 1B. We now add to the discussion that the absence of lipid mediators as a limitation of the study.

Question 7: Since there is an analytical bias towards non-polar lipids, have the authors measured these compounds also via more traditional clinical lipid panels? How about cholesterol levels?

Response: This is an excellent suggestion. Unfortunately, the clinical laboratory lipid panels are not routinely measured in trauma patients. In fact, we could not find an instance where lipids were measured at the time of admission in our patient cohorts.

Question 8: The healthy subject baseline group is very small (HC, n=17). Furthermore, the authors acknowledge the differences in sampling between the HC and trauma groups. PE is a strong function of daily nutrition and sampling (e.g. Sweeney, G., Nazir, D., Clarke, C. & Goettsche, G. Ethanolamine and choline phospholipids in nascent very-low-density lipoprotein particles. *Clin Invest Med* 19, 243–250 (1996)). Could this be relevant for the observed differences between HC and trauma patients?

Response: Thank you for raising these important points. The control samples were intentionally drawn from non-fasting healthy subjects to act as suitable controls for the admission time point for the trauma patients (designated as the 0h timepoint). While we had only 17 control subjects, a recent paper (Tarazona et al., 2020) concludes that the minimal desirable sample size for omics (i.e. Metabolomics/Proteomics) is around 16. This number was shown to yield statistical power of 0.6-0.8 for differential analysis. In addition, we show that the heterogeneity among healthy subjects is much lower than trauma patients in the UMAP plot (Fig2A). Overall, while we agree that more healthy controls may have been preferable, we think the number of healthy subjects was adequate to perform the downstream analysis.

As for the difference of PE, we are confident that the late increases in PE species is not influenced by nutritional sources since none of these critically ill trauma patients received nutritional support over the first 3 days of hospitalization. It is noteworthy that the subset of trauma patients that exhibit an increase in PE at 72 h are those that remain critically ill longer than those that do not experience an increase in PE. The increases in PE in these patients is even higher than the healthy controls. Thus, the higher circulating PE levels mostly likely represents changes in endogenous processes that determine levels of circulating PE. We speculate that this is part of lipid reprogramming that takes place in the setting of persistent critical illness.

2) Meta-analysis of COVID-19 datasets

This aspect of the manuscript is not convincing. It is certainly desirable to extend the primary experimental research findings to more generalisable insight. But the attempt to do this in the context of COVID-19 falls short.

Response to general comments: Thank you for raising these important concerns over our analysis in COVID-19 datasets. Our point-by-point responses are as follows:

Question 1: First, as mentioned in the introduction, a major limitation in studying critical illness is the lack of knowledge in precise onset of the process. This is in fact the case for COVID-19 but not the trauma cohorts described here.

Response: This is an excellent point and one rational for why trauma can serve as an excellent derivation dataset for other critical illnesses (i.e., Severe COVID-19). While we certainly expect there to be major difference in critical illness caused by trauma vs. other etiologies, there are also likely to be human responses that are in common during the response to acute critical illness. We were intrigued to see that critically ill Covid-19 patients also exhibit a rise in PE. While much more work is needed to understand the basis of this common response, we are very cautious not to over-interpret these findings. We have further modified our description to state (line 229 to 233), **“Unlike trauma, the onset of critical illness in Covid-19 patients can be highly variable relative to the onset of infection and the time the infection started is often unclear. To assist with the comparison between the trauma and COVID-19 datasets, we set the 0 timepoint in the COVID-19 datasets as the day of symptom onset for non-severe patients or day of progression for severe patients.”**

Question 2: The utilised analytical (metabolomic and lipidomic technologies) differ considerably between the 4 studies compared here (e.g. untargeted vs targeted analysis, without and with chromatography, calibrators for quantification, etc). Therefore, direct comparison of results will require more detailed and systematic cross-comparison which in addition to the biological metabolite variations need to take into account also the associated technical variabilities of the different studies. **Accordingly**, line 234 reads **“Eight lipids from PE class... four PC or PI were higher in the non-resolving trauma patients (72h) or severely ill COVID-19 patients in at least one dataset (out of 4 in total). This seems like a very low bar for inclusion.**

Response: We appreciate the reviewer’s concerns about the differences in the techniques used across the four datasets to measure lipid levels. Here, we kept only lipids species from the four databases that had the same ID from the Human Metabolome Database (<https://hmdb.ca/>). To address the variability in the techniques used across the four datasets we applied only indirect comparisons. We carried out the indirect comparison by scaling and centering all the lipids within each dataset to identify the lipids within each dataset that were different between controls and patients with trauma or Covid-19. This allowed us to identify common lipids that were elevated across all the datasets. We now acknowledge in the results, “Because there were differences in the methods used to quantify lipid species across the four datasets, we could only carry out indirect comparisons of the relative changes of lipids species within each dataset across the datasets.”

In order to address the reviewer’s concern about the representation and inclusion of the PE species used to generate the Lipid Reprogramming Score (LRS) within the four datasets, we have re-done this analysis. This led to the identification of 5 PE species elevated in common in 3 out of 4 of the datasets as the optimal set of PE species to calculate the LRS. However, the 5 PE species also includes at least one PE species from all four datasets.

We first summarize the number of PE species elevated in common with the PAMPer database when including one, two or all three of the datasets in addition to the PAMPer dataset. It can be seen in the table below (**Table S5**) that by including two datasets (PAMPer + any 1 other dataset) 8 PE species identified in the PAMPer dataset can be found in common. Increasing the number of datasets to three (PAMPer + any 2 other datasets) the number of PE species in common drops to 5 out of 8. Only one PE species (PE(18:0/18:1)) was found to be in common to all 4 datasets.

At least in PAMPer	At least in PAMPer and at least (≥ 1) in other 3 datasets	At least in PAMPer and at least (≥ 2) in other 3 datasets	At least in PAMPer and at least ($= 3$) in other 3 datasets
9 species: PE(16:0/18:2), PE(16:0/20:4), PE(16:0/22:6), PE(18:0/18:1), PE(18:0/18:2),	8 species: PE(16:0/18:2), PE(16:0/20:4), PE(16:0/22:6), PE(18:0/18:1), PE(18:0/18:2),	5 species: PE(16:0/18:2), PE(16:0/20:4), PE(16:0/22:6), PE(18:0/18:1), PE(18:0/22:6)	1 specie: PE(18:0/18:1)

PE(18:0/20:4), PE(18:0/22:6), PE(18:1/18:2), PC(18:0/18:1)	PE(18:0/20:4), PE(18:0/22:6), PE(18:1/18:2)		
---	---	--	--

Table S5 Relationship of differentially lipids species across four trauma and Covid-19 datasets

We then conducted a sensitivity analysis by using 8 PE species (PAMPer + 1 other dataset), 5 PE species (PAMPer + 2 other datasets) or 1 PE specie (PAMPer + all 3 other datasets) to generate the LRS (**Table S5**). While the LRS based on 5 PE or 8 PE species yielded high correlations based on Pearson correlation coefficients ($r=0.97-0.98$), the r values dropped ($r=0.78-0.86$) when using the single PE common to all four datasets. The LRS (5 species) was associated with improved performance in COVID dataset (guo et al.) compared to LRS (8 species) (OR 9.98 vs 4.67). In addition, the performance (OR in PAMPer : 0.73 vs 0.69, OR in TD-2 : 0.73 vs 0.77) was comparable in Trauma datasets. This indicates that the PE species have a high degree of correlation with each other and that using a minimum of 5 PE species, all in common within 3 out of 4 datasets, should lead to the most stable model. We then re-performed all the analysis using the five PE species LRS calculation for the correlation analyses and indeed found better results. Thus, we replaced all our original analysis (LRS from 8 species) with the new analysis (LRS from 5 species). This is represented by changes in figures 6-7 and supplementary figures 6-7 as follows:

Fig 6E: p-value is lower for both two timepoints (<48h, d6-d14) and there is a statistical difference in d6-d14($p<0.05$).

Fig 6F: AUC value for LRS is increased from 0.788 to 0.828. Now its performance is superior to lymphocyte count and CRP.

Fig 6G: OR ratio of LRS is increased.

Fig S6F: Clear tendency for the association of LRS with severity of COVID-19 in the dataset of Shui et al..

Fig 7B: Increased correlation coefficient between LRS and CRP, LBP and other markers of the acute phase response.

2) Consistent performance of LRS (5 species) compared to LRS (8 species) in Trauma datasets:

Fig 6B-D, Fig S6G-I, Fig 7A, Fig S7A-B: All the explanatory and regression analysis are consistent after the modification in both PAMPer and TD-2 datasets.

3) Other changes:

Fig 6A: Add the section for sensitivity analysis.

Fig 7C: Change the format for summarized findings.

sFig 7B-C: Number of Positively correlated proteins changed from 151 to 141. The number of negatively correlated proteins changed from 24 to 25.

sFig 7F: Pathway of "HDL modeling" is replaced by "Scavenging of heme from plasma".

Finally, we modified the relevant sentence in the manuscript to read:

Line (235-241):

“We identified eight PE species and one PC specie significantly higher in the non-resolving group (72h) in PAMPer dataset, while three, six, and five of these PE were elevated in the TD-2, Covid-19 (Guo et al)¹⁶, and Covid-19 (Shui et al)¹⁷ datasets, respectively. Eight of these PE species could be identified when combining PAMPer with any single other database, five PE species were in common when combining PAMPer with any two of the other databases, and a single PE was found to be significantly elevated during critical illness in common across all four databases (Table S5). Thus, increases in PE consistently associate with critical illness in trauma and COVID-19.”

Line (249-252):

“A sensitivity analysis identified a model comprised of five PE showing the best performance (Table S6). Thus, we defined the LRS as the mean z-score of five PE species (PE(16:0/18:2),PE(16:0/20:4), PE(16:0/22:6), PE(18:0/18:1), PE(18:0/22:6)) representatives of PE from all four trauma and COVID-19 datasets.”

Table S6 Sensitivity analysis of LRS generation by different species of PE							
	r (PAMPer)	r (TD-2)	r (guo et al.)	r (shui et al.)	HR (PAMPer)	HR (TD-2)	OR (guo et al.)
LRS(8 Species)	ref.	ref.	ref.	ref.	0.69 (0.53-0.91)	0.73 (0.55-0.98)	4.67 (1.24-25.20)
LRS(5 Species)	0.98	0.98	0.97	0.97	0.73 (0.56-0.95)	0.77 (0.58-1.02)	9.98 (2.09-78.50)
LRS(1 Specie)	0.8	0.78	0.86	0.8	0.64 (0.49-0.83)	0.67 (0.51-0.88)	1.81 (0.80-4.80)

Pearson correlation coefficient was calculated between the LRS (5 species) or LRS (1 specie) and LRS (8 species) from **Table S5**.
 In PAMPer dataset, Hazard ratio for LRS was calculated by cox regression model with mixed effect and adjusting for Age, Gender, ISS, Head Injury and Arms. In TD-2 dataset, Hazard ratio for LRS was calculated by standard cox regression model and adjusting for Age, Gender, ISS, Head Injury. In dataset of guo et al., odd ratio for LRS was calculated by logistic regression model and adjusting for Age, Gender, Lymphocyte count and CRP.

By addressing the reviewer’s concern, we have optimized the formulation of the LRS and developed a LRS more representative of critical illness in trauma or Covid-19.

Question 3: Validation is not based on independent cohorts interrogated experimentally. One would need prospective trials to support the claims for the prognostic panel.

Response: We concur with the reviewer's point. Our study is limited to an exploration of the association between LRS and outcome in two trauma datasets that was then generalized by a retrospective analysis of two COVID-19 datasets. Therefore, we have removed all references to prognostication and use the term "association" in discussing the correlation between LRS and outcomes.

3) Generalized interpretation

Question 1: The last sentence of the discussion should elaborate more on the possible use of these data and the LRS, especially because a single marker measured in the clinic, CRP, can have a better predictive value. The same is true for clinical lipid measurements which might provide equally informative results (see also above).

Response:

Thank you for this excellent point. We have added to the last sentence to read (line 383-385) "The LRS, like other biomarkers of the host response (e.g., IL6 and CRP), may be useful as early parameters for outcome prediction linked to specific biologic processes."

For the reviewer's interest, we attach a comparative analysis addressing the association of the LRS and total lipid concentration with outcomes (see below). Our finding suggests that the LRS is an independent risk factor for outcome, while there was no significant association between total lipid concentration and outcome. Thus, it may be not be possible to use the currently available clinical lipid blood tests for outcome prediction.

Question 2: Could other examples of critical illness be mentioned for future validation of this lipid score, beyond COVID-19?

Response: We are pleased to mention other examples of causes of critical illness (e.g., sepsis and burns) that should be examined in future validation work. Specifically, we state (line 385-386), “It will also be of interest to determine if the LRS predicts outcomes in other etiologies of critical illness such as sepsis and burns.”.

Question 3: The results presented in Figure 4G are interesting. However, it is not clear why lipid metabolism should be at the center and so closely linked to early death.

Response: Thank for raising this important point. The plot presents a causal network among all the factors included in the causal modelling. The position for each variable is not intended to signify its potential importance relative to other variables in the model. A direct line indicates a potential causal relationship between any two variables. Here, we show that the lipid concentration may be an independent factor related to death. The connection between prehospital administration of plasma

and lipid levels implicates a causal relationship between lipid levels and pre-hospital plasma administration. We positioned plasma treatment and lipid levels to highlight the possibility that early plasma may improve outcomes, in part, through the preservation of lipid levels. We have made it clear in the description of the modelling how to interpret the connections in the diagram.

4) Other comments:

Question 1: End of line 378: reference cited should be 38 and not 39

Response: Thanks for pointing out the typo and the citation error. Since we added several citations to the introduction section in the revised version, the sentence now reads as follows:

“Several specific lipid species [e.g. PC(16:0/18:1), PC(18:0/18:1)] are known to contribute to inter-organ (liver, muscle and adipose tissue) communication⁴³.”

The refence is listed as follows:

“43. Liu, S., Alexander, R. K. & Lee, C.-H. Lipid metabolites as metabolic messengers in inter-organ communication. *Trends Endocrinol. Metab.* **25**, 356–363 (2014).”

Question 2: Line 375: glycolipids? Why?

Response: Thank you for pointing out the error. We changed glycolipids to glycerolipids when discussing the increases in TAG and DAG.

Question 3: Figure 7C is not clear in some parts: lipolysis and lipogenesis should not be on the y axis to avoid confusion; the lipids increasing in the right part of the figure are apparently coming from few classes only.

Response: Thank you for the helpful observation. We now make it clear that the Y-axis reflects the directional changes in lipids (all drop early and a few increase late in a subset of patients). We also color code the patient trajectories. Below we show the original figure 7C and also our new modified version.

Question 4: Line 336: a persistent lowering of circulating lipids through 72 hours...I don't think this is what is shown in figure 7C. Could the authors please check?

Response: We agree that this sentence could have been better stated. We have revised as follows, "A drop in circulating lipids persisted through 72h in patients destined to resolve their critical illness earlier." We have also modified Figure 7C to make the diagram easier to interpret this point.

Question 5: How was the fatty acid data displayed in Figure 2 calculated from the lipidomic measurements?

Response: The fatty acid reflects the sum of the concentrations for all species that contained the designated fatty acid. For example, the levels of FA 20:4 reflects the sum of the concentrations for all lipid species that contain FA 20:4.

Question 6: The main commonality is major reduction of overall lipid concentrations upon trauma.

Response: We concur. This is observed within an hour or two after major trauma, so clearly happens very fast. Importantly, this has not been shown before and, indeed, is one of the major discoveries of the study.

Question 7: Are HC the most relevant reference? A more appropriate reference condition might be patients after the they recovered.

Response: We agree that having the right controls is essential to the interpretation. In theory, the best controls would be the patients themselves prior to injury. For obvious reasons, this is not possible. The reviewer is correct that a baseline once the patient has recovered would also serve as an excellent control. Unfortunately, 30% of these patients die within 72 hours making this option impossible for some of the most interesting patients. The remaining patients may not return to their true baseline for weeks to months due to the prolonged time it takes to recover from polytrauma, especially in patients with severe traumatic brain injury. The study was not approved for delayed sampling in the surviving patients.

This leaves us with healthy age- and sex-matched non-fasting subjects as the best available option for controls. We note that the HC are most relevant to the 0h time point data but can also be viewed as a baseline level for non-fasting humans, and therefore useful for comparisons at the 24- and 72-hour time points as well.

Question 8: The correlation analysis shown in Fig 3A likely reflects some form of associations between different lipids as part of their lipoprotein status.

Response: This is an excellent point. We now state this possibility in the discussion section. The sentence reads as follows:

“The majority of circulating lipids are complexed with lipoproteins. The various classes of lipoproteins complexes vary in lipid composition; therefore, it is likely that the classes of lipoprotein complexes also vary over time after severe injury (Christinat and Masoodi, 2017).”

43. Christinat, N. & Masoodi, M. Comprehensive lipoprotein characterization using lipidomics analysis of human plasma. *J. Proteome Res.* **16**, 2947–2953 (2017).

Question 9: “Causal” vs “Casual” in the context of inference analysis and modeling. Please check spelling.

Response: We corrected the typo in the manuscript. Thank you.

Reviewer #2:

The studies provide data extensive data on circulating lipids in the plasma in patients who suffered major trauma, with an effort to identify patterns that would provide prognostic information. Some of the data was also applied to severe COVID-19 patients. While there is some merit in this study, there are several limitations. In the primary cohort, patients were randomized to receive FFP 2 units or standard of care, and as reported before, the FFP-treated patients had increased survival.

Comment 1: This is a major descriptive effort to try to identify patterns of changes in lipid levels that would add to our understanding of how trauma causes organ injury and death.

Comment 2: There were 996 lipids measured with TG the most abundant followed by PE, PC and DAG. Most lipids declined in following trauma.

Comment 3: Figure 2 shows some patterns of lipid changes depending on the patient outcomes with an unexplained increase in four lipid classes at 72 hours in patients that died or remained critically ill. here was no impact of the standard Injury Severity Score on lipid patterns.

Comment 4: The FFP-treated patients had a rise in lipid profile toward normal than dissipated at 24 and 72 hours.

Comment 5: Lower lipid levels seem to correlate with several know pro-inflammatory cytokines and coagulation factors (Figure 7s).

Comment 6: The authors try to make sense of all of this data by then putting it into probabalistic graphical models in an algorithm in Figure 4G. The findings seem to identify potentially causal factors including lipid concentrations, coagulation levels, crystalloid administration plus levels of pro-inflammatory factors.

Comment 7: The authors then try to validate these findings in other trauma data sets. The primary pattern of decline in lipid levels after trauma was identified. They then generated a Lipid Reprogramming Score (LRS) using 8 PE species that were common to the each data set. Figure 6 shows how this worked out in analyzing relationships to clinical course and outcomes. The LRS was then correlated with markers of inflammation and endothelial injury.

Response to comments 1-7: We concur with the reviewer's summary and conclusions.

Question 8: Then correlated the LRS with COVID-19 patients. This analysis seems especially weak and under-developed and tacked on - would probably need to be part of a separate report.

Response: We appreciate the reviewer's point of view. Admittedly, our goal for this analysis was rather limited; could we find evidence for similar patterns in circulating lipids in another form of acute critical illness? Since these data came from published datasets, and are limited by the information provided within these manuscripts, it is unlikely that we would be able to extend the analysis beyond what we have provided. In addition, we point out that the changes in PE were not correlated with outcomes in the original Covid-19 publications. Finally, we note that Reviewer #3 and Reviewer #4 found the inclusion of the Covid-19 findings a useful addition to our paper. We are very open to making changes as directed by the editors.

Question 9: Having worked through all the results and figures, I do not feel enlightened for pathogenesis of trauma induced injury and the prognostic signals seem so complex to make it difficult to see how they could be used well. The manuscript is long and somewhat tedious and the end result is not illuminating. It reads more as a compendium of findings without clear significance.

Response: We apologize that the reviewer found our paper a challenge to read. This is the first study to report, on a large scale, the circulating lipidomic changes in severely injured humans. Therefore, we felt it important to describe the major findings to provide the first full "landscape-type" description. This revealed dramatic changes that have not been seen before. Many of these changes correlated with outcomes and an intervention (early thawed plasma) that prevented mortality. This allowed us to make inferences and conduct causal modeling. This resulted in a picture where the sudden loss of circulating lipids associates with adverse outcomes. Preventing this drop in circulating lipids using early plasma provides mechanistic insights in a real world setting. The selective increase in a subset of lipids in patients with a slow recovery provides yet another novel finding. This work sets a new baseline for our understanding of the human response to severe injury and will guide future work in this important area of human medicine.

To address the readability, we have removed one of the main figures (Figure 3) and placed this in the supplemental section and only briefly mention this analysis of the relationships between lipid species. We also simplify the description for generating the Lipid Reprogramming Score in the results section. Instead, we provide more details of the steps taken to generate the LRS in the methods section. This allows us to focus the manuscript on the clinical relationships of the data.

Content related to Figure3:

Original:

“To better visualize the changes in individual lipid species, we created a correlation network of 412 lipids shown to differ between the resolving and non-resolving patients at 72h (**Fig 3A**). Only highly correlated relationships between each connected lipid pair in the correlation network (Pearson correlation coefficient $r > 0.7$) were kept. Lipids within each class were well correlated with each other. Furthermore, we identified a unique relationship for the inter-class networks. The dominant type of lipids that increased from baseline in non-resolving patients were from the DAG-TAG and PE classes (**Fig 3A**). DAG and PE are produced in the liver and kidney by the conversion of the same precursors (fatty acid-CoA and L-glycerol-3-phosphate), first to phosphatidic acid and then either DAG or PE. PE and other glycerophospholipids are generated by the addition of headgroups (e.g. ethanolamine for PE or choline for PC) while TAG is synthesized from DAG by the addition of a third acyl group by acyl transferase. Also evident from the figure is the suppression of the cholesterol (CE) and LPE families of lipids. The interconnections between biochemical pathways involved in the synthesis of the lipid classes are shown in **Fig 3B**. The pathways are color coded to show how these pathways relate to the changes in lipid levels in the non-resolving group.”

After modification:

“Interestingly, there was a high level of intra- and inter-correlation between the elevated lipid classes, including TAG\DAG\PE (Fig S3A-B). The interconnections between biochemical pathways involved in the synthesis of the lipid classes are shown in **Fig S3C**.”

Content related to generation of LRS:

Original:

To quantify the changes in lipids associated with critical illness in trauma and COVID-19 patients, we used eight PE species common to all four datasets to generate a Lipid Reprogramming Score (LRS) (Fig 6A). Three independent methods were used to define the relationship between the LRS and global lipidomic patterns and outcomes. First, a comparison between non-resolving and resolving trauma patients using logistical regression with Age, ISS, and treatment as co-variables yielded a ranking of lipids detected in PAMPer dataset (Table S3). The eight PE species ranked at ranking at 3, 41, 63, 109, 110, 142, 206, and 294 respectively (Volcano plot shown in Fig S6A). In addition, we found that 27 lipids belonging to TAG class of lipids and 7 additional PE lipids were significantly higher in non-resolving patients at 72h (adjusted $p < 0.01$, $\log \text{foldchange} > 0.4$). This differential analysis also yielded three LPC that were significantly lower. Next, we constructed a matrix that correlated the initial eight PE in the starting pool with these 37 differentially expressed lipids (Fig S6B). The starting PE were correlated positively with several other PE and 27 TAG, and negatively correlated with the three lower LPC species. This indicates that the eight PE common to

all four datasets may also be representative of an overall reprogramming that includes upregulation of TAG release and a suppression of LPC release into the circulation. We generated a LRS represented as a mean z-score for each patient across all three timepoints.

After modification:

We next sought to determine if a combination of PE species common to the four trauma and COVID-19 patient datasets could be optimized to generate a Lipid Reprogramming Score (LRS) based on PE species found to be elevated in the four datasets. (Fig 5A, see also methods). Briefly, eight PE species detected across four datasets were selected as the starting pool. All the eight PE species were highly correlated with the 37 other lipids (mostly TAG species, Table S3) identified as significant higher in non-resolving PAMPer patients (72h) by logistic regression taking into account cofounders, including ISS, age, and treatment (Fig S6A-B). The sensitivity analysis identified the model of 5 species showing the best performance (Table S6). Thus, we defined the Lipid Reprogramming Score (LRS) as the mean z-score of five PE species (PE(16:0/18:2), PE(16:0/20:4), PE(16:0/22:6), PE(18:0/18:1), PE(18:0/22:6)) in each datasets for subsequent analysis.

We have also worked to improve the readability throughout the manuscript.

Question 10: The authors conclude that they have provided a new paradigm for lipid response to severe and acute stress. But I do not find that I can really understand tangibly this new paradigm except in very broad strokes. Why does it have no relationship to the ISS or clinical outcomes in a more tangible way?

Response: We are pleased to address this central question in the following way. The reason for the limited relationship to ISS is that the majority of the PAMPer patients fell into the high ISS range. A study that included a representation across injury severities might be expected to show injury-specific patterns (i.e., the impact of severity, shock, injury patterns). As far as tangible paradigm change, the dramatic drop in circulating lipids associated with early outcomes (mortality and slow resolution) is a new paradigm that should be incorporated into models of the human acute response to severe injury (similar to the Cytokine Storm). That the early administration of plasma reduces mortality while mitigating the drop in glycerophospholipids provides new insights into the effects of early plasma in trauma. Finally, the selective increases in specific glycerophospholipids in patients that remain critically ill sheds new light into a biologic process initiated early in persistent critical illness. Importantly, these longitudinal findings are all made in severely injured humans from a rigorous multi-institutional study. To link these descriptive findings to outcomes and processes, we

carried out state-of-the-art causal modeling and developed a Lipid Reprogramming Score that correlates with outcomes.

Comment 11: The authors have acknowledged the limitations of their study well in the second to last paragraph of the discussion.

Response: Thank you.

Reviewer #3:

Wu et al provided an impressive study on lipidomic signatures in patients with critical illness provide a new paradigm for the lipid response to a severe and acute 407 systemic stress leading to critical illness.

Despite the complicated methodology, the work looks appropriate, and the authors had the brilliant idea of including a cohort of COVID-19 patients.

Response to general comments: We are thankful for the reviewer's positive feedback.

Question 1: Can the authors provide some more rational on the pathophysiological role of lipids in inflammation, maybe even with a supplementary material included in the introduction? I am aware this is unusual, however this paper can be a practice changing one in the middle term and revolutionize the field of inflammatory biomarkers, which has impact on a wide range of specialists (most of us would not have a basic knowledge on the topic).

Response: We agree that we could improve the background information in the introduction. We have added the following two sentences. Line(76-78) "Many lipids can serve as regulators of inflammation and immune responses (Duffney et al., 2018). In addition, certain lipids (e.g. triacylglycerides, fatty acid et al.) can serve as essential energy substrates for certain immune cell subsets (e.g. Memory T cell\Treg\M2 Macrophage)((Bantug et al., 2018; Kedia-Mehta and Finlay, 2019))."

Duffney, P. F. et al. Key roles for lipid mediators in the adaptive immune response. *J. Clin. Invest.* 128, 2724–2731 (2018).

Kedia-Mehta, N., and Finlay, D.K. (2019). Competition for nutrients and its role in controlling immune responses. *Nat. Commun.* 10, 2123.

Bantug, G.R., Galluzzi, L., Kroemer, G., and Hess, C. (2018). The spectrum of T cell metabolism in health and disease. *Nat. Rev. Immunol.* 18, 19–34.

Question 2: Do the authors have basic lipid profiles of these cohorts (cholesterol, LDL, HDL, and so on?) does these values may be influenced by baseline "body composition" or even by drugs which may interfere with lipid metabolism? If there is a rationale, the authors may provide these data (including BMI) or, if not possible, this should be mentioned as a limitation

Response: We agree that clinical lipid panels and BMI values would be of great interest in these patients. Unfortunately, the clinical lipid panels are not routinely measured in the emergency

department for severely injured trauma patients. We now include this as a limitation to the analysis (line:398-399).

BMI values were available on 89 patients who survive beyond 72h. We explored the relationship between the BMI and early lipid levels and LRS (shown below) in this subset. There was only a weak relationship between BMI and total lipid levels ($r=0.18$, $p=0.094$). The LRS was independent of BMI ($r=-0.03$, $p=0.73$), suggesting that the lipid reprogramming process may not be influenced by baseline body composition. We now mention this analysis in the paper and provide the new figures in Supplemental Figure S6D-E.

Question 3: The authors may provide in the discussion some perspective for future fields of study. In particular, I am fascinating about the rapid changes in profiles, which may be particularly useful in a subset of patients that mostly suffer from acute/hyperacute conditions which we often have difficulties for early diagnosis (I mostly think at sepsis in young children, infants and neonates, for which we do not yet have proper biomarkers)

Response: Thank you for this helpful guidance. We have added several sentences in the discussion section for the future direction. These read as follows, “In the future, circulating lipid profiles may be useful for prognostication or for guiding early interventions with plasma or other strategies to replace specific lipid deficiencies in trauma or in critical illness from other etiologies. The LRS score might

be used to predict which patients will follow a more complicated course following trauma or as the result of other causes of acute critical illness, such as sepsis”.

Reviewer #4:

This piece of research aims to tackle the problem of accuracy predicting outcomes of patients suffering with trauma and SARS CoV-2 (COVID-19) infection using a derived Lipid Reprogramming Score (LRS). This would prove useful especially in our current environment in which health care facilities across the globe are tacking severe COVID infection. In this regard, this piece of scientific research, from my point of view, aims to add knowledge to existing literature at this relevant time. That said, as I am not an expert on either COVID-19, Critical Care (trauma treatment) or lipid biology, I cannot comment on the clinical relevance of the data used. Therefore, I will only provide feedback related to the statistical methodology used in the paper.

As I already mentioned above, I think this piece of work has potential to contribute in helping tackle an important problem. The strengths of the paper is that the authors come up with the LRS that would be beneficial in making prognostic predictions for trauma and COVID-19 patients. Further to this, the authors have made an attempt to externally validate the use of LDS using real world data (TD-2).

However, I do find that the execution of the work to be limited in the following ways. First, there are too many analyses and figures for the readers to go through and make sense of the results contained in the paper. My impression is that the authors want to include all the results that they deem to be interesting results in the paper. However, by doing this, they (may) inadvertently make it difficult for the reader to follow what is being presented. I think it much better to conduct only the analyses that support the main theme of the paper. For example, it may be that some correlation plots / exploratory graphs may not be needed in passing the intended message. Second, I think somehow related to the first point, since there are many analyses, some were not conducted to the required standard. Having fewer analyses would enable greater depth. For example, not sufficient detail is provided about the performance measures of the internal and external validation as required for prediction models. Third, not sufficient justification was provided for the predictors in the model (specifically, were the predictors selected because of their clinical utility or was a variable model selection procedure utilised), sample sizes (not easy to tell in some instances) and distribution of the outcome and predictors were not provided for the each of the statistical model so I was not able to fully appraise them.

Response to general comments: We grateful for the reviewer's efforts in rigorously reviewing our analysis. By responding to the reviewer's comments, we have made important corrections to the paper and clarified many of the analysis.

To address the readability, we have removed one of the main figures (Figure 3) and placed this in the supplemental section and only briefly mention this analysis of the relationships between lipid species. We also simply the description for generating Lipid Reprogramming Score in results section. Instead, we provide more details in method section. This allows us to focus the manuscript on the clinical relationships of the data.

Content related to Figure3:

Original:

“To better visualize the changes in individual lipid species, we created a correlation network of 412 lipids shown to differ between the resolving and non-resolving patients at 72h (Fig 3A). Only highly correlated relationships between each connected lipid pair in the correlation network (Pearson correlation coefficient $r > 0.7$) were kept. Lipids within each class were well correlated with each other. Furthermore, we identified a unique relationship for the inter-class networks. The dominant type of lipids that increased from baseline in non-resolving patients were from the DAG-TAG and PE classes (Fig 3A). DAG and PE are produced in the liver and kidney by the conversion of the same precursors (fatty acid-CoA and L-glycerol-3-phosphate), first to phosphatidic acid and then either DAG or PE. PE and other glycerophospholipids are generated by the addition of headgroups (e.g. ethanolamine for PE or choline for PC) while TAG is synthesized from DAG by the addition of a third acyl group by acyl transferase. Also evident from the figure is the suppression of the cholesterol (CE) and LPE families of lipids. The interconnections between biochemical pathways involved in the synthesis of the lipid classes are shown in Fig 3B. The pathways are color coded to show how these pathways relate to the changes in lipid levels in the non-resolving group.”

After modification:

“Interestingly, there was a high level of intra- and inter-correlation between the elevated lipid classes, including TAG\DAG\PE (Fig S3A-B). The interconnections between biochemical pathways involved in the synthesis of the lipid classes are shown in **Fig S3C**.”

Content related to generation of LRS:

Original:

“To quantify the changes in lipids associated with critical illness in trauma and COVID-19 patients, we used eight PE species common to all four datasets to generate a Lipid Reprogramming

Score (LRS) (Fig 6A). Three independent methods were used to define the relationship between the LRS and global lipidomic patterns and outcomes. First, a comparison between non-resolving and resolving trauma patients using logistical regression with Age, ISS, and treatment as co-variables yielded a ranking of lipids detected in PAMPer dataset (Table S3). The eight PE species ranked at ranking at 3, 41, 63, 109, 110, 142, 206, and 294 respectively (Volcano plot shown in Fig S6A). In addition, we found that 27 lipids belonging to TAG class of lipids and 7 additional PE lipids were significantly higher in non-resolving patients at 72h (adjusted $p < 0.01$, $\log \text{foldchange} > 0.4$). This differential analysis also yielded three LPC that were significantly lower. Next, we constructed a matrix that correlated the initial eight PE in the starting pool with these 37 differentially expressed lipids (Fig S6B). The starting PE were correlated positively with several other PE and 27 TAG, and negatively correlated with the three lower LPC species. This indicates that the eight PE common to all four datasets may also be representative of an overall reprogramming that includes upregulation of TAG release and a suppression of LPC release into the circulation. We generated a LRS represented as a mean z-score for each patient across all three timepoints.”

After modification:

“We next sought to determine if a combination of PE species common to the four trauma and COVID-19 patient datasets could be optimized to generate a Lipid Reprogramming Score (LRS) based on PE species found to be elevated in the four datasets. (Fig 5A, see also methods). Briefly, eight PE species detected across four datasets were selected as the starting pool. All the eight PE species were highly correlated with the 37 other lipids (mostly TAG species, Table S3) identified as significant higher in non-resolving PAMPer patients (72h) by logistic regression taking into account cofounders, including ISS, age, and treatment (Fig S6A-B). The sensitivity analysis identified the model of 5 species showing the best performance (Table S6). Thus, we defined the Lipid Reprogramming Score (LRS) as the mean z-score of five PE species (PE(16:0/18:2), PE(16:0/20:4), PE(16:0/22:6), PE(18:0/18:1), PE(18:0/22:6)) in each datasets for subsequent analysis.”

We have also worked to improve the readability throughout the manuscript.

A. Minor:

Question 1: The reference for the PAMPer trial should be numbered (18) and not (19).

Response: We have corrected the citation.

Question 2: Line 103 says that there were non-survivors (n=72) and survivors (n=121), a total of 193 patients selected for lipidome analysis. However, Fig S1, contradicts this by showing that the 193 selected consisted of non-survivors (n=83) and survivors (n=110).

Response: We correct the typo in the manuscript. There were 83 non-survivors and 110 survivors.

Question 3: What was the distribution (in Fig S1) of the non-survivors (plasma / soc)? How were they distributed across the different sites in the original data?

Response: We attach a table for the distribution of non-surviving patients across different sites. We do not find any significant difference for the distribution of patients in our sampling process.

	Standard Care (Sampling/Total)	Plasma (Sampling/Total)	p-value(chisq.test)
Site1	11/20	5/13	0.567
Site2	20/22	5/8	0.196
Site3	4/6	2/4	1
Site4	5/11	3/6	1
Site5	14/22	9/14	1
Site6	3/8	2/8	1

Question 4: Please replace all references of “logistical regression” with “logistic regression”. For example, see lines 192,528,539, 541, 543 etc

Response: Thank you for this correction. We have made the change.

Question 5: The statement in line 539 should be re-written to explicitly communicate how survival and severity was categorical (so as not to leave any room for misinterpretation).

Response: We add sentences to explain the definition for survival and severity. These read,” For trauma patients, non-survival refers to death within 72h and severity, described as non-resolution, was defined by an ICU LOS of \geq to 7d or non-survival at timepoints beyond 72h.”

For COVID-19 patients, we defined severity categories using the World Health Organization (WHO) Clinical Score.

WHO Working Group on the Clinical Characterisation and Management of COVID-19 infection (2020). A minimal common outcome measure set for COVID-19 clinical research. *Lancet Infect. Dis.* 20, e192–e197.

Question 6: It is not clear to me what is mean in line 540 by “only main effect of each factor was evaluated”.

Response: We deleted the sentence to avoid misunderstandings.

Question 7: In line 543, I think “across” should be replaced with “for each”.

Response: We made the recommended change.

Question 8: In line 440/441, “trail” should be replaced with “trial”.

Response: We correct the typo in the manuscript.

Question 9: Line 425, “analyze” should read “analyzed”.

Response: We correct the typo in the manuscript.

Question 10: In line 170, was there a (clinical) justification for categorising ISS? If this was done arbitrarily or for convenience, I think this should be made clear.

Response: There is no accepted standard for categorizing the ISS, however many authors (including our group in the past) have defined ISS as mild (<15), moderate (15-24) and severe (>24) as these relate to mortality in trauma patients.

B. Major:

Question 11: For the statistical model fitted in the PAMPer trial, the clustered nature of the data (sites) was accounted for using a hierarchical model using ICC (0.05), I have not seen how the clustered nature of the data used for this study was accounted for in the analysis. This may have type 1 error (false positive) rate implications.

Response: This is a very helpful point. The raw logistical regression model (Fig 3E) is now replaced by a generalized estimating equation model for adjusting cluster effect. The raw cox regression model (Fig 6D) has been replaced by a mixed cox regression model for adjusting cluster effect. The results remain consistent with the original analysis after the correction.

Fig 3E Forest plot showing odds ratios from logistic regression of clinical factors; lipid concentration; thawed plasma effect for early-nonsurvivors versus others (left: Original logistical model, right: generalized estimating equation model).

Fig 6D(New) Forest plot showing hazard ratio of clinical factors and LRS score for recovery using a Cox regression model (left: original cox regression model, right: mixed effect model).

Question 12: How were the 15 control patients (line 106) chosen? Were they matched to ensure valid comparison with the other 193 patients? Table 1 does not provide any information about them. This should be reported, for example, Guo et. al report the controls they used and explain how they were matched.

Response: The non-fasting healthy subjects (age>18) were randomly selected from a established healthy cohort. In our preliminary analysis, we did not find a strong effect of age\gender\BMI on the lipid profile. Thus, we did not apply a strict inclusion criterion for the healthy subjects. We have now modified Table 1 and added more information for the healthy subjects. There was no significant

difference in the distribution of age ($p=0.836$) and gender ($p=0.839$) when we compared the group of healthy subjects and trauma patients.

Table 1: Demographic characteristics of the patients by outcome

Variables	Healthy Subject (N=17)	Resolving (N=41)	Non-resolving (N=101)	Early-Nonsurvivors (N=51)	p-value
Demographics					
Age (Median [IQR])	38 (\pm 31)	48 (\pm 34)	46 (\pm 37)	46 (\pm 42)	0.836 [#]
Sex (% Male)	12 (70.6%)	31 (75.6%)	78 (77.2%)	36 (70.6%)	0.809 [#]
Race (% White)		35 (85.4%)	89 (88.1%)	48 (94.1%)	0.365
Injury					
ISS (Median [IQR])		21 (\pm 10)	30 (\pm 16)	24 (\pm 23)	<0.001
Head AIS (Median)		0 (\pm 3.0)	3.0 (\pm 2.0)	3.0 (\pm 4.0)	<0.001
TBI (%)		14 (34.1%)	66 (65.3%)	29 (56.9%)	0.003
GCS (Median [IQR])		14 (\pm 7.0)	3.0 (\pm 9.0)	3.0 (\pm 8.0)	<0.001
SBP<70mmHg (%)		19 (46.3%)	41 (40.6%)	25 (49.0%)	0.580
HR (Median [IQR])		120 (\pm 16)	120 (\pm 21)	120 (\pm 39)	0.218
Injury type (%)		30 (73.2%)	93 (92.1%)	47 (92.2%)	0.017
Prehospital					
Treatment arm					
Standard care		25 (61.0%)	48 (47.5%)	36 (70.6%)	0.021
FFP (%)		16 (39.0%)	53 (52.5%)	15 (29.4%)	
Transport time		39 (\pm 18)	44 (\pm 17)	42 (\pm 18)	0.771
CPR (%)		0 (0%)	3 (2.97%)	5 (9.80%)	0.044
Intubation (%)		13 (31.7%)	65 (64.4%)	40 (78.4%)	<0.001
Blood (%)		11 (26.8%)	32 (31.7%)	22 (43.1%)	0.214
Crystalloid		800 (\pm 1400)	830 (\pm 1300)	1000 (\pm 1600)	0.891
PRBC (Median)		0 (\pm 1.0)	0 (\pm 1.0)	0 (\pm 2.0)	0.233
Hospital					
Transfusion 24h		2.0 (\pm 8.0)	7.0 (\pm 14)	12 (\pm 20)	<0.001
PRBC 24h (Median)		2.0 (\pm 5.0)	5.0 (\pm 7.0)	8.0 (\pm 10)	<0.001
Plasma 24h		0 (\pm 0)	2.0 (\pm 4.0)	4.0 (\pm 8.0)	<0.001
Platelets 24h		0 (\pm 0)	0 (\pm 1.0)	1.0 (\pm 2.0)	0.002
Crystalloid 24h		4800 (\pm 3800)	5300 (\pm 4000)	4600 (\pm 3000)	0.095
Vasopressors 24h		19 (46.3%)	68 (67.3%)	44 (86.3%)	<0.001
INR (Median [IQR])		1.2 (\pm 0.20)	1.3 (\pm 0.36)	1.6 (\pm 0.72)	<0.001
Other outcomes					
Coagulopathy (%)		16 (39.0%)	54 (53.5%)	44 (86.3%)	<0.001
ALI (%)		2 (4.88%)	47 (46.5%)	3 (5.88%)	<0.001
NI (%)		3 (7.32%)	43 (42.6%)	\	<0.001

Pearson's χ^2 test was used for calculating p value of categorical variables. Kruskal-Wallis test was used for calculating p value of continuous variables. ISS, injury severity score; AIS, abbreviated injury score; TBI, traumatic brain injury; GCS,

Glasgow coma score; SBP, systolic blood pressure; HR, heart rate; FFP, fresh frozen plasma; CPR, cardiopulmonary resuscitation; PRBC, packed red blood cells; INR, international normalized ratio; ALI, acute lung injury; NI, nosocomial infection; MOF, multiple organ failure; ICU, intensive care unit; LOS, length of stay.

Test was conducted across both healthy subjects and three outcome groups of trauma patients.

Question 13: I would recommend that the exploratory data analysis for the predictors and outcome used for each model be conducted. As far as I can tell, Table 1 gives information on all the 193 patients. For reader interested in determining sample size for each analysis, it is not easy to decipher. For example, for the logistic model of early death (early-non survivors), I reckon data for 51 patients were analysed. However, I am having a difficult time telling the sample size for the model analysing COVID-19 data (from Guo et. al – was the entire dataset on which their paper was based used? Or was a subset used? This should be clear in the paper).

Response: We thank the reviewer for these helpful comments. The selection for the predictors is based on our previous knowledge on the factors that may influence the patient outcome. We add a description for the sample size for each model in the figure legends and results section.

Question 14: The authors include references to missing data in line 472, can I confirm that the percentage in Table 1 do not include missing values. In other words, do we have any missing values for the variables? I suspect no missingness based on presentation of the table.

Response: Correct. Table 1 does not include variables with missing values. However, others variables (history of drug, lipids) indeed contain missing values. We exclude all the missing values in the history information and impute the lipids concentration for statistical analysis.

Question 15: Prognostic models, need to be trained and tested using the same variables. Is there a justification for developing the survival model (time to discharge by ICU) using the variables reported in 545/6 but externally validating the model on TD-2 data (line 548) that contains all but one variable used in the internal validation? As I understand this is not typically done.

Response: We thank the reviewer for identifying this important point. We want to assure that the readers understand that the cox regression model was performed to identify the association between LRS and outcome and not to create a prognostic model. We acknowledge that there is systemic bias between the PAMPer trial and TD-2 datasets. As we mentioned in the manuscript, the TD-2 dataset is dominated by the resolving patients with limited non-survivors. On the contrary, the PAMPer trial is dominated by non-resolving patients with many non-survivors. We now remove any reference to

prognostication in our modeling analysis. We also refer to the TD-2 dataset as an external validation model.

Question 16: How were the variables included in each model fitted selected? Was this based on clinical utility or a variable selection procedure?

Response: The selection for the predictors is based on our previous knowledge (from the literature) of the factors that may influence trauma patient outcomes. For trauma, we use the variables age, sex, ISS, severe head injury, plasma treatment arms and lipid levels for the model.

Question 17: How was the prognostic /prediction model developed evaluated? Prognostic models need to be internally validated based on their (i) Clinical utility [1], (ii) Discrimination (iii) and Calibration. The above measures can be corrected for bias and optimism using resampling methods (bootstrap or k-fold cross validation). Without these performance measures, it is difficult to objectively conclude the usefulness of the prognostic model used. Please see this papers as well.

Response: Thank you for this valuable guidance. Our goal for using the cox regression model was to identify the association between LRS (based on 5 PE lipid species) and patient outcome (and not prognostication). In fact, the discrimination and calibration analysis (Regular cox regression model without considering arms effect since TD-2 dataset did not contain this variable) for both inner validation (bootstrapping with n=60) and external validation showed poor performance (see below). Therefore, we are now cautious not to imply that we are developing prognostic models.

	Training	Internal validation	External validation
C Index	0.670	0.651	0.619

Table Performance of Discrimination in internal validation and external validation of Cox regression model.

Figure Performance of Calibration in external validation of Cox regression model.

Ms. No.: NCOMMS-20-46047-T

Title: Lipidomic Signatures Align with Inflammatory Patterns and Outcomes in Critical Illness

Summary of the major concerns and general response

We greatly appreciate the reviewers' efforts to review our manuscript and for their insightful comments to improve our paper. By addressing the concerns raised by the reviewers, we believe the paper has been substantially improved. Our revisions include two major re-analysis of the data based on the valuable comments of the reviewers. This included an analysis to establish the prognostic significance of our Lipid Reprogramming Score (LRS) and lipids in general in trauma and COVID-19. We also significantly re-wrote sections of the results to improve the readability and uploaded the raw and processed data from the lipidomic analysis to Mendeley Data (<https://data.mendeley.com/datasets/7stf7dtxcz/draft?a=3e078e7f-5068-4b8e-a5a9-ef414db279bd>, Reserved DOI: doi:10.17632/7stf7dtxcz.1) and in the supplementary materials (Supplementary Dataset 1). These major changes include the following:

1. We conducted a sensitivity analysis for the generation of the Lipid Reprogramming Score (LRS) to identify the phosphatidylethanolamines (PE) most representative of the two trauma and two COVID-19 datasets. Briefly, a set of five PE that included PE from all four datasets yielded a LRS that was highly associated with adverse outcomes (critical illness) in both trauma and COVID-19 patients. We have also now carried out a new and complete analysis of the prognostic value of both the LRS and a multivariate individual lipid signature in trauma and COVID-19. Central to our analyses is a two-step machine learning approach based on feature selection (using L1 regularization i.e., LASSO to avoid overfitting) followed by classification using the down-selected features (lipids). This approach takes into account the overall correlation structure of the high-dimensional lipidomic dataset and selects a minimal set of predictive lipid biomarkers. The reference model used for trauma was the Injury Severity Score (ISS) + IL6 and the reference model used for COVID-19 was lymphocyte count + C-reactive protein (CRP) levels. In both disease processes, the inclusion of the LRS or a single phosphatidylethanolamine (PE) specie (Trauma: PE (18:0/1:1), Covid-19: PE (16:0/22:6)) improved upon the prognostic value over the reference models.
2. We adopted new statistical models to account for cluster effect based on standard logistic and cox regression models. This corrected a key limitation of the previous analysis without altering the conclusions of study.

3. To increase the readability, we removed one of the main figures (Figure 3) and placed this in the supplemental section and only briefly mention this analysis of the relationships between lipid species in the results. We also significantly simplified the description of the generation of the Lipid Reprogramming Score in the results section. We now have placed a more detailed description of the steps used to develop the score in the Methods section. We also removed a tangential analysis using the COVID-19 databases from the manuscript involving a correlation of the LRS with proteomics from COVID-19.
4. We have added considerable detail on the technical aspects of the targeted lipidomic analysis platform (Metabolon Inc.) and uploaded the Raw data to Mendeley Data (<https://data.mendeley.com/datasets/7stf7dtxcz/draft?a=3e078e7f-5068-4b8e-a5a9-ef414db279bd>, Reserved DOI: doi:10.17632/7stf7dtxcz.1) and in the supplementary materials (Supplementary Dataset 1).
5. The plasma given to patients en route to the hospital in the treatment arm is now referred to as thawed Plasma (TP) instead of Fresh frozen plasma (FFP). This is more accurate because the plasma had been thawed prior to being placed on the helicopters. Therefore, the reviewers will see that FFP has been changed to TP throughout the manuscript.

Point-by-point Responses to the Reviewers' Comments

Reviewer #1:

In the manuscript by Wu et al, the authors propose the use of a lipidomic approach to assess the status of acute critical illnesses and the introduction of a Lipid Reprogramming Score (LRS) as a prognostic tool to evaluate progression towards severe conditions. The relevance of this work is high, in principle, as it introduces the use of quantitative molecular readouts to understand the physiological processes underlying insurgence of a critical condition.

The authors made important efforts in terms of sample and data analysis. However, the manuscript as it stands need to be revised in a major way. It is hoped that the following comments will be helpful. Please note that this review is focussed and limited to the lipidomic aspects of this study rather than trauma medicine or general data analysis and statistics. Additional reviewers should be consulted for the latter two areas.

Response to general comments: We sincerely thank the reviewer for carefully reviewing our manuscript and for the suggestions to improve the paper. The point-by-point responses are as follows:

Major areas of concern:

1) Lipidomic Methodology and reporting of raw data

Question 1: Line 113: “In the quality control analysis, the median relative standard deviation (RSD) for the lipid panel was 4%”. How was the quality control sample generated and how often was it analysed during the experiment? What were the RSD values for the 900 lipids? What was the RSD threshold for lipid selection?

Response: We are pleased to provide this important information. The QC sample was generated by combining a small aliquot from entire set of samples into a single pooled CMTX (Sample matrix). Four aliquots of the CMTX were run on each plate of 36 samples. One each was injected at the beginning and end of the run, with the other two roughly evenly spaced in between the remaining samples. A similar process was carried out for three process blanks (water) and MTRX7 (a well characterized plasma sample Metabolon runs on every plate). Unfortunately, Metabolon does not report on the RSD values for each biochemical, only the median value. There is no threshold for lipid selection (or any biochemical selection in the untargeted dataset) based on RSD. Metabolon calculated the median RSD from all species detected in 100% of the CMTX samples. We have now added this information in the methods section.

Question 2: Line 464: "Lipid species were background-subtracted using concentrations detected In process blanks..." what was the signal to noise (S/N) threshold for acceptance of results?

Response: Background levels were estimated/calculated from the median levels of the three process blanks (water) if there were detectable levels in at least 2 of the 3 blanks in each batch. The background level was subtracted from each sample in the batch prior to any run day normalization. If the value was negative, it was set to zero or “not detected”. Metabolon explained to us that after extensive analysis, they have found that implementing this step gives much better data quality for some lipid species that can be found on plastic tubes or other artifactual sources in their system. We have now added the approach to subtract background levels to the methods section.

Question 3: The nomenclature of all lipids reported is based on sn-1 or -2 location of fatty acids (PE 16:0/18:2); this might not be possible with the used setup unless the authors can explain how the position of the fatty acids was verified. An alternative possible nomenclature

for the same species might be according to PE 16:0_18:2. Please see Shorthand notation for lipid structures derived from mass spectrometry. *J. Lipid Res.* 2013. 54: 1523–1530.

Response: Unfortunately, the complex lipid panel (CLP) performed by Metabolon does not distinguish between SN1 and SN2 location of the side chains. The nomenclature used simply lists the 2 sidechains present without attempting to ascribe which sidechain resides at which position. We have now added this information to the methods section.

Question 4: Line 472: "Lipid with over 80% missing values were discarded..." did you mean over 20%?

Response: Thank you for pointing out the typo. The corrected version states, "*Lipids with over 20% missing values were discarded*". We attached a histogram plot for the distribution of missing lipids. Based on the exploratory analysis, we kept all lipid species with over 80% non-missing values for the subsequent analysis.

Question 5: No raw lipidomic data is reported.

Response: We have uploaded the raw and processed lipidomic datasets into Mendeley Data (<https://data.mendeley.com/datasets/7stf7dtxcz/draft?a=3e078e7f-5068-4b8e-a5a9-ef414db279bd>) and in the supplementary materials. This information is now available to the reviewers and would also be available to the readers of our paper.

Question 6: While 900 lipids are measured, close to 2/3 of the lipidomic panel covers glycerolipids. No lipid mediators were included which are more closely linked to physiological stress induced by imbalances in redox biology or inflammation. What is the reason for this? In any case, it should be discussed at least.

Response: We agree that levels of lipid mediators would be of considerable interest. However, largely for technical reasons, Metabolon does not include mediators in their CLP platform. Since the CLP is technically a targeted assay, they have chosen to detect the 14 lipid classes shown in Fig. 1B. We now add to the discussion that the absence of lipid mediators as a limitation of the study.

Question 7: Since there is an analytical bias towards non-polar lipids, have the authors measured these compounds also via more traditional clinical lipid panels? How about cholesterol levels?

Response: This is an excellent suggestion. Unfortunately, the clinical laboratory lipid panels are not routinely measured in trauma patients. In fact, we could not find an instance where lipids were measured at the time of admission in our patient cohorts.

Question 8: The healthy subject baseline group is very small (HC, n=17). Furthermore, the authors acknowledge the differences in sampling between the HC and trauma groups. PE is a strong function of daily nutrition and sampling (e.g. Sweeney, G., Nazir, D., Clarke, C. & Goettsche, G. Ethanolamine and choline phospholipids in nascent very-low-density lipoprotein particles. *Clin Invest Med* 19, 243–250 (1996)). Could this be relevant for the observed differences between HC and trauma patients?

Response: Thank you for raising these important points. The control samples were intentionally drawn from non-fasting healthy subjects to act as suitable controls for the admission time point for the trauma patients (designated as the 0h timepoint). While we had only 17 control subjects, a recent report (Tarazona et al., 2020) concludes that the minimal desirable sample size for omics (i.e. Metabolomics/Proteomics) is around 16. This number was shown to yield statistical power of 0.6-0.8 for differential analysis. In addition, we show that the heterogeneity among healthy subjects is much lower than trauma patients in the UMAP plot (Fig2A). Overall, while we agree that more healthy controls may have been preferable, we found that the number of healthy subjects was adequate to perform the downstream analysis.

As for the difference of PE, we are confident that the late increases in PE species is not influenced by nutritional sources since none of these critically ill trauma patients received nutritional support over the first 3 days of hospitalization. It is noteworthy that the subset of trauma patients that exhibit an increase in PE at 72 h are those that remain critically ill longer than those that do not have an increase in PE. The increases in PE in these patients is even higher than the healthy controls. Thus, the higher circulating PE levels mostly likely represents changes in endogenous processes that determine levels of circulating PE. We speculate that this is part of lipid reprogramming that takes

place in the setting of persistent critical illness and one of the novel biological insights provided by our analysis.

2) Meta-analysis of COVID-19 datasets

This aspect of the manuscript is not convincing. It is certainly desirable to extend the primary experimental research findings to more generalisable insight. But the attempt to do this in the context of COVID-19 falls short.

Response to general comments: Thank you for raising these important concerns over our analysis in COVID-19 datasets. Our point-by-point responses are as follows:

Question 1: **First, as mentioned in the introduction, a major limitation in studying critical illness is the lack of knowledge in precise onset of the process. This is in fact the case for COVID-19 but not the trauma cohorts described here.**

Response: This is an excellent point and one rational for why trauma can serve as an excellent derivation dataset for other critical illnesses (i.e., Severe COVID-19). While we certainly expect there to be major differences in critical illness caused by trauma vs. other etiologies, there are also likely to be human responses that are in common during acute critical illnesses. We were intrigued to see that critically ill Covid-19 patients also exhibit a rise in PE. While much more work is needed to understand the basis of this common response, we are very cautious not to over-interpret these findings. We have further modified our description to state (line 236 to 237), *“Unlike trauma, the onset of critical illness in Covid-19 patients can be highly variable relative to the onset of infection and the time the infection started is often unclear. To assist with the comparison between the trauma and COVID-19 datasets, we set the 0 timepoint in the COVID-19 datasets as the day of symptom onset for non-severe patients or day of progression for severe patients.”*

Question 2: **The utilised analytical (metabolomic and lipidomic technologies) differ considerably between the 4 studies compared here (e.g. untargeted vs targeted analysis, without and with chromatography, calibrators for quantification, etc). Therefore, direct comparison of results will require more detailed and systematic cross-comparison which in addition to the biological metabolite variations need to take into account also the associated technical variabilities of the different studies. Accordingly, line 234 reads “Eight lipids from PE class... four PC or PI were higher in the non-resolving trauma patients (72h) or severely ill**

COVID-19 patients in at least one dataset (out of 4 in total). This seems like a very low bar for inclusion.

Response: We appreciate the reviewer’s concerns about the differences in the techniques used across the four datasets to measure lipid levels. Here, we kept only lipids species from the four databases that had the same ID from the Human Metabolome Database (<https://hmdb.ca/>). To address the variability in the techniques used across the four datasets we applied only indirect comparisons. We carried out the indirect comparison by scaling and centering all the lipids within each dataset to identify the lipids within each dataset that were different between controls and patients with trauma or Covid-19. This allowed us to identify common lipids that were elevated across all the datasets. We now acknowledge in the results, “*Because there were differences in the methods used to quantify lipid species across the datasets, we only carried out indirect comparisons of the relative changes (Z-scores) of lipids species within each dataset across the datasets.*” (lines 227-229)

In order to address the reviewer’s concern about the representation and inclusion of the PE species used to generate the Lipid Reprogramming Score (LRS) within the four datasets, we have re-done this analysis. This led to the identification of 5 PE species elevated in common in 3 out of 4 of the datasets as the optimal set of PE species to calculate the LRS. However, the 5 PE species also includes at least one PE species from all four datasets.

We first summarize the number of PE species elevated in common with the PAMPer database when including one, two or all three of the datasets in addition to the PAMPer dataset. It can be seen in the table below (**Table S5**) that by including two datasets (PAMPer + any 1 other dataset) 8 PE species identified in the PAMPer dataset can be found in common. Increasing the number of datasets to three (PAMPer + any 2 other datasets) the number of PE species in common drops to 5 out of 8. Only one PE species (PE(18:0/18:1)) was found to be in common to all 4 datasets.

	At least in PAMPer	At least in PAMPer and at least (>=1) in other 3 datasets	At least in PAMPer and at least (>=2) in other 3 datasets	At least in PAMPer and at least (>=3) in other 3 datasets
Numbers of lipids Species	9	8	5	1
Name	PE(16:0/18:2),PE(16:0/20:4), PE(16:0/22:6), PE(18:0/18:1), PE(18:0/18:2), PE(18:0/20:4), PE(18:0/22:6), PE(18:1/18:2), PC(18:0/18:1)	PE(16:0/18:2),PE(16:0/20:4), PE(16:0/22:6), PE(18:0/18:1), PE(18:0/18:2), PE(18:0/20:4), PE(18:0/22:6), PE(18:1/18:2)	PE(16:0/18:2),PE(16:0/20:4), PE(16:0/22:6), PE(18:0/18:1), PE(18:0/22:6)	PE(18:0/18:1)

We then conducted a sensitivity analysis by using 8 PE species (PAMPer + 1 other dataset), 5 PE species (PAMPer + 2 other datasets) or 1 PE specie (PAMPer + all 3 other datasets) to generate the LRS (**Table S5**). While the LRS based on 5 PE or 8 PE species yielded high correlations with outcomes based on Pearson correlation coefficients ($r=0.97-0.98$), the r values dropped ($r=0.78-0.86$)

when using the single PE common to all four datasets. The LRS (5 species) was associated with improved performance in COVID dataset (Guo et al.) compared to LRS (8 species) (OR 9.98 vs 4.67). In addition, the performance (OR in PAMPer : 0.73 vs 0.69, OR in TD-2 : 0.73 vs 0.77) was comparable in Trauma datasets. This indicates that the PE species have a high degree of correlation with each other and that using a minimum of 5 PE species, all in common within 3 out of 4 datasets, should lead to the most stable model. We then re-performed all the analysis using the five PE species LRS calculation for the correlation analyses and indeed found better results. Thus, we replaced all our original analysis (LRS from 8 species) with the new analysis (LRS from 5 species). This resulted in adjustments to figures 6-7 and supplementary figures 6-7.

We modified the relevant sentence in the manuscript to read:

Lines (242-248):

“We identified eight PE species and one PC specie significantly higher in the non-resolving group (72h) in PAMPer dataset, while three, six, and five of these PE were elevated in the TD-2, Covid-19 (Guo et al)¹⁶, and Covid-19 (Shui et al)¹⁷ datasets, respectively. Eight of these PE species could be identified when combining PAMPer with any single other database, five PE species were in common when combining PAMPer with any two of the other databases, and a single PE was found to be significantly elevated during critical illness in common across all four databases (Table S5). Thus, increases in PE consistently associate with critical illness in trauma and COVID-19.”

Line (256-259):

“A sensitivity analysis identified a model comprised of five PE showing the best performance (Table S6). Thus, we defined the LRS as the mean z-score of five PE species (PE(16:0/18:2), PE(16:0/20:4), PE(16:0/22:6), PE(18:0/18:1), PE(18:0/22:6)) representative of PE from all four trauma and COVID-19 datasets.”

Table S6 Sensitivity analysis of LRS generation by different species of PE							
	r (PAMPer)	r (TD-2)	r (guo et al.)	r (shui et al.)	HR (PAMPer)	HR (TD-2)	OR (guo et al.)
LRS(8 Species)	ref.	ref.	ref.	ref.	0.69 (0.53-0.91)	0.73 (0.55-0.98)	4.67 (1.24-25.20)
LRS(5 Species)	0.98	0.98	0.97	0.97	0.73 (0.56-0.95)	0.77 (0.58-1.02)	9.98 (2.09-78.50)
LRS(1 Specie)	0.8	0.78	0.86	0.8	0.64 (0.49-0.83)	0.67 (0.51-0.88)	1.81 (0.80-4.80)

Pearson correlation coefficient was calculated between the LRS (5 species) or LRS (1 specie) and LRS (8 species) from **Table S5**.
 In PAMPer dataset, Hazard ratio for LRS was calculated by cox regression model with mixed effect and adjusting for Age, Gender, ISS, Head Injury and Arms. In TD-2 dataset, Hazard ratio for LRS was calculated by standard cox regression model and adjusting for Age, Gender, ISS, Head Injury. In dataset of guo et al., odd ratio for LRS was calculated by logistic regression model and adjusting for Age, Gender, Lymphocyte count and CRP.

By addressing the reviewer’s concern, we have optimized the formulation of the LRS and developed a LRS more representative of critical illness in trauma or Covid-19.

Question 3: Validation is not based on independent cohorts interrogated experimentally. One would need prospective trials to support the claims for the prognostic panel.

Response: We thank the reviewer for making this important point to the novel insights provided by our findings. We also agree that prospective cohorts should be used to establish the prognostic value of lipid levels. Based on the reviewer's comment and comment #17 from reviewer 4, we have now carried out an extensive analysis of the prognostic value of the LRS and the five individual PE species that comprise the LRS in trauma and COVID-19. To do this, we used our two prospective trauma cohorts (PAMPer trial and TD-2 datasets) and a prospectively collected COVID-19 dataset from Guo et al.

For trauma, we set a model using the injury severity score (ISS) and IL6, two variables known to correlate with mortality and complications, as the reference model. For COVID-19, we set a model based on lymphocyte count (Lym) and CRP levels as the reference model as both have been shown to correlate with disease severity in COVID-19 patients. We then calculated the increase in the AUC resulting from adding the LRS or each of the five PE species that comprise the LRS (Please see figure below) to the model. We found that LRS alone moderately improved the AUC (increase in AUC=0.018 of PAMPer dataset, increase in AUC=0.036 of Covid-19 dataset of Guo et al.) in trauma and COVID-19. We also found that a single PE that was specific to the disease process improved the AUC to an even greater extent than the LRS. For trauma this was PE(18:0/18:1)(increase in AUC=0.075) and for COVID-19 this was PE(16:0/22:6)(increase in AUC=0.062).

Figure S7A. The increase in AUC from the inclusion of the LRS or its components to the reference model in the training set. (Trauma: 73 patients in standard-of-care arm of the PAMPer dataset, Covid-19: 45 patients in training cohort (C1) from Guo et al.)

We further utilized an established two-step machine learning approach to identify a minimal set of predictive lipid biomarkers and clinical features for predicting the outcome in the PAMPer dataset (standard-of-care arm). This approach was based on feature selection (L1 Regularization) followed by classification (Support Vector Machine) using the down-selected features. The results suggest that PE (18:0/18:1) was the top selected feature among lipids and only IL6 and ISS ranked higher overall (See Figure S7B). Taken together, this indicated that a single PE well-represented in the datasets may have the highest prognostic value after taking into account the overall correlation structure of the lipid-omic dataset.

Figure S7B. Frequency of top10 feature selected by *two-step machine learning approach* in PAMPer dataset (73 patients in standard-of-care arm of PAMPer dataset, see also methods).

We then developed a prognostic model for predicting the recovery pattern of trauma patients (resolving vs. non-resolving). We set the standard-of-care arm in PAMPer study as training dataset. The plasma arm was set as an internal test set. The TD-2 dataset was set as an external test set. Three logistic regression models with 10-fold cross validation were developed in the training set and tested in the two test sets. The performance of discrimination was assessed by ROC curves and AUC values. The performance of calibration was assessed using a calibration curve and by calculating the brier score. The prediction was underestimated when true probability is around 0.25-0.5 in the reference model (ISS+IL6) but performed well at probabilities above 0.5.

Figure S7D. Calibration curve of three prognostic models in 73 patients from standard-of-care arm of the PAMPer dataset.

Addition of either LRS or PE (18:0/18:1) to ISS+IL-6 improved the prediction accuracy (left panel, AUC: ISS+IL6, 0.798; ISS+IL6+LRS, 0.816; ISS+IL6+PE (18:0/18:1), 0.873). The brier score was also lower with the inclusion of LRS or PE (18:0/18:1) levels (ISS+IL6:0.177, ISS+IL6+LRS:0.166, ISS+IL6+PE (18:0/18:1):0.139). The overall discrimination performance was improved in the training set and the two test sets (Middle panel, AUC in test set1: ISS+IL6, 0.876; ISS+IL6+LRS, 0.916; ISS+IL6+PE (18:0/18:1), 0.900, Right panel, AUC in test set2: AUC: ISS+IL6, 0.797; ISS+IL6+LRS, 0.814; ISS+IL6+PE (18:0/18:1), 0.841).

Figure 5D & S7EF. ROC curves for three prognostic models in the training dataset (left), internal test dataset (middle) and external test dataset (right).

Similarly, we adopted the two-step machine learning approach to identify a minimal set of predictive lipid biomarkers and clinical features for predicting the outcome in the Covid-19 dataset (C1 of Guo et al.). The results suggested that PE (16:0/22:6) was the top selected feature among lipids. The clinical feature of lymphocyte count and CRP were ranked at 6th and 7th respectively.

Figure S7C. Frequency of top10 feature selected by two-step machine learning approach in Covid-19 dataset (45 patients in C1 of Guo et al, see also methods).

We also developed a reference prognostic model (Lym+CRP) for predicting disease progression in COVID-19 patients (non-severe vs. severe). Here, LRS or PE (16:0/22:6) as single variables were used for prediction since there was no additional value observed by adding these to Lym + CRP levels. We set the cohort 1 and 2 in dataset of Guo et al. as the training set and test set, respectively. The reference model was trained via logistic regression with 10-fold cross validation. The model PE (16:0/22:6) showed the best performance of calibration compared to other two models (Brier Score: Lym+CRP:0.177; LRS: 0.166; PE (16:0/22:6): 0.139.).

Figure S7G. Calibration curve of three prognostic models in 45 patients from the Covid-19 dataset (C1 of Guo et al.)

The performance of discrimination was comparable between the reference model and LRS in both training and test datasets. However, the performance of PE (16:0/22:6) alone was better than the reference model (AUC 0.917 vs. 0.958). Interestingly, we noticed that only one patient (XG43) was mis-labeled when we used PE (16:0/22:6) levels in test-set. This is comparable to the random forest (17 proteins+9 metabolites) model from the original manuscript. Thus, we present a much-simplified model based on a single molecule for predicting the outcome of Covid-19.

Figure 5G&S7HI. ROC curves of three prognostic models in training set (left), test set (middle). Right: Performance of PE (16:0/22:6) in the test set of 10 COVID-19 patients.

All metrics for performance of discrimination and calibration can be found in Table S7(see below).

Table S7 Performance of prognostic value for LRS and individual PE species in Trauma and COVID-19				
	Discrimination (AUC Value)	Added AUC value	Calibration (Brier Score)	Decreased Brier Score
Trauma				
Train set (Standard Arm)				
ISS+IL6	0.798(0.696-0.901)	Ref	0.177	Ref
ISS+IL6+LRS	0.816(0.719-0.913)	0.018	0.166	-0.011
ISS+IL6+PE(18:0/18:1)	0.873(0.795-0.952)	0.075	0.139	-0.038
Internal test set (Plasma Arm)				
ISS+IL6	0.876(0.787-0.966)	Ref	0.126	Ref
ISS+IL6+LRS	0.916(0.845-0.988)	0.040	0.116	-0.010
ISS+IL6+PE(18:0/18:1)	0.900(0.821-0.978)	0.024	0.136	0.010
External test set (TD-2 Dataset)				
ISS+IL6	0.797(0.703-0.891)	Ref	0.225	Ref
ISS+IL6+LRS	0.814(0.721-0.907)	0.017	0.214	-0.011
ISS+IL6+PE(18:0/18:1)	0.841(0.750-0.932)	0.044	0.198	-0.027
COVID-19				
Train set (C1 of guo et al.)				
Lym+CRP	0.786 (0.631–0.941)	Ref	0.196	Ref
LRS	0.814 (0.684–0.944)	0.028	0.174	-0.022
PE(16:0/22:6)	0.862 (0.753–0.971)	0.076	0.150	-0.046
Test set (C2 of guo et al.)				
Lym+CRP	0.917 (0.728–1.000)	Ref	0.123	Ref
LRS	0.833 (0.557–1.000)	-0.084	0.150	0.027
PE(16:0/22:6)	0.958 (0.843–1.000)	0.045	0.075	-0.048
Logistic regression model with 10-fold cross-validation was used in train set. The same model was used to predict in the validation set.				
The performance of the model was evaluated by AUC value and brier score. 95% confidence interval was calculated for AUC value.				
ISS: Injury Severity Score; LRS: Lipid Reprogramming Score.				

This analysis has been added to the manuscript along with the following text:

Results:

Line 286 to 303

We next explored the prognostic value of the LRS and the five individual PE species that comprise the LRS for predicting whether trauma patients would progress to a non-resolving pattern (Table S7). Here, we set the standard-of-care arm in PAMPer dataset as the training set (n=73). The TP arm from PAMPer dataset was set as an internal test set (n=69) and the TD-2 dataset was set as an external test set (n=86). Compared to the reference model(Raymond et al., 2020) (ISS+IL6, AUC=0.798), adding the LRS moderately improved the performance of discrimination (AUC=0.816, added AUC =0.018) in the training set (Fig 5D, sFig 7A). Interestingly, of the five PE that comprise the LRS, PE (18:0/18:1) also greatly improved the performance of discrimination (AUC=0.873, added AUC =0.075) in the training set (Fig 5D, sFig 7A). We further utilized an established two-step machine learning approach (Ackerman et al., Nature Medicine 2018; Suscovich et al Science Translational Medicine 2020; Das et al PLoS Pathogens 2020) to identify a minimal set of predictive lipid biomarkers and clinical features for predicting the outcome in the PAMPer dataset (standard-of-care arm). This approach was based on feature selection (L1 Regularization – LASSO to avoid overfitting) followed by classification (Support Vector Machine) using the down-selected features. The results suggested that PE (18:0/18:1) was the top selected feature among lipids and it is only

behind IL6 and ISS ranked higher overall (**sFig 7B**). The performance of calibration (**sFig 7C**) was also improved by adding either the LRS or the single PE (18:0/18:1) to ISS+IL6 (Brier Score: ISS+IL6, 0.177; ISS+IL6+LRS, 0.166; ISS+IL6+PE (18:0/18:1), 0.139). The results were consistent in the internal (**sFig 7D**, AUC: ISS+IL6, 0.876; ISS+IL6+LRS, 0.916; ISS+IL6+PE (18:0/18:1), 0.900) and external test sets (**sFig 7E**, AUC: ISS+IL6, 0.797; ISS+IL6+LRS, 0.814; ISS+IL6+PE (18:0/18:1), 0.841).

Line 312 to 326

C-reactive protein (CRP) and lymphocyte count (Lym) are known to correlate with worse outcomes in COVID-19 patients (Wynants et al., 2020). We compared the LRS and its five individual PE species with these two variables to classify severe versus non-severe patients in both a training set (C1, n=45) and test set (C2, n=10) (**Table S7**). We found that the LRS alone moderately improved the performance of discrimination (AUC=0.814, added AUC=0.028), however a single PE specie from the LRS (PE (16:0/22:6)) alone greatly improved the performance (AUC=0.862, added AUC=0.076) (**Fig5G**, **sFig 7A**). The two-step machine learning approach also revealed that the PE (16:0/22:6) were the top selected features (**sFig7C**). The performance of calibration was also improved by using PE (16:22:6) (Brier Score: Lym+CRP:0.177; LRS: 0.166; PE (16:0/22:6): 0.139; **sFig7G**). The performance of PE (16:0/22:6) also had the highest AUC compared to other two models in the test set (**sFig 7F**, AUC: CRP+Lym, 0.917; LRS, 0.833, PE (16:0/22:6), 0.958). Interestingly, we also noticed that PE (16:0/22:6) showed similar performance for prognostication as the random forest model based on 17 proteins and 9 metabolites reported in the original manuscript. It is notable that only one patient (XG43) was mislabeled in the test cohort (C1) using PE (16:0/22:6) for prognostication (**sFig 7G**). Thus, similar to our observations in trauma, a single PE specie derived from the LRS performed well of prognostication for severe disease in COVID-19.

Methods:

Line 611-616

Multi-variable regression analysis

For prognostic model generation, we built logistic regression models for predicting patients who developed non-resolving pattern in the trauma cohorts (Training set: standard-of-care arm in PAMPer dataset, Test set: plasma arm in PAMPer dataset or TD-2 dataset) or who progressed into severe COVID-19 in the cohort of Guo et al (Train:C1, Test:C2). All models were internally evaluated by 10-fold cross-validation. The performance of discrimination and calibration were

assessed by ROC curve, and AUC values. Performance of calibration was assessed calibration curves, and brier scores respectively.

Line 618-627

A two-step machine learning approach

The importance of clinical features and lipids commonly detected in trauma and Covid-19 datasets were evaluated by two-step feature selection strategy as previously reported (Ackerman et al., Nature Medicine 2018; Suscovich et al Science Translational Medicine 2020; Das et al PLoS Pathogens 2020). Briefly, two models (least absolute shrinkage and selection operator: lasso; support vector machine with the radial basis function: SVM-RBF) were fitted sequentially with 100 times repeated and nested 5-fold cross-validation (Both outer and inner sampling were 5 folds). The lasso model is fitted into inner sampling of each 5-fold and the hyperparameter were tuned by the performance of classification (accuracy). Then a fold-specific classifier was trained by SVM-RBF in the selected features from the lasso model in the same fold. The performance (accuracy) of the fold-specific classifier was internally evaluated in the outer-sampling. The frequency of selected features was summarized in all models with accuracy >0.8. Top10 selected features were kept for visualization.

3) Generalized interpretation

Question 1: The last sentence of the discussion should elaborate more on the possible use of these data and the LRS, especially because a single marker measured in the clinic, CRP, can have a better predictive value. The same is true for clinical lipid measurements which might provide equally informative results (see also above).

Response:

Thank you for this excellent point. We have added to the last sentence to read (line 410-412) “*The LRS or individual PE species (PE (18:0/18:1), PE (16:0/22:6)), like other biomarkers of the host response (e.g., IL6 and CRP), may be useful as parameters for outcome prediction linked to specific biologic processes.*”

For the reviewer’s interest, we attach a comparative analysis addressing the association of the LRS and total lipid concentration with outcomes (see below). Our finding suggests that the LRS is an independent risk factor for outcome, while there was no significant association between total lipid concentration and outcome. Thus, it may be not be possible to use the currently available clinical lipid blood tests for outcome prediction.

Question 2: Could other examples of critical illness be mentioned for future validation of this lipid score, beyond COVID-19?

Response: We are pleased to mention other examples of causes of critical illness (e.g., sepsis and burns) that should be examined in future validation work. Specifically, we state (line 412-414), “It will also be of interest to determine if the LRS predicts outcomes in other etiologies of critical illness such as sepsis and burns.”

Question 3: The results presented in Figure 4G are interesting. However, it is not clear why lipid metabolism should be at the center and so closely linked to early death.

Response: Thank you for raising this important point. The plot presents a causal network among all the factors included in the causal modelling. The position for each variable is not intended to signify its potential importance relative to other variables in the model. A direct line indicates a potential causal relationship between any two variables. Here, we show that the lipid concentration may be an

independent factor related to death. The connection between prehospital administration of plasma and lipid levels implicates a causal relationship between lipid levels and pre-hospital plasma administration. We positioned plasma treatment and lipid levels to highlight the possibility that early plasma may improve outcomes, in part, through the preservation of lipid levels. We have made it clear in the description of the modelling how to interpret the connections in the diagram.

4) Other comments:

Question 1: End of line 378: reference cited should be 38 and not 39

Response: Thank you for pointing out the typo and the citation error. Since we added several citations to the introduction section in the revised version, the sentence now reads as follows:

“Several specific lipid species [e.g. PC(16:0/18:1), PC(18:0/18:1)] are known to contribute to inter-organ (liver, muscle and adipose tissue) communication⁴³.”

The reference is listed as follows:

“43. Liu, S., Alexander, R. K. & Lee, C.-H. Lipid metabolites as metabolic messengers in inter-organ communication. *Trends Endocrinol. Metab.* **25**, 356–363 (2014).”

Question 2: Line 375: glycolipids? Why?

Response: Thank you for pointing out the error. We changed glycolipids to glycerolipids when discussing the increases in TAG and DAG.

Question 3: Figure 7C is not clear in some parts: lipolysis and lipogenesis should not be on the y axis to avoid confusion; the lipids increasing in the right part of the figure are apparently coming from few classes only.

Response: Thank you for the helpful observation. We now make it clear that the Y-axis reflects the directional changes in lipids (all drop early and a few lipid species increase late in a subset of patients). We also color code the patient trajectories. Below we show the original figure 7C and also our new modified version.

Question 4: Line 336: a persistent lowering of circulating lipids through 72 hours...I don't think this is what is shown in figure 7C. Could the authors please check?

Response: We agree that this sentence could have been better stated. We have revised as follows, "A drop in circulating lipids persisted through 72h in patients destined to resolve their critical illness earlier." We have also modified Figure 7C to make the diagram easier to interpret this point.

Question 5: How was the fatty acid data displayed in Figure 2 calculated from the lipidomic measurements?

Response: The fatty acid reflects the sum of the concentrations for all species that contained the designated fatty acid. For example, the levels of FA 20:4 reflects the sum of the concentrations for all lipid species that contain FA 20:4.

Question 6: The main commonality is major reduction of overall lipid concentrations upon trauma.

Response: We concur. This is observed within an hour or two after major trauma, so clearly happens very fast. Importantly, this has not been shown before and, indeed, is one of the major biologically relevant discoveries of the study. We emphasize this point in several sections of the manuscript.

Question 7: Are HC the most relevant reference? A more appropriate reference condition might be patients after the they recovered.

Response: We agree that having the right controls is essential to the interpretation. In theory, the best controls would be the patients themselves prior to injury. For obvious reasons, this is not possible. The reviewer is correct that a baseline once the patient has recovered would also serve as an excellent control. Unfortunately, 30% of these patients die within 72 hours making this option impossible for some of the most interesting patients. The remaining patients may not return to their true baseline for weeks to months due to the prolonged time it takes to recover from polytrauma, especially in patients with severe traumatic brain injury. The study was not approved for delayed sampling in the surviving patients.

This leaves us with healthy age- and sex-matched non-fasting subjects as the best available option for controls. We note that the HC are most relevant to the 0h time point data but can also be viewed as a baseline level for non-fasting humans, and therefore useful for comparisons at the 24- and 72-hour time points as well.

Question 8: The correlation analysis shown in Fig 3A likely reflects some form of associations between different lipids as part of their lipoprotein status.

Response: This is an excellent point. We now state this possibility in the discussion section. The sentence reads as follows:

Line 390-392:

“The majority of circulating lipids are complexed with lipoproteins. The various classes of lipoproteins complexes vary in lipid composition; therefore, it is likely that the classes of lipoprotein complexes also vary over time after severe injury (Christinat and Masoodi, 2017).”

43. Christinat, N. & Masoodi, M. Comprehensive lipoprotein characterization using lipidomics analysis of human plasma. *J. Proteome Res.* **16**, 2947–2953 (2017).

Question 9: “Causal” vs “Casual” in the context of inference analysis and modeling. Please check spelling.

Response: We corrected the typo in the manuscript. Thank you.

Reviewer #2:

The studies provide data extensive data on circulating lipids in the plasma in patients who suffered major trauma, with an effort to identify patterns that would provide prognostic information. Some of the data was also applied to severe COVID-19 patients. While there is some merit in this study, there are several limitations. In the primary cohort, patients were randomized to receive FFP 2 units or standard of care, and as reported before, the FFP-treated patients had increased survival.

Comment 1: This is a major descriptive effort to try to identify patterns of changes in lipid levels that would add to our understanding of how trauma causes organ injury and death.

Comment 2: There were 996 lipids measured with TG the most abundant followed by PE, PC and DAG. Most lipids declined in following trauma.

Comment 3: Figure 2 shows some patterns of lipid changes depending on the patient outcomes with an unexplained increase in four lipid classes at 72 hours in patients that died or remained critically ill. here was no impact of the standard Injury Severity Score on lipid patterns.

Comment 4: The FFP-treated patients had a rise in lipid profile toward normal than dissipated at 24 and 72 hours.

Comment 5: Lower lipid levels seem to correlate with several know pro-inflammatory cytokines and coagulation factors (Figure 7s).

Comment 6: The authors try to make sense of all of this data by then putting it into probabalistic graphical models in an algorithm in Figure 4G. The findings seem to identify potentially causal factors including lipid concentrations, coagulation levels, crystalloid administration plus levels of pro-inflammatory factors.

Comment 7: The authors then try to validate these findings in other trauma data sets. The primary pattern of decline in lipid levels after trauma was identified. They then generated a Lipid Reprogramming Score (LRS) using 8 PE species that were common to the each data set. Figure 6 shows how this worked out in analyzing relationships to clinical course and outcomes. The LRS was then correlated with markers of inflammation and endothelial injury.

Response to comments 1-7: We concur with the reviewer's summary and conclusions.

Question 8: Then correlated the LRS with COVID-19 patients. This analysis seems especially weak and under-developed and tacked on - would probably need to be part of a separate report.

Response: We appreciate the reviewer's point of view. Admittedly, our goal for this analysis was limited; could we find evidence for similar patterns in circulating lipids in another form of acute critical illness? Since these data came from published datasets and are limited by the information provided within these manuscripts, it is unlikely that we would be able to extend the analysis beyond what we have provided. In addition, we point out that the changes in PE were not correlated with outcomes in the original Covid-19 publications.

Based on the reviewer's concern and to simplify the manuscript, we have removed the analysis between the LRS and COVID-19 proteomic analysis as tangential to the main theme of the paper.

Finally, we note that Reviewer #3 and Reviewer #4 found the inclusion of the Covid-19 findings a useful addition to our paper. We are very open to making additional changes as directed by the editors.

Question 9: Having worked through all the results and figures, I do not feel enlightened for pathogenesis of trauma induced injury and the prognostic signals seem so complex to make it difficult to see how they could be used well. The manuscript is long and somewhat tedious and the end result is not illuminating. It reads more as a compendium of findings without clear significance.

Response: We apologize that the reviewer found our paper a challenge to read. This is the first study to report, on a large scale, the circulating lipidomic changes in severely injured humans. Therefore, we felt it important to describe the major findings to provide the first full "landscape-type" description. This revealed dramatic changes that have not been seen before. Many of these changes correlated with outcomes and an intervention (early thawed plasma) that prevented mortality. This allowed us to make inferences and conduct causal modeling. The causal modeling and the assessment of the prognostic value of the lipid programming score (LRS) each support the key biologic insights provided by the study. These include:

- 1) There is a sudden loss of circulating lipids that associates with adverse outcomes.
- 2) Preventing this drop in circulating lipids using early plasma (pre-hospital plasma-a novel approach to treat trauma patients) is associated with less mortality and provides potential mechanistic insights. Causal modeling was performed to establish a potential causal link between the drop in lipids and early death.
- 3) The selective increase in a subset of lipids in patients with a slow recovery provides yet another novel finding. Patients while in the fasting state and destined to remain critically ill show a selective early increase in phosphatidylethanolamines. We used this finding to

develop a Lipid Reprogramming Score (LRS) and show the prognostic value of a subset of phosphatidylethanolamines in both trauma and COVID-19 patients.

To address the concern about the readability, we made the following major changes:

- 1) We have removed main Figure 3 and placed this in the supplemental section and only briefly mention this analysis of the relationships between lipid species. This takes one of the more complicated and tangential analysis from the main analysis sequence.
- 2) We also removed the correlation analysis between the LRS and COVID-19 proteomic analysis as this was a tangential analysis not central to the overall story.
- 3) We also simplify the description for generating the Lipid Reprogramming Score in results section. Instead, we provide more details in method section. This allows us to focus the manuscript on the clinical relationships of the data.

This led to the following changes in the manuscript:

Content related to Figure 3:

Original:

“To better visualize the changes in individual lipid species, we created a correlation network of 412 lipids shown to differ between the resolving and non-resolving patients at 72h (**Fig 3A**). Only highly correlated relationships between each connected lipid pair in the correlation network (Pearson correlation coefficient $r > 0.7$) were kept. Lipids within each class were well correlated with each other. Furthermore, we identified a unique relationship for the inter-class networks. The dominant type of lipids that increased from baseline in non-resolving patients were from the DAG-TAG and PE classes (**Fig 3A**). DAG and PE are produced in the liver and kidney by the conversion of the same precursors (fatty acid-CoA and L-glycerol-3-phosphate), first to phosphatidic acid and then either DAG or PE. PE and other glycerophospholipids are generated by the addition of headgroups (e.g. ethanolamine for PE or choline for PC) while TAG is synthesized from DAG by the addition of a third acyl group by acyl transferase. Also evident from the figure is the suppression of the cholesterol (CE) and LPE families of lipids. The interconnections between biochemical pathways involved in the synthesis of the lipid classes are shown in **Fig 3B**. The pathways are color coded to show how these pathways relate to the changes in lipid levels in the non-resolving group.”

After modification:

“Interestingly, there was a high level of intra- and inter-correlation between the elevated lipid classes, including TAG\DAG\PE (Fig S3A-B). The interconnections between biochemical pathways involved in the synthesis of the lipid classes are shown in Fig S3C.”

Content related to generation of LRS:

Original:

To quantify the changes in lipids associated with critical illness in trauma and COVID-19 patients, we used eight PE species common to all four datasets to generate a Lipid Reprogramming Score (LRS) (Fig 6A). Three independent methods were used to define the relationship between the LRS and global lipidomic patterns and outcomes. First, a comparison between non-resolving and resolving trauma patients using logistical regression with Age, ISS, and treatment as co-variables yielded a ranking of lipids detected in PAMPer dataset (Table S3). The eight PE species ranked at ranking at 3, 41, 63, 109, 110, 142, 206, and 294 respectively (Volcano plot shown in Fig S6A). In addition, we found that 27 lipids belonging to TAG class of lipids and 7 additional PE lipids were significantly higher in non-resolving patients at 72h (adjusted $p < 0.01$, $\log \text{foldchange} > 0.4$). This differential analysis also yielded three LPC that were significantly lower. Next, we constructed a matrix that correlated the initial eight PE in the starting pool with these 37 differentially expressed lipids (Fig S6B). The starting PE were correlated positively with several other PE and 27 TAG, and negatively correlated with the three lower LPC species. This indicates that the eight PE common to all four datasets may also be representative of an overall reprogramming that includes upregulation of TAG release and a suppression of LPC release into the circulation. We generated a LRS represented as a mean z-score for each patient across all three timepoints.

After modification:

“We next sought to determine if a combination of PE species common to the four trauma and COVID-19 patient datasets could be optimized to generate a Lipid Reprogramming Score (LRS) based on PE species found to be elevated in the four datasets. (Fig 5A, see also methods). Briefly, eight PE species detected across four datasets were selected as the starting pool. All the eight PE species were highly correlated with the 37 other lipids (mostly TAG species, Table S3) identified as significant higher in non-resolving PAMPer patients (72h) by logistic regression taking into account cofounders, including ISS, age, and treatment (Fig S6A-B). The sensitivity analysis identified the model of 5 species showing the best performance (Table S6). Thus, we defined the Lipid Reprogramming Score (LRS) as the mean z-score of five PE species (PE(16:0/18:2), PE(16:0/20:4), PE(16:0/22:6), PE(18:0/18:1), PE(18:0/22:6)) in each datasets for subsequent analysis.”

Removed: “ Association between LRS and the proteome for COVID-19 patients”

“To further identify possible factors or pathways contributing to a pathologic lipidome signature, we correlated the LRS with circulating proteomic data from the COVID-19 study published by Guo, et al (Shen et al., 2020). Using 42 subjects with both metabolomic and proteomic data, we identified 141 proteins that correlated positively (spearman correlation coefficient $r > 0.3$) with the LRS (**Fig S8C**). Pathway enrichment analysis revealed that the LRS was associated with neutrophil degranulation, platelet degranulation, and the complement cascade (**Fig S8C and Fig S8E**). Negatively correlated (spearman correlation coefficient $r < -0.3$) proteins ($n=25$) were enriched in regulation of insulin-like growth factor-1 (IGF-1) transport and uptake, and post-translational protein phosphorylation (**Fig S8D and Fig S8F**). To further seek biological significance, we selected 40 representative proteins from the positive and negative correlating groups to construct a correlation matrix (**Fig 6B**). Components of the LRS were clustered in the module comprised acute phase proteins, the complement cascade, and immunoglobins, and were correlated negatively with modules associated with IGF-1. Our findings using data from COVID-19 patients suggests that excessive acute phase and immune responses and impaired metabolism associates with a pathologic circulating lipid signature across several causes of acute critical illness.”

We have also worked to improve the readability throughout the manuscript.

Question 10: The authors conclude that they have provided a new paradigm for lipid response to severe and acute stress. But I do not find that I can really understand tangibly this new paradigm except in very broad strokes. Why does it have no relationship to the ISS or clinical outcomes in a more tangible way?

Response: We are pleased to address this central question in the following way. The reason for the limited relationship to ISS is that the majority of the PAMPer patients fell into the high ISS range. A study that included a representation across injury severities might be expected to show injury-specific patterns (i.e., the impact of severity, shock, injury patterns). As far as tangible paradigm change, the dramatic drop in circulating lipids associated with early outcomes (mortality and slow resolution) is a new paradigm that should be incorporated into models of the human acute response to severe injury (similar to the Cytokine and Genomic Storm concepts). That the early administration of plasma reduces mortality while mitigating the drop in glycerophospholipids provides new insights into the effects of early plasma in trauma that associate with outcomes.

Finally, the selective increases in specific glycerophospholipids in patients that remain critically ill sheds new light into a biologic process initiated early in persistent critical illness. Importantly, these longitudinal findings are all made in severely injured humans from a rigorous multi-institutional study. To link these descriptive findings to outcomes and processes, we carried out state-of-the-art causal modeling and developed a Lipid Reprogramming Score that correlates with outcomes. We now also show the prognostic value of the Lipid Reprogramming Score and levels of individual phosphatidylethanolamine species in trauma and COVID-19.

Comment 11: The authors have acknowledged the limitations of their study well in the second to last paragraph of the discussion.

Response: Thank you.

Reviewer #3:

Wu et al provided an impressive study on lipidomic signatures in patients with critical illness provide a new paradigm for the lipid response to a severe and acute 407 systemic stress leading to critical illness.

Despite the complicated methodology, the work looks appropriate, and the authors had the brilliant idea of including a cohort of COVID-19 patients.

Response to general comments: We are thankful for the reviewer's positive feedback.

Question 1: Can the authors provide some more rational on the pathophysiological role of lipids in inflammation, maybe even with a supplementary material included in the introduction? I am aware this is unusual, however this paper can be a practice changing one in the middle term and revolutionize the field of inflammatory biomarkers, which has impact on a wide range of specialists (most of us would not have a basic knowledge on the topic).

Response: We agree that we could improve the background information in the introduction. We have added the following two sentences. Lines (79-81) "*Many lipids can serve as regulators of inflammation and immune responses (Duffney et al., 2018). In addition, certain lipids (e.g. triacylglycerides, fatty acid et al.) can serve as essential energy substrates for certain immune cell subsets (e.g. Memory T cell/Treg/M2 Macrophage)((Bantug et al., 2018; Kedia-Mehta and Finlay, 2019)).*"

Duffney, P. F. et al. Key roles for lipid mediators in the adaptive immune response. *J. Clin. Invest.* 128, 2724–2731 (2018).

Kedia-Mehta, N., and Finlay, D.K. (2019). Competition for nutrients and its role in controlling immune responses. *Nat. Commun.* 10, 2123.

Bantug, G.R., Galluzzi, L., Kroemer, G., and Hess, C. (2018). The spectrum of T cell metabolism in health and disease. *Nat. Rev. Immunol.* 18, 19–34.

Question 2: Do the authors have basic lipid profiles of these cohorts (cholesterol, LDL, HDL, and so on?) does these values may be influenced by baseline "body composition" or even by drugs which may interfere with lipid metabolism? If there is a rationale, the authors may provide these data (including BMI) or, if not possible, this should be mentioned as a limitation

Response: We agree that clinical lipid panels and BMI values would be of great interest in these patients. Unfortunately, clinical lipid panels are not routinely measured on admission or during the care of severely injured trauma patients. We now include this as a limitation to the analysis (line:424-425).

“Clinical lipid panels are not routinely measured in severely injured trauma patients. Therefore, we could not correlate changes in these commonly assessed lipids with our lipid panels.”

BMI values were available on 89 patients who survived beyond 72h. We explored the relationship between the BMI and early lipid levels and LRS (shown below) in this subset. There was only a weak relationship between BMI and total lipid levels ($r=0.18$, $p=0.094$). The Lipid Reprogramming Score was independent of BMI ($r= -0.03$, $p=0.73$), suggesting that the lipid reprogramming process may not be influenced by baseline body composition. We now add this analysis in the paper (line 268-270) and provide the new figures in Supplemental Figure S6D-E.

Question 3: The authors may provide in the discussion some perspective for future fields of study. In particular, I am fascinating about the rapid changes in prophyiles, which may be particularly useful in a subset of patients that mostly suffer from acute/hyperacute conditions which we often have difficulties for early diagnosis (I mostly think at sepsis in young children, infants and neonates, for which we do not yet have proper biomarkers)

Response: Thank you for this helpful guidance. We have added several sentences in the discussion section for the future direction. These read as follows:

(lines 379-381), *“In the future, circulating lipid profiles may be useful for prognostication or for guiding early interventions with plasma or other strategies to replace specific lipid deficiencies in trauma or in critical illness from other etiologies.*

lines 410-414

“Thus, the LRS or individual PE species (PE(18:0/18:1),PE(16:0/22:6)), like other biomarkers of the host response (e.g., IL6 and CRP), may be useful as early parameters for outcome prediction linked to specific biologic processes. It will also be of interest to determine if the LRS or PE species predicts outcomes in other etiologies of critical illness such as sepsis and burns.”.

Reviewer #4:

This piece of research aims to tackle the problem of accuracy predicting outcomes of patients suffering with trauma and SARS CoV-2 (COVID-19) infection using a derived Lipid Reprogramming Score (LRS). This would prove useful especially in our current environment in which health care facilities across the globe are tacking severe COVID infection. In this regard, this piece of scientific research, from my point of view, aims to add knowledge to existing literature at this relevant time. That said, as I am not an expert on either COVID-19, Critical Care (trauma treatment) or lipid biology, I cannot comment on the clinical relevance of the data used. Therefore, I will only provide feedback related to the statistical methodology used in the paper.

As I already mentioned above, I think this piece of work has potential to contribute in helping tackle an important problem. The strengths of the paper is that the authors come up with the LRS that would be beneficial in making prognostic predictions for trauma and COVID-19 patients. Further to this, the authors have made an attempt to externally validate the use of LDS using real world data (TD-2).

However, I do find that the execution of the work to be limited in the following ways. First, there are too many analyses and figures for the readers to go through and make sense of the results contained in the paper. My impression is that the authors want to include all the results that they deem to be interesting results in the paper. However, by doing this, they (may) inadvertently make it difficult for the reader to follow what is being presented. I think it much better to conduct only the analyses that support the main theme of the paper. For example, it may be that some correlation plots / exploratory graphs may not be needed in passing the intended message. Second, I think somehow related to the first point, since there are many analyses, some were not conducted to the required standard. Having fewer analyses would enable greater depth. For example, not sufficient detail is provided about the performance measures of the internal and external validation as required for prediction models. Third, not sufficient justification was provided for the predictors in the model (specifically, were the predictors selected because of their clinical utility or was a variable model selection procedure utilised), sample sizes (not easy to tell in some instances) and distribution of the outcome and predictors were not provided for the each of the statistical model so I was not able to fully appraise them.

Response to general comments: We grateful for the reviewer's efforts in rigorously reviewing our analysis. By responding to the reviewer's comments, we have made important corrections to the paper and clarified many of the analysis.

To address the concern about the number of analysis and figures, we made the following major changes:

1. We have removed main Figure 3 and placed this in the supplemental section and only briefly mention this analysis of the relationships between lipid species. This takes one of the more complicated and tangential analysis from the main analysis sequence.
2. We also removed the correlation analysis between the LRS and COVID-19 proteomic analysis as this was a tangential analysis not central to the overall story.
3. We also simplify the description for generating the Lipid Reprogramming Score in results section. Instead, we provide more details in method section. This allows us to focus the manuscript on the clinical relationships of the data.

Changes within the manuscript include:

Content related to Figure3:

Original:

“To better visualize the changes in individual lipid species, we created a correlation network of 412 lipids shown to differ between the resolving and non-resolving patients at 72h (Fig 3A). Only highly correlated relationships between each connected lipid pair in the correlation network (Pearson correlation coefficient $r > 0.7$) were kept. Lipids within each class were well correlated with each other. Furthermore, we identified a unique relationship for the inter-class networks. The dominant type of lipids that increased from baseline in non-resolving patients were from the DAG-TAG and PE classes (Fig 3A). DAG and PE are produced in the liver and kidney by the conversion of the same precursors (fatty acid-CoA and L-glycerol-3-phosphate), first to phosphatidic acid and then either DAG or PE. PE and other glycerophospholipids are generated by the addition of headgroups (e.g. ethanolamine for PE or choline for PC) while TAG is synthesized from DAG by the addition of a third acyl group by acyl transferase. Also evident from the figure is the suppression of the cholesterol (CE) and LPE families of lipids. The interconnections between biochemical pathways involved in the synthesis of the lipid classes are shown in Fig 3B. The pathways are color coded to show how these pathways relate to the changes in lipid levels in the non-resolving group.”

After modification:

“Interestingly, there was a high level of intra- and inter-correlation between the elevated lipid classes, including TAG\DAG\PE (Fig S3A-B). The interconnections between biochemical pathways involved in the synthesis of the lipid classes are shown in Fig S3C.”

Content related to generation of LRS:

Original:

“To quantify the changes in lipids associated with critical illness in trauma and COVID-19 patients, we used eight PE species common to all four datasets to generate a Lipid Reprogramming Score (LRS) (Fig 6A). Three independent methods were used to define the relationship between the LRS and global lipidomic patterns and outcomes. First, a comparison between non-resolving and resolving trauma patients using logistical regression with Age, ISS, and treatment as co-variables yielded a ranking of lipids detected in PAMPer dataset (Table S3). The eight PE species ranked at ranking at 3, 41, 63, 109, 110, 142, 206, and 294 respectively (Volcano plot shown in Fig S6A). In addition, we found that 27 lipids belonging to TAG class of lipids and 7 additional PE lipids were significantly higher in non-resolving patients at 72h (adjusted $p < 0.01$, $\log \text{foldchange} > 0.4$). This differential analysis also yielded three LPC that were significantly lower. Next, we constructed a matrix that correlated the initial eight PE in the starting pool with these 37 differentially expressed lipids (Fig S6B). The starting PE were correlated positively with several other PE and 27 TAG, and negatively correlated with the three lower LPC species. This indicates that the eight PE common to all four datasets may also be representative of an overall reprogramming that includes upregulation of TAG release and a suppression of LPC release into the circulation. We generated a LRS represented as a mean z-score for each patient across all three timepoints.”

After modification:

“We next sought to determine if a combination of PE species common to the four trauma and COVID-19 patient datasets could be optimized to generate a Lipid Reprogramming Score (LRS) based on PE species found to be elevated in the four datasets. (Fig 5A, see also methods). Briefly, eight PE species detected across four datasets were selected as the starting pool. All the eight PE species were highly correlated with the 37 other lipids (mostly TAG species, Table S3) identified as significant higher in non-resolving PAMPer patients (72h) by logistic regression taking into account cofounders, including ISS, age, and treatment (Fig S6A-B). The sensitivity analysis identified the model of 5 species showing the best performance (Table S6). Thus, we defined the Lipid Reprogramming Score (LRS) as the mean z-score of five PE species (PE(16:0/18:2), PE(16:0/20:4), PE(16:0/22:6), PE(18:0/18:1), PE(18:0/22:6)) in each datasets for subsequent analysis.”

Removed: “Association between LRS and the proteome for COVID-19 patients”

“To further identify possible factors or pathways contributing to a pathologic lipidome signature, we correlated the LRS with circulating proteomic data from the COVID-19 study published by Guo, et al¹⁶. Using 42 subjects with both metabolomic and proteomic data, we identified 141 proteins that correlated positively (spearman correlation coefficient $r > 0.3$) with the LRS (Fig S8C). Pathway enrichment analysis revealed that the LRS was associated with neutrophil degranulation, platelet degranulation, and the complement cascade (Fig S8C and Fig S8E). Negatively correlated (spearman correlation coefficient $r < -0.3$) proteins ($n=25$) were enriched in regulation of insulin-like growth factor-1 (IGF-1) transport and uptake, and post-translational protein phosphorylation (Fig S8D and Fig S8F). To further seek biological significance, we selected 40 representative proteins from the positive and negative correlating groups to construct a correlation matrix (Fig 6B). Components of the LRS were clustered in the module comprised acute phase proteins, the complement cascade, and immunoglobins, and were correlated negatively with modules associated with IGF-1. Our findings using data from COVID-19 patients suggests that excessive acute phase and immune responses and impaired metabolism associates with a pathologic circulating lipid signature across several causes of acute critical illness.”

We have also worked to improve the readability throughout the manuscript.

A. Minor:

Question 1: The reference for the PAMPer trial should be numbered (18) and not (19).

Response: We have corrected the citation.

Question 2: Line 103 says that there were non-survivors ($n=72$) and survivors ($n=121$), a total of 193 patients selected for lipidome analysis. However, Fig S1, contradicts this by showing that the 193 selected consisted of non-survivors ($n=83$) and survivors ($n=110$).

Response: We correct the typo in the manuscript. There were 83 non-survivors and 110 survivors.

Question 3: What was the distribution (in Fig S1) of the non-survivors (plasma / soc)? How were they distributed across the different sites in the original data?

Response: We attach a table for the distribution of non-surviving patients across different sites. We do not find any significant difference for the distribution of patients in our sampling process.

	Standard Care	Plasma	p-value(chisq.test)
--	---------------	--------	---------------------

	(Sampling/Total)	(Sampling/Total)	
Site1	11/20	5/13	0.567
Site2	20/22	5/8	0.196
Site3	4/6	2/4	1
Site4	5/11	3/6	1
Site5	14/22	9/14	1
Site6	3/8	2/8	1

Question 4: Please replace all references of “logistical regression” with “logistic regression”.
For example, see lines 192,528,539, 541, 543 etc

Response: Thank you for this correction. We have made the change.

Question 5: The statement in line 539 should be re-written to explicitly communicate how survival and severity was categorical (so as not to leave any room for misinterpretation).

Response: We add a sentence to explain the definition for survival and severity. These read (lines 599-602), “*For trauma patients, two outcomes Non-resolving (Survival with ICU stay ≥ 7 days or non-survival with death day >3 days) and Early-nonsurvivors (Non-survival with death day ≤ 3 days) were used. For COVID-19 patients, we defined severity categories using the World Health Organization (WHO) Clinical Score(WHO Working Group on the Clinical Characterisation and Management of COVID-19 infection, 2020).*”

WHO Working Group on the Clinical Characterization and Management of COVID-19 infection (2020). A minimal common outcome measure set for COVID-19 clinical research. *Lancet Infect. Dis.* 20, e192–e197.

Question 6: It is not clear to me what is mean in line 540 by “only main effect of each factor was evaluated”.

Response: We deleted the sentence to avoid any misunderstanding.

Question 7: In line 543, I think “across” should be replaced with “for each”.

Response: We made the recommended change.

Question 8: In line 440/441, “trail” should be replaced with “trial”.

Response: We correct the typo in the manuscript.

Question 9: Line 425, “analyze” should read “analyzed”.

Response: We correct the typo in the manuscript.

Question 10: In line 170, was there a (clinical) justification for categorising ISS? If this was done arbitrarily or for convenience, I think this should be made clear.

Response: There is no accepted standard for categorizing the ISS, however many authors (including our group in the past) have defined ISS as mild (<15), moderate (15-24) and severe (>24) as these relate to mortality in trauma patients. There is a great jump of mortality when ISS increases from 14(0.5%) to 16(4.83%) and from 24(3.09%) to 25(28.02%) (Rozenfeld et al., 2014).

Rozenfeld, M., Radomislensky, I., Freedman, L., Givon, A., Novikov, I., and Peleg, K. (2014). ISS groups: are we speaking the same language? *Inj. Prev.* 20, 330–335.

B. Major:

Question 11: For the statistical model fitted in the PAMPer trial, the clustered nature of the data (sites) was accounted for using a hierarchical model using ICC (0.05), I have not seen how the clustered nature of the data used for this study was accounted for in the analysis. This may have type 1 error (false positive) rate implications.

Response: This is a very helpful point. The raw logistical regression model (Fig 3E) is now replaced by a generalized estimating equation model for adjusting cluster effect. The raw cox regression model (Fig 6D) has been replaced by a mixed cox regression model for adjusting cluster effect. The results remain consistent with the original analysis after the correction.

Fig 3E. Forest plot showing odds ratios from logistic regression of clinical factors; lipid concentration; thawed plasma effect for early-nonsurvivors vs. others (left: Original logistical model, right: generalized estimating equation model).

Fig 6D (New). Forest plot showing hazard ratio of clinical factors and LRS for recovery using a Cox regression model (left: original cox regression model, right: mixed effect model).

Question 12: How were the 17 control patients (line 106) chosen? Were they matched to ensure valid comparison with the other 193 patients? Table 1 does not provide any information about them. This should be reported, for example, Guo et. al report the controls they used and explain how they were matched.

Response: The 17 non-fasting healthy subjects (age>18) were selected from an established healthy cohort to represent the age and sex distribution of the patient cohort. In our preliminary analysis, we did not find a strong effect of age\gender\BMI on the lipid profile. There was no significant difference in the distribution of age (p=0.836) or gender (p=0.839) when we compared the group of

healthy subjects and trauma patients. We have now modified Table 1 and added more information for the healthy subjects (Please see below).

Table 1: Demographic characteristics of the patients by outcome

Variables	Healthy Subject (N=17)	Resolving (N=41)	Non-resolving (N=101)	Early-Nonsurvivors (N=51)	p-value
Demographics					
Age (Median [IQR])	38 (± 31)	48 (± 34)	46 (± 37)	46 (± 42)	0.836 [#]
Sex (% Male)	12 (70.6%)	31 (75.6%)	78 (77.2%)	36 (70.6%)	0.809 [#]
Race (% White)		35 (85.4%)	89 (88.1%)	48 (94.1%)	0.365
Injury					
ISS (Median [IQR])		21 (± 10)	30 (± 16)	24 (± 23)	<0.001
Head AIS (Median)		0 (± 3.0)	3.0 (± 2.0)	3.0 (± 4.0)	<0.001
TBI (%)		14 (34.1%)	66 (65.3%)	29 (56.9%)	0.003
GCS (Median [IQR])		14 (± 7.0)	3.0 (± 9.0)	3.0 (± 8.0)	<0.001
SBP<70mmHg (%)		19 (46.3%)	41 (40.6%)	25 (49.0%)	0.580
HR (Median [IQR])		120 (± 16)	120 (± 21)	120 (± 39)	0.218
Injury type (%)		30 (73.2%)	93 (92.1%)	47 (92.2%)	0.017
Prehospital					
Treatment arm					
Standard care		25 (61.0%)	48 (47.5%)	36 (70.6%)	0.021
FFP (%)		16 (39.0%)	53 (52.5%)	15 (29.4%)	
Transport time		39 (± 18)	44 (± 17)	42 (± 18)	0.771
CPR (%)		0 (0%)	3 (2.97%)	5 (9.80%)	0.044
Intubation (%)		13 (31.7%)	65 (64.4%)	40 (78.4%)	<0.001
Blood (%)		11 (26.8%)	32 (31.7%)	22 (43.1%)	0.214
Crystalloid		800 (± 1400)	830 (± 1300)	1000 (± 1600)	0.891
PRBC (Median)		0 (± 1.0)	0 (± 1.0)	0 (± 2.0)	0.233
Hospital					
Transfusion 24h		2.0 (± 8.0)	7.0 (± 14)	12 (± 20)	<0.001
PRBC 24h (Median)		2.0 (± 5.0)	5.0 (± 7.0)	8.0 (± 10)	<0.001
Plasma 24h		0 (± 0)	2.0 (± 4.0)	4.0 (± 8.0)	<0.001
Platelets 24h		0 (± 0)	0 (± 1.0)	1.0 (± 2.0)	0.002
Crystalloid 24h		4800 (± 3800)	5300 (± 4000)	4600 (± 3000)	0.095
Vasopressors 24h		19 (46.3%)	68 (67.3%)	44 (86.3%)	<0.001
INR (Median [IQR])		1.2 (± 0.20)	1.3 (± 0.36)	1.6 (± 0.72)	<0.001
Other outcomes					
Coagulopathy (%)		16 (39.0%)	54 (53.5%)	44 (86.3%)	<0.001
ALI (%)		2 (4.88%)	47 (46.5%)	3 (5.88%)	<0.001
NI (%)		3 (7.32%)	43 (42.6%)	\	<0.001

Pearson's χ^2 test was used for calculating p value of categorical variables. Kruskal-Wallis test was used for calculating p value of continuous variables. ISS, injury severity score; AIS, abbreviated injury score; TBI, traumatic brain injury; GCS,

Glasgow coma score; SBP, systolic blood pressure; HR, heart rate; FFP, fresh frozen plasma; CPR, cardiopulmonary resuscitation; PRBC, packed red blood cells; INR, international normalized ratio; ALI, acute lung injury; NI, nosocomial infection; MOF, multiple organ failure; ICU, intensive care unit; LOS, length of stay.

Test was conducted across both healthy subjects and three outcome groups of trauma patients.

Question 13: I would recommend that the exploratory data analysis for the predictors and outcome used for each model be conducted. As far as I can tell, Table 1 gives information on all the 193 patients. For reader interested in determining sample size for each analysis, it is not easy to decipher. For example, for the logistic model of early death (early-non survivors), I reckon data for 51 patients were analysed. However, I am having a difficult time telling the sample size for the model analysing COVID-19 data (from Guo et. al – was the entire dataset on which their paper was based used? Or was a subset used? This should be clear in the paper).

Response: We thank the reviewer for these helpful comments. The selection for the predictors was based on our previous knowledge on the factors that may influence patient outcomes. We also conducted the exploratory analysis by univariate logistic regression (Please see below). We selected the variables of lipid concentration, TRISS, treatment arms, TBI and sex for modeling early-mortality in the PAMPer dataset. The LRS, ISS, head injury, treatment arms, age and sex were selected for modeling for a non-resolving pattern or time (days) to discharged from the ICU. The lymphocyte count, CRP, LRS, age and sex were selected for modeling severe COVID-19. We add a description for the sample size for each model in the figure legends and results section.

Figure. Association between outcome (Left: Early-non-survivors, Middle: Non-resolving, Right: Severe Covid-19) and variables by univariate analysis. Dash line represents a p-value of 0.05.

Abbreviations: Conc: Concentration, TRISS: Trauma and injury severity score, PH: Prehospital, TBI: Traumatic brain injury, ISS: injury severity score, LRS: Lipid Reprogramming score, Lym:

Lymphocyte count, CRP: C-reactive protein. GGT: Gamma-Glutamyl Transferase, ALT: Alanine Aminotransferase, DBIL: Direct Bilirubin, TBIL: Total Bilirubin.

Question 14: The authors include references to missing data in line 472, can I confirm that the percentage in Table 1 do not include missing values. In other words, do we have any missing values for the variables? I suspect no missingness based on presentation of the table.

Response: Correct. Table 1 does not include variables with missing values. However, other variables (medication history, lipids) contained missing values. We exclude all the missing values in the history information and impute the lipids concentration for statistical analysis.

Question 15: Prognostic models, need to be trained and tested using the same variables. Is there a justification for developing the survival model (time to discharge by ICU) using the variables reported in 545/6 but externally validating the model on TD-2 data (line 548) that contains all but one variable used in the internal validation? As I understand this is not typically done.

Response: We thank the reviewer for identifying this important point. We want to assure that the readers understand that the cox regression model was performed to identify the association between LRS and outcome and not to create a prognostic model. We acknowledge that there is systemic bias between the PAMPer trial and TD-2 datasets. As we mention in the manuscript, the TD-2 dataset is dominated by the resolving patients with limited non-survivors. On the contrary, the PAMPer trial is dominated by non-resolving patients with many non-survivors.

We also included the prognostic analysis recommended by the reviewer (see response to Question 17, below). The variables are consistent in both training and test datasets.

Question 16: How were the variables included in each model fitted selected? Was this based on clinical utility or a variable selection procedure?

Response:

For the association analysis, the selection for the predictors was based on our previous knowledge (from the literature) of the factors known to influence trauma patient outcomes together with the exploratory analysis (see also the response to question 13).

For the prognostic model, we used injury severity score (ISS) and IL6 as the reference model in trauma since this model was previously used (Raymond et al., 2020). The lymphocyte count and

CRP were set as the reference model for COVID-19 according to the literature (Wynants et al., 2020).

Question 17: How was the prognostic /prediction model developed evaluated? Prognostic models need to be internally validated based on their (i) Clinical utility [1], (ii) Discrimination (iii) and Calibration. The above measures can be corrected for bias and optimism using resampling methods (bootstrap or k-fold cross validation). Without these performance measures, it is difficult to objectively conclude the usefulness of the prognostic model used.

Response: Thank you for this valuable guidance. Based on our previous association analysis, we explored if there is added prognostic value of the LRS or the individual phosphatidylethanolamine (PE) species that comprise the LRS in trauma or COVID-19. The reason we include the individual PE species is that the LRS was calculated (based on the average level of 5 PE species) to reflect biological insights but was not optimized for outcome prediction

For trauma, we set a model using the injury severity score (ISS) and IL6, two variables known to correlate with mortality and complications, as the reference model. For COVID-19, we set a model based on lymphocyte count (Lym) and CRP levels as the reference model as both have been shown to correlate with disease severity in COVID-19 patients. We then calculated the increase in the AUC resulting from adding the LRS or each of the five PE species that comprise the LRS (Please see figure below) to the model. We found that LRS alone moderately improved the AUC (increase in AUC=0.018 of PAMPer dataset, increase in AUC=0.036 of Covid-19 dataset of Guo et al.) in trauma and COVID-19. We also found that a single PE that was specific to the disease process improved the AUC to an even greater extent than the LRS. For trauma this was PE (18:0/18:1) (increase in AUC=0.075) and for COVID-19 this was PE (16:0/22:6) (increase in AUC=0.062).

Figure S7A. The increase in AUC from the inclusion of the LRS or its components to the reference model in the training set. (Trauma: 73 patients in standard-of-care arm of the PAMPer dataset, Covid-19: 45 patients in training cohort (C1) from Guo et al.)

We further utilized an established two-step machine learning approach to identify a minimal set of predictive lipid biomarkers and clinical features for predicting the outcome in the PAMPer dataset (standard-of-care arm). This approach was based on feature selection (L1 Regularization) followed by classification (Support Vector Machine) using the down-selected features. The results suggest that PE (18:0/18:1) was the top selected feature among lipids and only IL6 and ISS ranked higher overall (See Figure S7B). Taken together, this indicated that a single PE well-represented in the datasets may have the highest prognostic value after taking into account the overall correlation structure of the lipid-omic dataset.

Figure S7B. Frequency of top10 feature selected by *two-step machine learning approach* in PAMPer dataset (73 patients in standard-of-care arm of PAMPer dataset, see also methods).

We then developed a prognostic model for predicting the recovery pattern of trauma patients (resolving vs. non-resolving). We set the standard-of-care arm in PAMPer study as training dataset. The plasma arm was set as an internal test set. The TD-2 dataset was set as an external test set. Three logistic regression models with 10-fold cross validation were developed in the training set and tested in the two test sets. The performance of discrimination was assessed by ROC curves and AUC values. The performance of calibration was assessed using a calibration curve and by calculating the brier score. The prediction was underestimated when true probability is around 0.25-0.5 in the reference model (ISS+IL6) but performed well at probabilities above 0.5.

Figure S7D. Calibration curve of three prognostic models in 73 patients from standard-of-care arm of the PAMPer dataset.

Addition of either LRS or PE (18:0/18:1) to ISS+IL-6 improved the prediction accuracy (left panel, AUC: ISS+IL6, 0.798; ISS+IL6+LRS, 0.816; ISS+IL6+PE (18:0/18:1), 0.873). The brier score was also lower with the inclusion of LRS or PE (18:0/18:1) levels (ISS+IL6:0.177, ISS+IL6+LRS:0.166, ISS+IL6+PE (18:0/18:1):0.139). The overall discrimination performance was improved in the training set and the two test sets (Middle panel, AUC in test set1: ISS+IL6, 0.876; ISS+IL6+LRS, 0.916; ISS+IL6+PE (18:0/18:1), 0.900, Right panel, AUC in test set2: AUC: ISS+IL6, 0.797; ISS+IL6+LRS, 0.814; ISS+IL6+PE (18:0/18:1), 0.841).

Figure 5D & S7EF. ROC curves for three prognostic models in the training dataset (left), internal test dataset (middle) and external test dataset (right).

Similarly, we adopted the two-step machine learning approach to identify a minimal set of predictive lipid biomarkers and clinical features for predicting the outcome in the Covid-19 dataset (C1 of Guo et al.). The results suggested that PE (16:0/22:6) was the top selected feature among lipids. The clinical feature of lymphocyte count and CRP were ranked at 6th and 7th respectively.

Figure S7C. Frequency of top10 feature selected by two-step machine learning approach in Covid-19 dataset (45 patients in C1 of Guo et al, see also methods).

We also developed a reference prognostic model (Lym+CRP) for predicting disease progression in COVID-19 patients (non-severe vs. severe). Here, LRS or PE (16:0/22:6) as single variables were used for prediction since there was no additional value observed by adding these to Lym + CRP levels. We set the cohort 1 and 2 in dataset of Guo et al. as the training set and test set, respectively. The reference model was trained via logistic regression with 10-fold cross validation. The model PE (16:0/22:6) showed the best performance of calibration compared to other two models (Brier Score: Lym+CRP:0.177; LRS: 0.166; PE (16:0/22:6): 0.139.).

Figure S7G. Calibration curve of three prognostic models in 45 patients from the Covid-19 dataset (C1 of Guo et al.)

The performance of discrimination was comparable between the reference model and LRS in both training and test datasets. However, the performance of PE (16:0/22:6) alone was better than the reference model (AUC 0.917 vs. 0.958). Interestingly, we noticed that only one patient (XG43) was mis-labeled when we used PE (16:0/22:6) levels in test-set. This is comparable to the random forest (17 proteins+9 metabolites) model from the original manuscript. Thus, we present a much-simplified model based on a single molecule for predicting the outcome of Covid-19.

Figure 5G&S7HI. ROC curves of three prognostic models in training set (left), test set (middle). Right: Performance of PE (16:0/22:6) in the test set of 10 COVID-19 patients.

All metrics for performance of discrimination and calibration can be found in Table S7(see below).

Table S7 Performance of prognostic value for LRS and individual PE species in Trauma and COVID-19				
	Discrimination (AUC Value)	Added AUC value	Calibration (Brier Score)	Decreased Brier Score
Trauma				
Train set (Standard Arm)				
ISS+IL6	0.798(0.696-0.901)	Ref	0.177	Ref
ISS+IL6+LRS	0.816(0.719-0.913)	0.018	0.166	-0.011
ISS+IL6+PE(18:0/18:1)	0.873(0.795-0.952)	0.075	0.139	-0.038
Internal test set (Plasma Arm)				
ISS+IL6	0.876(0.787-0.966)	Ref	0.126	Ref
ISS+IL6+LRS	0.916(0.845-0.988)	0.040	0.116	-0.010
ISS+IL6+PE(18:0/18:1)	0.900(0.821-0.978)	0.024	0.136	0.010
External test set (TD-2 Dataset)				
ISS+IL6	0.797(0.703-0.891)	Ref	0.225	Ref
ISS+IL6+LRS	0.814(0.721-0.907)	0.017	0.214	-0.011
ISS+IL6+PE(18:0/18:1)	0.841(0.750-0.932)	0.044	0.198	-0.027
COVID-19				
Train set (C1 of guo et al.)				
Lym+CRP	0.786 (0.631–0.941)	Ref	0.196	Ref
LRS	0.814 (0.684–0.944)	0.028	0.174	-0.022
PE(16:0/22:6)	0.862 (0.753–0.971)	0.076	0.150	-0.046
Test set (C2 of guo et al.)				
Lym+CRP	0.917 (0.728–1.000)	Ref	0.123	Ref
LRS	0.833 (0.557–1.000)	-0.084	0.150	0.027
PE(16:0/22:6)	0.958 (0.843–1.000)	0.045	0.075	-0.048
Logistic regression model with 10-fold cross-validation was used in train set. The same model was used to predict in the validation set.				
The performance of the model was evaluated by AUC value and brier score. 95% confidence interval was calculated for AUC value.				
ISS: Injury Severity Score; LRS: Lipid Reprogramming Score.				

This analysis has been added to the manuscript along with the following text:

Results:

Line 286 to 303

We next explored the prognostic value of the LRS and the five individual PE species that comprise the LRS for predicting whether trauma patients would progress to a non-resolving pattern (Table S7). Here, we set the standard-of-care arm in PAMPer dataset as the training set (n=73). The TP arm from PAMPer dataset was set as an internal test set (n=69) and the TD-2 dataset was set as an external test set (n=86). Compared to the reference model(Raymond et al., 2020) (ISS+IL6, AUC=0.798), adding the LRS moderately improved the performance of discrimination (AUC=0.816, added AUC =0.018) in the training set (Fig 5D, sFig 7A). Interestingly, of the five PE that comprise the LRS, PE (18:0/18:1) also greatly improved the performance of discrimination (AUC=0.873, added AUC =0.075) in the training set (Fig 5D, sFig 7A). We further utilized an established two-step machine learning approach (Ackerman et al., Nature Medicine 2018; Suscovich et al Science Translational Medicine 2020; Das et al PLoS Pathogens 2020) to identify a minimal set of predictive lipid biomarkers and clinical features for predicting the outcome in the PAMPer dataset (standard-of-care arm). This approach was based on feature selection (L1 Regularization – LASSO to avoid overfitting) followed by classification (Support Vector Machine) using the down-selected features. The results suggested that PE (18:0/18:1) was the top selected feature among lipids and it is only

behind IL6 and ISS ranked higher overall (**sFig 7B**). The performance of calibration (**sFig 7C**) was also improved by adding either the LRS or the single PE (18:0/18:1) to ISS+IL6 (Brier Score: ISS+IL6, 0.177; ISS+IL6+LRS, 0.166; ISS+IL6+PE (18:0/18:1), 0.139). The results were consistent in the internal (**sFig 7D**, AUC: ISS+IL6, 0.876; ISS+IL6+LRS, 0.916; ISS+IL6+PE (18:0/18:1), 0.900) and external test sets (**sFig 7E**, AUC: ISS+IL6, 0.797; ISS+IL6+LRS, 0.814; ISS+IL6+PE (18:0/18:1), 0.841).

Line 312 to 326

C-reactive protein (CRP) and lymphocyte count (Lym) are known to correlate with worse outcomes in COVID-19 patients (Wynants et al., 2020). We compared the LRS and its five individual PE species with these two variables to classify severe versus non-severe patients in both a training set (C1, n=45) and test set (C2, n=10) (**Table S7**). We found that the LRS alone moderately improved the performance of discrimination (AUC=0.814, added AUC=0.028), however a single PE specie from the LRS (PE (16:0/22:6)) alone greatly improved the performance (AUC=0.862, added AUC=0.076) (**Fig5G**, **sFig 7A**). The two-step machine learning approach also revealed that the PE (16:0/22:6) were the top selected features (**sFig7C**). The performance of calibration was also improved by using PE (16:22:6) (Brier Score: Lym+CRP:0.177; LRS: 0.166; PE (16:0/22:6): 0.139; **sFig7G**). The performance of PE (16:0/22:6) also had the highest AUC compared to other two models in the test set (**sFig 7F**, AUC: CRP+Lym, 0.917; LRS, 0.833, PE (16:0/22:6), 0.958). Interestingly, we also noticed that PE (16:0/22:6) showed similar performance for prognostication as the random forest model based on 17 proteins and 9 metabolites reported in the original manuscript. It is notable that only one patient (XG43) was mislabeled in the test cohort (C1) using PE (16:0/22:6) for prognostication (**sFig 7G**). Thus, similar to our observations in trauma, a single PE specie derived from the LRS performed well of prognostication for severe disease in COVID-19.

Methods:

Line 611-616

Multi-variable regression analysis

For prognostic model generation, we built logistic regression models for predicting patients who developed non-resolving pattern in the trauma cohorts (Training set: standard-of-care arm in PAMPer dataset, Test set: plasma arm in PAMPer dataset or TD-2 dataset) or who progressed into severe COVID-19 in the cohort of Guo et al (Train:C1, Test:C2). All models were internally evaluated by 10-fold cross-validation. The performance of discrimination and calibration were

assessed by ROC curve, and AUC values. Performance of calibration was assessed calibration curves, and brier scores respectively.

Line 618-627

A two-step machine learning approach

The importance of clinical features and lipids commonly detected in trauma and Covid-19 datasets were evaluated by two-step feature selection strategy as previously reported (Ackerman et al., Nature Medicine 2018; Suscovich et al Science Translational Medicine 2020; Das et al PLoS Pathogens 2020). Briefly, two models (least absolute shrinkage and selection operator: lasso; support vector machine with the radial basis function: SVM-RBF) were fitted sequentially with 100 times repeated and nested 5-fold cross-validation (Both outer and inner sampling were 5 folds). The lasso model is fitted into inner sampling of each 5-fold and the hyperparameter were tuned by the performance of classification (accuracy). Then a fold-specific classifier was trained by SVM-RBF in the selected features from the lasso model in the same fold. The performance (accuracy) of the fold-specific classifier was internally evaluated in the outer-sampling. The frequency of selected features was summarized in all models with accuracy >0.8. Top10 selected features were kept for visualization.

Reviewer comments, further round review–

Reviewer #1 (Remarks to the Author):

Major areas of concern:

1) Lipidomic Methodology and reporting of raw data

Question 1: Line 113: "In the quality control analysis, the median relative standard deviation (RSD) for the lipid panel was 4%". How was the quality control sample generated and how often was it analysed during the experiment? What were the RSD values for the 900 lipids? What was the RSD threshold for lipid selection?

Response: We are pleased to provide this important information. The QC sample was generated by combining a small aliquot from entire set of samples into a single pooled CMTX (Sample matrix). Four aliquots of the CMTX were run on each plate of 36 samples. One each was injected at the beginning and end of the run, with the other two roughly evenly spaced in between the remaining samples. A similar process was carried out for three process blanks (water) and MTRX7 (a well characterized plasma sample Metabolon runs on every plate). Unfortunately, Metabolon does not report on the RSD values for each biochemical, only the median value. There is no threshold for lipid selection (or any biochemical selection in the untargeted dataset) based on RSD. Metabolon calculated the median RSD from all species detected in 100% of the CMTX samples. We have now added this information in the methods section.

In that case we don't know if lipids with, say technical RSD>30%, were included in the analysis. With a median RSD in the lower single digit it seems unlikely that many species could display very high RSD. Nevertheless, the restriction to median RSD remains a shortcoming in reporting of raw data. The authors should instead report RSD for each analyte. This becomes particularly relevant when small groups of lipids, or even a single lipid species are used (such as the two distinct PE used for improvement of prognostic value in trauma and COVID-19, respectively).

Question 2: Line 464: "Lipid species were background-subtracted using concentrations detected in process blanks..." what was the signal to noise (S/N) threshold for acceptance of results?

Response: Background levels were estimated/calculated from the median levels of the three process blanks (water) if there were detectable levels in at least 2 of the 3 blanks in each batch. The background level was subtracted from each sample in the batch prior to any run day normalization. If the value was negative, it was set to zero or "not detected". Metabolon explained to us that after extensive analysis, they have found that implementing this step gives much better data quality for some lipid species that can be found on plastic tubes or other artifactual sources in their system. We have now added the approach to subtract background levels to the methods section.

Using the described approach without any rejection criteria based on S/N ratios will result in any signal above blank to be included and represented, irrespective of how unrealistically low its concentration might be. This is unusual practice in targeted lipidomics. Signal to noise ratios for individual lipids should be reported instead (see also reply to response 5).

Question 3: The nomenclature of all lipids reported is based on sn-1 or -2 location of fatty acids (PE 16:0/18:2); this might not be possible with the used setup unless the authors can explain how the position of the fatty acids was verified. An alternative possible nomenclature for the same species might be according to PE 16:0_18:2. Please see Shorthand notation for lipid structures derived from mass spectrometry. *J. Lipid Res.* 2013. 54: 1523–1530.

Response: Unfortunately, the complex lipid panel (CLP) performed by Metabolon does not distinguish between SN1 and SN2 location of the side chains. The nomenclature used simply lists the 2 sidechains present without attempting to ascribe which sidechain resides at which position. We have now added this information to the methods section.

This reviewer understands the complication of assigning fatty acid positions based on lipidomic

approaches. And that is why it needs to be reported in an accepted form to reflect this limitation. In the revised version of the manuscript the nomenclature reporting is inappropriate in all the files, including the supplemental tables. It needs to be changed accordingly throughout.

Question 5: No raw lipidomic data is reported.

Response: We have uploaded the raw and processed lipidomic datasets into Mendeley Data (<https://data.mendeley.com/datasets/7stf7dtxcz/draft?a=3e078e7f-5068-4b8e-a5a9-ef414db279bd>) and in the supplementary materials. This information is now available to the reviewers and would also be available to the readers of our paper.

Inclusion of the lipidomic data is noted. It is reported as final concentration values which is fine. However, in that case, the authors have to state which standards were used and at which concentrations in order to derive these values from the raw data. Furthermore, without any indication of intensity and S/N cut-off values, it is not possible to judge how far (or close) to noise levels a particular lipid species would be situated. (See also reply to response 2).

Question 3: Validation is not based on independent cohorts interrogated experimentally. One would need prospective trials to support the claims for the prognostic panel.

Response: We thank the reviewer for making this important point to the novel insights provided by our findings. We also agree that prospective cohorts should be used to establish the prognostic value of lipid levels. Based on the reviewer's comment and comment #17 from reviewer 4, we have now carried out an extensive analysis of the prognostic value of the LRS and the five individual PE species that comprise the LRS in trauma and COVID-19. To do this, we used our two prospective trauma cohorts (PAMPer trial and TD-2 datasets) and a prospectively collected COVID-19 dataset from Guo et al.

For trauma, we set a model using the injury severity score (ISS) and IL6, two variables known to correlate with mortality and complications, as the reference model. For COVID-19, we set a model based on lymphocyte count (Lym) and CRP levels as the reference model as both have been shown to correlate with disease severity in COVID-19 patients. We then calculated the increase in the AUC resulting from adding the LRS or each of the five PE species that comprise the LRS (Please see figure below) to the model. We found that LRS alone moderately improved the AUC (increase in AUC=0.018 of PAMPer dataset, increase in AUC=0.036 of Covid-19 dataset of Guo et al.) in trauma and COVID-19. We also found that a single PE that was specific to the disease process improved the AUC to an even greater extent than the LRS. For trauma this was PE(18:0/18:1)(increase in AUC=0.075) and for COVID-19 this was PE(16:0/22:6)(increase in AUC=0.062).

The fact that slight changes of lipid species numbers and identities which are utilized in the analysis (and based on their commonality in different datasets) affects outcomes leaves some doubt on the generalizability of the results beyond this study. The authors acknowledge this in the discussion on page 13 which reads "The specificity of single PE species for prognostication for trauma and COVID-19 might represent characteristics specific to the patients, the causes of the critical illness, or the differences in the methods used to measure the lipids in the studies."

We further utilized an established two-step machine learning approach to identify a minimal set of predictive lipid biomarkers and clinical features for predicting the outcome in the PAMPer dataset (standard-of-care arm). This approach was based on feature selection (L1 Regularization) followed by classification (Support Vector Machine) using the down-selected features. The results suggest that PE (18:0/18:1) was the top selected feature among lipids and only IL6 and ISS ranked higher overall (See Figure S7B). Taken together, this indicated that a single PE well-represented in the datasets may have the highest prognostic value after taking into account the overall correlation structure of the lipid-omic dataset.

Emphasis on a single PE species calls for detailed analytical characteristics for precisely this species as opposed to composite values among 900 other lipids. (Please see also comments above).

Reviewer #2 (Remarks to the Author):

Excellent and thorough revisions in response to the initial review. The limitations of the studies are now better described also.

Reviewer #3 (Remarks to the Author):

the authors addressed all comments and significantly improved the paper

Reviewer #4 (Remarks to the Author):

Firstly, as before, I limit my comments to the statistical methods and their appropriateness based on the description of the data and the outcomes analyzed. As I do not have expertise in COVID-19, Critical Care (trauma treatment) or lipid biology, I cannot comment on the clinical relevance of the data used for this research.

Secondly, I appreciate the efforts that the authors have taken to address the concerns raised in my previous review. The work done and revisions made have improved the quality of the paper.

Thirdly, I have two comments. The first is about correction of a typo in Fig S6 (panels G, H and I) where "patients" is written as "pateints". The second comment regards the description in lines 614-616 of the revised manuscript. Specifically, the area under the Receiver Operating Characteristic (ROC) curve, is used to assess discrimination ability of a model (and not calibration). Also, Brier scores are an overall measure of fit for a model (which include aspects of both calibration and discrimination). These distinctions are important. Please see the following paper.

Steyerberg EW, Vickers AJ, Cook NR, Gerds T, Gonen M, Obuchowski N, Pencina MJ, Kattan MW. Assessing the performance of prediction models: a framework for some traditional and novel measures. *Epidemiology* (Cambridge, Mass.). 2010 Jan;21(1):128.

Ms. No.: NCOMMS-20-46047C

Title: Lipidomic Signatures Align with Inflammatory Patterns and Outcomes in Critical Illness

Summary of the major concerns and general response

We thank all the reviewers and editors for their careful re-review of our manuscript. Based on the comments of reviewers 1 and 4, we made the following changes:

1. We provide the additional technical details of the targeted lipidomic analysis platform (Metabolon Inc.).
2. We followed all suggestions for minor edits and all typos were corrected.
3. We clarify how the ROC curves and Brier scores were used and interpreted in the manuscript.

Point-by-point Responses to the Reviewers' Comments

REVIEWER COMMENTS

Reviewer #1 (Remarks to the Author):

Major areas of concern:

1) Lipidomic Methodology and reporting of raw data

Question 1: Line 113: “In the quality control analysis, the median relative standard deviation (RSD) for the lipid panel was 4%”. How was the quality control sample generated and how often was it analysed during the experiment? What were the RSD values for the 900 lipids? What was the RSD threshold for lipid selection?

Response: We are pleased to provide this important information. The QC sample was generated by combining a small aliquot from entire set of samples into a single pooled CMTX (Sample matrix). Four aliquots of the CMTX were run on each plate of 36 samples. One each was injected at the beginning and end of the run, with the other two roughly evenly spaced in between the remaining samples. A similar process was carried out for three process blanks (water) and MTRX7 (a well characterized plasma sample Metabolon runs on every plate). Unfortunately, Metabolon does not report on the RSD values for each lipid, only the median value. There is no threshold for lipid selection (or any biochemical selection in the untargeted dataset) based on RSD. Metabolon

calculated the median RSD from all species detected in the CMTX samples. We have now added this information in the methods section.

Additional comments under question #1:

In that case we don't know if lipids with, say technical RSD>30%, were included in the analysis. With a median RSD in the lower single digit it seems unlikely that many species could display very high RSD. Nevertheless, the restriction to median RSD remains a shortcoming in reporting of raw data. The authors should instead report RSD for each analyte. This becomes particularly relevant when small groups of lipids, or even a single lipid species are used (such as the two distinct PE used for improvement of prognostic value in trauma and COVID-19, respectively).

Response:

Metabolon Inc provided us a report with the median RSD values for each class of lipids. The table below highlights the RSD values for the 14 classes targeted in the assay (CMTRX samples) along with the number of analytes contributing to the value. The three lipids classes with the highest median RSD were MAG (20.73%), DCER (15.28%) and PI (14.16%). Importantly, these lipids classes were not the major focuses in our analysis. In addition, there were only 7 individual lipids with RSD more than 30% (DAG (12:0_18:1):30.05%, DAG (16:1_22:6): 50.21%, DAG (18:0_22:6): 57.63%, DCER (26:0):49.99%, MAG (20:0): 45.20%, MAG (24:0): 36.09%, TAG53:7-FA18:3: 33;97%.) and this would not influence our global analysis (Compared to the total 998 lipids detected). The RSD values of the two PE species (PE (18:0_18:1), PE (16:0_22:6)) that were found to be predictive of patient outcomes were 9.82% and 5.51%, respectively. We have added both the table (Table S8) and the RSD values for the two PE species to the manuscript.

CLASS	MEDIAN RSD (%)	NUMBER
CE	2.78	26
CER	5.47	11
DAG	4.63	57
DCER	15.28	13
HCER	7.15	12
LCER	7.45	11
LPC	4.71	8
LPE	11.92	7
MAG	20.73	19

PC	6.76	35
PE	10.22	45
PI	14.16	5
SM	3.43	12
TAG	3.47	507

Table S8 Summary of median relative standard deviation (RSD) values in 14 classes of metabolomics assay

The modified section in the manuscript reads as follows:

Line 297-299:

*Interestingly, of the five PE that comprise the LRS, PE (18:0_18:1) (**RSD:9.82%**) also greatly improved the performance of discrimination (AUC=0.873, added AUC =0.075) in the training set (Fig 5D, sFig 7A).*

Line 322-323:

*“a single PE specie from the LRS (PE (16:0_22:6), **RSD:5.51%**) alone greatly improved the performance (AUC=0.862, added AUC=0.076) (Fig5G, sFig 7A).”*

Line 525-528:

*The internal standard served as a technique replicate and was run multiples times throughout the experiment. Instrument variability was evaluated by calculating median relative SD (RSD) from the quality control sample matrix. **The median RSD values for 14 lipid classes can be found in Table S8.***

Question 2: Line 464: "Lipid species were background-subtracted using concentrations detected in process blanks..." what was the signal to noise (S/N) threshold for acceptance of results?

Response: Background levels were estimated/calculated from the median levels of the three process blanks (water). If there were detectable levels in at least 2 of the 3 blanks in each batch, the background level was subtracted from each sample in the batch prior to any run day normalization. If the value was negative, it was set to zero or “not detected”. Metabolon explained to us that after extensive analysis, they have found that implementing this step gives much better data quality for some lipid species that can be found on plastic tubes or other artifactual sources in their system. We have now added the approach to subtract background levels to the methods section.

Additional comments under question under question #2:

Using the described approach without any rejection criteria based on S/N ratios will result in any signal above blank to be included and represented, irrespective of how unrealistically low its concentration might be. This is unusual practice in targeted lipidomics. Signal to noise ratios for individual lipids should be reported instead (see also reply to response 5).

Response:

We are pleased to provide this information. Metabolon explained us that the targeted metabolomics assay they performed (the complexed lipids panel: CLP) differs in several important ways from other targeted assays. Commonly with chromatographic methods, signal to noise is calculated based on a comparison of the signal intensity when a peak is eluting vs the signal intensity before or after the peak. However, there is no chromatographic method associated with the CLP assay and thus quantitation is based on intensity of signal as opposed to area under the curve as with other chromatographic methods. Thus, there is no direct way to determine the noise level. This is why a PRCS water blank is used, so we can get a sense of what the signal is when a blank sample is injected. Moreover, the CLP assay also differs in a number of ways from smaller targeted assays that use calibration curves. A calibration curve typically consists of at least five calibrator samples that contain a range of known concentrations of each analyte, as well as isotopically labeled internal standards; the calibrators and the unknown samples are subjected to the same analysis, and the intensity ratios between analytes and internal standards measured in each unknown sample are compared to those in the calibrator samples to calculate the analyte concentrations in the unknown samples. Clearly, this approach would be impractical if not impossible for quantifying the >1100 analytes included in the CLP, many of which lack commercially available authentic standards and isotopically labeled internal standards. Instead, the CLP relies on a single known concentration of each isotopically labeled IS, in what is known as a single-point calibration. Therefore, the CLP does not provide lower limit of detection (LLOD) and lower limit of quantification (LLOQ) for each analyte, which are typically reported for smaller targeted assays.

It is acknowledged that the reviewer's point is valid in that there may be some species measured with unrealistically low concentrations based on the methodology of the assay. However, we did not intend to focus on individual lipids when making biological conclusions. Instead, we used a compositional analysis to drive our understanding and interpretation. The exception to this, was the use of two PE species for predictive modelling and these were both present at higher and easily detectable levels.

Question 3: The nomenclature of all lipids reported is based on sn-1 or -2 location of fatty acids (PE 16:0/18:2); this might not be possible with the used setup unless the authors can explain how the

position of the fatty acids was verified. An alternative possible nomenclature for the same species might be according to PE 16:0_18:2. Please see Shorthand notation for lipid structures derived from mass spectrometry. *J. Lipid Res.* 2013. 54: 1523–1530.

This reviewer understands the complication of assigning fatty acid positions based on lipidomic approaches. And that is why it needs to be reported in an accepted form to reflect this limitation. In the revised version of the manuscript the nomenclature reporting is inappropriate in all the files, including the supplemental tables. It needs to be changed accordingly throughout.

Response: Thank you for providing the reference and the recommendation for the Shorthand notation. We have changed the manuscript (including supplemental files) to use the Shorthand notation for individual lipid species.

Question 4: No raw lipidomic data is reported.

Response: We have uploaded the raw and processed lipidomic datasets into Mendeley Data (<https://data.mendeley.com/datasets/7stf7dtxcz/draft?a=3e078e7f-5068-4b8e-a5a9-ef414db279bd>) and in the supplementary materials. This information is now available to the reviewers and would also be available to the readers of our paper.

Additional comments under question #4:

Inclusion of the lipidomic data is noted. It is reported as final concentration values which is fine. However, in that case, the authors have to state which standards were used and at which concentrations in order to derive these values from the raw data. Furthermore, without any indication of intensity and S/N cut-off values, it is not possible to judge how far (or close) to noise levels a particular lipid species would be situated. (See also reply to response 2).

Response:

We are pleased to provide the information. The CLP assay utilizes internal standards across the 14 lipid classes monitored, with most classes comprising multiple standards in order to maximize the similarity between each analyte and its internal standard. As per the example below shown for the PC standards, Metabolon includes a range of lipid standards which cover both carbon length and number of double bonds. Experimental lipids are matched to specific standards based on the combination of carbon length and number of double bonds.

Shown below is the number of standards which are included per class.

CEs have 8 standards

CERs have 8 standards
SMs have 4 standards
DAGs have 8 standards
TAGs have 8 standards
MAGs have 1 standard
LPCs have 1 standard
LPEs have 1 standard
PCs have 10 standards
PEs have 8 standards
PIs have 8 standards

The validity of the CLP's quantification methodology is confirmed by its quantitative precision and accuracy, which have been verified through a rigorous validation process. In particular, the accuracy of the lipid concentration measurements has been validated by comparison to a benchmark technology (a combination of thin-layer chromatography and gas chromatography / mass spectrometry).

Question #5:

Emphasis on a single PE species calls for detailed analytical characteristics for precisely this species as opposed to composite values among 900 other lipids. (Please see also comments above).

Response: We appreciate the reviewer's suggestion. As for the for detailed analytical characteristics, these two lipids have RSD values of 9.82% and 5.51%, respectively.

While a single PE species emerged as the most important for each of the clinical conditions, the multivariate machine learning models clearly showed that adding a few other key features to the single PE species improved performance beyond what that single PE species achieved. Further, the composite values do not span 900 other lipids, rather they include very specific lipid species/features identified by rigorous feature selection. The importance of these composites lies in better (compared to a single feature) accounting for variability in a human cohort (i.e., the decision boundaries based on composites are statistically more robust than decision boundaries based on a single PE species). Most importantly, the value of the manuscript lies in demonstrating how a machine learning approach on multi-omic data can be used in the context of trauma biology to identify robust prognostic markers.

Reviewer #2 (Remarks to the Author):

Excellent and thorough revisions in response to the initial review. The limitations of the studies are now better described also.

Response: Thank you

Reviewer #3 (Remarks to the Author):

The authors addressed all comments and significantly improved the paper

Response: Thank you

Reviewer #4 (Remarks to the Author):

Firstly, as before, I limit my comments to the statistical methods and their appropriateness based on the description of the data and the outcomes analyzed. As I do not have expertise in COVID-19, Critical Care (trauma treatment) or lipid biology, I cannot comment on the clinical relevance of the data used for this research.

Secondly, I appreciate the efforts that the authors have taken to address the concerns raised in my previous review. The work done and revisions made have improved the quality of the paper.

Response: Thank you

Thirdly, I have two comments. The first is about correction of a typo in Fig S6 (panels G, H and I) where "patients" is written as "pateints". The second comment regards the description in lines 614-616 of the revised manuscript. Specifically, the area under the Receiver Operating Characteristic (ROC) curve, is used to assess discrimination ability of a model (and not calibration). Also, Brier scores are an overall measure of fit for a model (which include aspects of both calibration and discrimination). These distinctions are important. Please see the following paper.

Steyerberg EW, Vickers AJ, Cook NR, Gerds T, Gonen M, Obuchowski N, Pencina MJ, Kattan MW. Assessing the performance of prediction models: a framework for some traditional and novel measures. *Epidemiology* (Cambridge, Mass.). 2010 Jan;21(1):128.

Response: Thank you. We have made the correction to the typos in Fig S6. Thank you for providing the very helpful reference. We have edited the relevant sentence (Line 614-616) to reflect the correct use and interpretation of ROC curves and brier scores as follows:

The performance of discrimination was assessed by ROC curves and AUC values. Performance of calibration was assessed by calibration curves. Brier scores were used to evaluate the overall fit of the model.

Reviewer comments, further round review–

Reviewer #1 (Remarks to the Author):

The shorthand nomenclature has not been introduced neither to the supplemental material provided in xls files nor the Mendeley link provided. This needs to be rectified. The response to query #5 (recommendation for PE to be measured via different platform) has not been fully addressed either. Furthermore, disclosure of methodological details such as internal standard information must be included.

Ms. No.: NCOMMS-20-46047C

Title: Lipidomic Signatures Align with Inflammatory Patterns and Outcomes in Critical Illness

Summary of the major concerns and general response

We greatly appreciate the reviewers' efforts to review our manuscript and for their insightful comments to improve our paper. By addressing the concerns raised by the reviewers, we believe the paper has been substantially improved. These major changes include the following:

1. We applied another lipidomic platform in matched subsets from PAMPer trial to technically validate the results of 5 PE species. The validation assay confirmed selective elevations of 5 PE species identified in the Metabolon platform in the patients that resolve slowly.
2. We provided additional details about the disclosure about the two platforms and the internal standards.
3. The shorthand nomenclature was corrected across the manuscript and supplementary materials.

Point-by-point Responses to the Reviewers' Comments

REVIEWER COMMENTS

Reviewer #1 (Remarks to the Author):

Question 1: The shorthand nomenclature has not been introduced neither to the supplemental material provided in xls files nor the Mendeley link provided. This needs to be rectified.

Response: We have corrected the shorthand nomenclature in the supplemental material and raw dataset in Mendeley link.

Question 2: The response to query #5 (recommendation for PE to be measured via different platform) has not been fully addressed either.

Response: We utilized another lipidomic platform (LC-HRMS, Platform 2, see also methods) to quantitatively measure the concentrations of 5 PE species (PE(18:0_18:1), PE(16:0_18:2), PE(16:0_20:4), PE(18:0_22:6), PE(16:0_22:6)). In the previous version of the paper, these 5 PE species were selectively up-regulated in non-resolving trauma or severe covid-19 patients and were, therefore, used to construct the Lipid Reprogramming Score (LRS). In order to assure that the patients selected for this validation assay were representative of the overall cohort and that injury severity would not influence the differences in PE levels, we conducted propensity score matching between resolving (n=14) and non-resolving trauma patients (n=15) from the PAMPer trial (**Table S9**). We also included samples from 8 non-fasting healthy subjects as a control group for comparison.

The results are shown in Fig S9. These 5 PE species were highly correlated (PE (18:0_18:1): $r=0.93$, PE (16:0_18:2): $r=0.92$, PE (16:0_20:4): $r=0.83$, PE (18:0_22:6): $r=0.84$, PE (16:0_22:6): $r=0.86$) between platform of LC-MS/MS (Metabolon methods, Platform 1, PF1) and LC-HRMS (Platform 2,

PF2) (**Fig S9A**). Consistently, all 5 PE species were selectively up-regulated in non-resolving trauma patients at 72h (**Fig S9B**). We note that there are some differences in the absolute levels of PE between the two platforms, however the relative differences between the patient subgroups and healthy controls recapitulated the findings from the Metabolon analysis.

Figure S9. Technical Validation of 5 PE species for construction of LRS.

A) Correlation of 5 PE species from LRS between two technical platforms (PF1, PF2: see also methods) from selected subjects (8 Non-fasting healthy controls, 29 Trauma patients at 72h timepoint). R: Pearson correlation coefficient.

B) Comparison of the plasma concentrations of 5 PE species of PF2 in healthy controls (HC) (n=7, one outlier was excluded), resolving trauma patients at 72h (n=14), Non-resolving trauma patients at 72h (n=15). P-value were calculated by student t-test between two groups.

Table S9. Characteristics of the matched male patients for technical validation of 5 PE species

Variables	Resolving (N=14)	Non-resolving (N=15)	p-value
Demographics			
Age (Median [IQR])	34 (± 30)	40 (± 25)	0.71
Race (% White)	13 (92.9%)	13 (86.7%)	1
Injury characteristics			
ISS (Median [IQR])	22 (± 14)	20 (± 8.0)	0.913
Head AIS (Median [IQR])	2.0 (± 3.8)	2.0 (± 2.5)	0.568
TBI (%)	7 (50.0%)	6 (40.0%)	0.867
GCS (Median [IQR])	14 (± 11)	7.0 (± 12)	0.363
SBP<70mmHg (%)	9 (64.3%)	6 (40.0%)	0.349
HR (Median [IQR])	120 (± 8.0)	120 (± 14)	0.872
Injury type (% Blunt)	10 (71.4%)	11 (73.3%)	1
Prehospital			
Treatment arm			
Standard care (%)	8 (57.1%)	6 (40.0%)	0.581
TP (%)	6 (42.9%)	9 (60.0%)	
Transport time (Median [IQR])	46 (± 31)	50 (± 30)	0.948
CPR (%)	0 (0%)	0 (0%)	\
Intubation (%)	5 (35.7%)	8 (53.3%)	0.562
Blood (%)	5 (35.7%)	6 (40.0%)	1
Crystalloid (Median [IQR])	380 (± 1200)	1300 (± 800)	0.102
PRBC (Median [IQR])	0 (± 2.0)	0 (± 1.0)	0.94
Hospital			
Transfusion 24h (Median [IQR])	4.0 (± 9.5)	9.0 (± 13)	0.205
PRBC 24h (Median [IQR])	4.0 (± 4.8)	7.0 (± 6.0)	0.324
Plasma 24h (Median [IQR])	0 (± 3.5)	2.0 (± 6.0)	0.224
Platelets 24h (Median [IQR])	0 (± 1.0)	0 (± 1.5)	0.979
Crystalloid 24h (Median [IQR])	4600 (± 4200)	5000 (± 2700)	0.913
Vasopressors 24h (%)	9 (64.3%)	10 (66.7%)	1
INR (Median [IQR])	1.3 (± 0.15)	1.3 (± 0.27)	0.567
Outcome			
Coagulopathy (%)	7 (50.0%)	10 (66.7%)	0.594
ALI (%)	1 (7.14%)	6 (40.0%)	0.103
NI (%)	2 (14.3%)	7 (46.7%)	0.138
MOF (%)	12 (85.7%)	14 (93.3%)	0.95
Vent days (Median [IQR])	2.0 (± 2.5)	8.0 (± 9.5)	<0.001
ICU LOS (Median [IQR])	3.0 (± 2.0)	15 (± 14)	<0.001
Hospital LOS (Median [IQR])	12 (± 11)	20 (± 16)	0.027

Pearson's χ^2 test was used for calculating p value of categorical variables. Kruskal-Wallis test was used for calculating p value of continuous variables.

Abbreviations: ISS, injury severity score; AIS, abbreviated injury score; TBI, traumatic brain injury; GCS, Glasgow coma score; SBP, systolic blood pressure; TP, thawed plasma; CPR, cardiopulmonary

resuscitation; PRBC, packed red blood cells; INR, international normalized ratio; ALI, acute lung injury; NI, nosocomial infection; MOF, multiple organ failure; ICU, intensive care unit; LOS, length of stay.

We add this related information in the main text as:

Results section:

Line 260-266:

To further technically validate the results, we utilized second platform (LC-HRMS, Platform2) to quantify the circulating concentrations of the five PE species in trauma patients (n=29) matched in characteristics with the original cohorts. These patients included both resolving and non-resolving patients as well as eight healthy controls (Table S8). All five PE species were highly correlated between the two platforms (LC-MS/MS, Platform 1, PF1; LC-HRMS, Platform 2, PF2) (Fig S9A). All five were also significantly upregulated in nonresolving trauma patients at 72h in the validation assay (Fig S9B).

Methods section:

Line 517-547:

Targeted Lipidomic (PEs) Analysis

Samples preparation

In order to exclude the effect of injury characteristics, we conducted propensity score matching of age, ISS and TBI between resolving (n=14, one sample was not available) and non-resolving male patients (n=15) in PAMPer trial (Table S9). We also included samples from 8 non-fasting healthy subjects as a control group for comparison. Plasma samples from trauma patients of 72h and healthy subjects were analysed in the metabolic core at university of Pittsburgh. Metabolic quenching, lysis, and lipid extraction was performed via Folch extraction. Briefly, 400µL of water, 500µL methanol and 1mL chloroform was added to 100µL plasma and spiked with 5µL PE-UltimateSPLASH deuterated internal standard mix (Avanti Polar Lipids – 330826 Birmingham, AL). Samples were vortexed for 2 minutes

and rested on ice for 10 minutes before phase separation via centrifugation at 3000 x g for 25 minutes at 4°C. 800µL of organic phase was dried to completion under nitrogen gas and resuspended in 1:1 acetonitrile:isopropanol. 2µL of sample was subjected to online LC-MS analysis. Calibration curves were prepared using purified PE species with side chain lengths: 16:0-18:2, 16:0-22:6, 16:0-20:4, 18:0-18:1, and 18:0-22:6 by serial dilution from 15µM down to 0.117µM for absolute quantification.

LC-HRMS Method

Analyses were performed by untargeted LC-HRMS. Briefly, Samples were injected via a Thermo Vanquish UHPLC and separated over a reversed phase Thermo Accucore C-18 column (2.1×100mm, 5µm particle size) maintained at 55°C. For the 30 minute LC gradient, the mobile phase consisted of the following: solvent A (50:50 H₂O:ACN 10mM ammonium acetate / 0.1% acetic acid) and solvent B (90:10 IPA:ACN 10mM ammonium acetate / 0.1% acetic acid). Initial loading condition is 30% B. The gradient was the following: Over 2 minutes, increase to 43%B, continue increasing to 55%B over 0.1 minutes, continue increasing to 65%B over 10 minutes, continue increasing to 85%B over 6 minutes, and finally increasing to 100% over 2 minutes. Hold at 100% for 5 minutes, followed by equilibration at 30%B for 5 minutes. The Thermo IDX tribrid mass spectrometer was operated in positive ESI mode. A data-dependent MS² method scanning in Full MS mode from 200 to 1500 m/z at 120,000 resolution with an AGC target of 5e4 for triggering ms² fragmentation using stepped HCD collision energies at 20,40,and 60% in the orbitrap at 15,000 resolution. Source ionization settings were set to 3.5 kV for spray voltage in positive mode. Source gas parameters were 35 sheath gas, 5 auxiliary gas at 300°C, and 1 sweep gas. Calibration was performed prior to analysis using the Pierce™ FlexMix Ion Calibration Solutions (Thermo Fisher Scientific). Standard peak areas were then extracted manually using Quan Browser (Thermo Fisher Xcalibur ver. 2.7), normalized to deuterated internal standard peak area and converted to concentrations using the calibration curves. The calibration curves of 5 PE species were shown below:

Question 3: Furthermore, disclosure of methodological details such as internal standard information must be included.

Response: We are pleased to provide more information about the methodological details of the Metabolon platform. The methods of the targeted high-resolution LC-HRMS (Platform 2, PF2) Lipidomic Analysis could be found above in the response to Question 2. 5 PE species as the internal standard were purchased (Avanti Polar Lipids – 330826 Birmingham, AL) and sent to the metabolic core in the university of Pittsburgh.

As for the method of LC-MS/MS used by Metabolon, the company shared a recently approved patent with us about internal standards. Thus, we modified our description in methods section as below (highlighted by red color):

Global lipidomic profiling

Lipidomic profiling was performed through the Complex Lipid Panel™ technique at Metabolon (Metabolon Inc, Morrisville, NC 27560, USA). Briefly; lipids were extracted from the plasma using automated BUME extraction (Löfgren et al., 2012). Samples were analyzed using differential mobility spectrometry (DMS) interface (SCIEX) and a high flow LC-30AD solvent delivery unit (Shimadzu). The analysis consists of two separate injections on two instruments, each sample was run once on the platform using methods that combine DMS ‘on’ and ‘off’ as well as positive and negative ionization modes. The following lipid classes were quantified with i) DMS ‘on’ and in negative ionization mode: PC, PE, LPC, LPE, ii) DMS ‘on’ and in positive ionization mode: SM, iii) DMS ‘off’ and in negative ionization mode: FFA, iv) DMS ‘off’ and in positive ionization mode: TAG, DAG, CE, CER. *Internal standards were selected based on the combination of carbon length and the number of double bonds i.e. CE (8), CER (8), SM (4), DAG (8), TAG (8), MAG (1), LPC (1), LPE (1), PC (10), PE (8) and PI (8). The detailed information about the synthesis and exact structures of internal standards can be found in patent (<https://patents.google.com/patent/US11181535B2/en>, Table 1-8).* Lipid class concentrations were calculated, and fatty acid (FA) compositions were determined by calculating the proportion of each class comprised by summation of individual FAs. All the lipid quantifications were median-centered and missing values were minimum-imputed per lipid species.

Lipid species concentrations were background-subtracted using the concentrations detected in process blanks (water extracts) and run day normalized. Background levels were estimated/calculated from the median levels of the three process blanks (water) if there were detectable levels in at least 2 of the 3 blanks in each batch. The background level was subtracted from each sample in the batch prior to any run day normalization.

The QC sample was generated by combining a small aliquot from the entire set of samples into a single pooled CMTX (Client Sample matrix). Four aliquots of the CMTX were run on each plate of 36 samples. One each was injected at the beginning and end of the run, with the other two roughly evenly spaced between the remaining samples. The internal standard was run multiple times throughout the experiment. Instrument variability was evaluated by calculating median relative SD (RSD) from the quality control sample matrix. The median RSD values for 14 lipid classes can be found in Table S8.

Reviewer comments, further round review–

Reviewer #5 (Remarks to the Author):

The authors responses to the comments of reviewer 1 were for the most part satisfactory

Question 1 was adequately addressed

Question 2 was adequately addressed

Question 3 was addressed to some extent however if the question states "Furthermore, disclosure of methodological details such as internal standard information must be included." This has not been fully addressed.

I believe that when the experimental methods are reported, there is an expectation that someone, experienced in the field, could replicate the experiment. This is not the case with the Metabolon data. They do not provide details of which internal standards, nor the concentrations, were used in these analyses. The patent that is referred to provides many examples and lists of lipid standards, but this does not allow someone to replicate these results. Ideally, we would know exactly which internal standard (specific species) was used to quantify each and every lipid species. This data is not provided, and I cannot really see any reason for Metabolon withholding this information.

As it stands the only way for someone to repeat these analyses is to pay Metabolon to do this and even then, there is no guarantee they will use the same set of internal standards. I think this should be a requirement for commercial companies to provide full details for their analyses just as all academic researchers are required to do.

This should be put to Metabolon (and other companies) and if they do not agree, then it should be made clear their data will not be accepted.

I appreciate that this may be an issue outside the control of the authors. However, until this is made clear then commercial companies will continue to provide substandard reporting of their analyses.

Ms. No.: NCOMMS-20-46047D

Title: Lipidomic Signatures Align with Inflammatory Patterns and Outcomes in Critical Illness

Summary of the major concerns and general response

We thank the reviewers for the time and effort they put into re-reviewing our paper. We are also grateful for the opportunity to clarify a few points in the paper.

Point-by-point Responses to the Reviewers' Comments

REVIEWER COMMENTS

Reviewer #5 (Remarks to the Author):

The authors responses to the comments of reviewer 1 were for the most part satisfactory

Question 1 was adequately addressed

Question 2 was adequately addressed

Question 3 was addressed to some extent however if the question states “Furthermore, disclosure of methodological details such as internal standard information must be included.” This has not been fully addressed.

I believe that when the experimental methods are reported, there is an expectation that someone, experienced in the field, could replicate the experiment. This is not the case with the Metabolon data. They do not provide details of which internal standards, nor the concentrations, were used in these analyses. The patent that is referred to provides many examples and lists of lipid standards, but this does not allow someone to replicate these results. Ideally, we would know exactly which internal standard (specific species) was used to quantify each and every lipid species. This data is not provided, and I cannot really see any reason for Metabolon withholding this information.

As it stands the only way for someone to repeat these analyses is to pay Metabolon to do this and even then, there is no guarantee they will use the same set of internal standards. I think this should be a requirement for commercial companies to provide full details for their analyses just as all academic researchers are required to do.

This should be put to Metabolon (and other companies) and if they do not agree, then it should be made clear their data will not be accepted.

I appreciate that this may be an issue outside the control of the authors. However, until this is made clear then commercial companies will continue to provide substandard reporting of their analyses.

Response:

We appreciate the reviewer’s point and therefore had further discussions with Metabolon. Metabolon uses a panel that has an expanded set of internal standards, containing over 50 deuterium-labeled lipid molecular species across 13 lipid classes that mimic the biochemistry found in human plasma. These standards were developed by SCIEX, in collaboration with Avanti Polar Lipids and Metabolon Inc. Further details can be found in the patent literature (<https://patents.google.com/patent/US11181535B2/en>, Table 1-8). This is now commercially available and metabolon has consistently used this platform since 2015 (<https://sciex.com/products/consumables/lipidyzer-platform-kits>). The Internal Standards Kit is comprised of 13 lipid classes and includes labeled molecular species for each class: Lysophosphatidylcholines (LPC), Lysophosphatidylethanolamines (LPE), Phosphatidylcholines (PC), Phosphatidylethanolamines (PE), Sphingomyelins (SM), Diacylglycerols (DAG), Triacylglycerols

(TAG), Free Fatty Acids (FFA), Cholesteryl Esters (CE), Ceramides (CER), Dihydroceramides (DCER), Hexosylceramides (HCER) and Lactosylceramides (LCER) (<https://sciex.com/products/consumables/lipidyzer-platform-kits>). Copied below the table highlights the details of the Kit.

Supplementary Data10 List of internal standards and their concentration in global lipidomic platform

Lipid class	Internal Standards	MW (g.mol ⁻¹)	Concentration (mg.L ⁻¹)
CE	dCE(16:0)	631.62	0.13
	dCE(16:1)	629.61	0.13
	dCE(18:1)	657.64	0.57
	dCE(18:2)	655.62	1.43
	dCE(20:3)	681.64	0.15
	dCE(20:4)	679.62	0.18
	dCE(20:5)	677.61	0.18
	dCE(22:6)	703.62	0.22
DAG	dDAG(16:0_16:0)	577.56	0.004
	dDAG(16:0_18:0)	605.59	0.005
	dDAG(16:0_18:1)	603.57	0.006
	dDAG(16:0_18:2)	601.56	0.005
	dDAG(16:0_18:3)	599.54	0.00135
	dDAG(16:0_20:4)	625.56	0.0015
	dDAG(16:0_20:5)	623.54	0.00145
	dDAG(16:0_22:6)	649.56	0.0016
CER	dCER(d16:0)	546.97	0.02
DCER	dDCER(16:0)	548.99	0.004
HCER	dHCER(16:0)	709.11	0.03
LCER	dLCER(16:0)	871.25	0.03
SM	dSM(16:0)	709.61	0.1
	dSM(18:1)	735.62	0.1
	dSM(24:0)	821.73	0.1
	dSM(24:1)	819.72	0.1
LPC	dLPC(16:0)	504.69	0.1
LPE	dLPE(18:0)	486.64	0.05
PC	dPC(16:0_16:1)	740.6	0.0575
	dPC(16:0_18:1)	768.63	0.2525
	dPC(16:0_18:2)	766.62	0.255
	dPC(16:0_18:3)	764.6	0.065
	dPC(16:0_20:3)	792.63	0.0725
	dPC(16:0_20:4)	790.62	0.2775
	dPC(16:0_20:5)	788.6	0.07
	dPC(16:0_22:4)	818.65	0.075
	dPC(16:0_22:5)	816.63	0.0775
	dPC(16:0_22:6)	814.62	0.145
	PE	dPE(18:0_18:1)	750.59
dPE(18:0_18:2)		748.58	0.01
dPE(18:0_18:3)		746.56	0.0021

	dPE(18:0_20:3)	774.59	0.0027
	dPE(18:0_20:4)	772.58	0.01
	dPE(18:0_20:5)	770.56	0.0022
	dPE(18:0_22:5)	798.6	0.0024
	dPE(18:0_22:6)	796.58	0.01
TAG	dTAG50:1-FA16:0	841.81	0.13
	dTAG52:1-FA18:0	869.84	0.14
	dTAG52:2-FA18:1	867.82	0.14
	dTAG52:3-FA18:2	865.8	0.14
	dTAG52:4-FA18:3	863.79	0.04
	dTAG54:4-FA20:3	891.82	0.04
	dTAG54:5-FA20:4	889.8	0.04
	dTAG56:7-FA22:6	913.8	0.038

Please note, we further validated the key finding of our study using targeted LC-MS assay in our core laboratory at the University of Pittsburgh. These two independent platforms correlated well on, at least in quantifying these 5 PE species.

We have added this information into the following sections of the manuscript:

Plasma lipidomic profiling

Lipidomic profiling was performed through the Complex Lipid Panel™ technique at Metabolon (Metabolon Inc, Morrisville, NC 27560, USA). Briefly; lipids were extracted from the plasma using automated BUME extraction⁵³. Samples were analyzed using differential mobility spectrometry (DMS) interface (SCIEX) and a high flow LC-30AD solvent delivery unit (Shimadzu). Each sample was run once on the platform using a method that combines DMS ‘on’ and ‘off’ as well as positive and negative ionization modes. The following lipid classes were quantified with i) DMS ‘on’ and in negative ionization mode: PC, PE, LPC, LPE, ii) DMS ‘on’ and in positive ionization mode: SM, iii) DMS ‘off’ and in negative ionization mode: FFA, iv) DMS ‘off’ and in positive ionization mode: TAG, DAG, CE, CER. **The internal standards were selected based on the combination of carbon length and the number of double bonds. Metabolon maintains assay-specific internal standards based on superiority compared to single standards.** The panel has an expanded set of internal standards, containing over 50 deuterium-labeled lipid molecular species across 14 lipid classes that mimic the biochemistry found in human plasma. These standards were developed by SCIEX, in collaboration with Avanti Polar Lipids and Metabolon Inc (<https://sciex.com/products/consumables/lipidyzer-platform-kits>). Full list of internal standards can be found in **supplementary Data10**. Further details can be found in the patent literature (<https://patents.google.com/patent/US11181535B2/en>, Table 1-8). Lipid class

concentrations were calculated, and fatty acid (FA) compositions were determined by calculating the proportion of each class comprised by summation of individual FAs.

Data Availability Statement (clear explanations as to why internal standards are not made publicly available and whether/how others can get access to them)

The individual internal standard of Plasma lipidomic profiling by Metabolon Inc (Morrisville, NC 27560, USA) is commercially available(<https://sciex.com/products/consumables/lipidyzer-platform-kits>) and can be found in **supplementary Data10**.